# Understanding Self-supervised Learning with Dual Deep Networks

## Abstract

We propose a novel theoretical framework to understand self-supervised learning methods that employ dual pairs of deep ReLU networks (e.g., SimCLR, BYOL). First, we prove that in each SGD update of SimCLR, the weights at each layer are updated by a *covariance operator* that specifically amplifies initial random selectivities that vary across data samples but survive averages over data augmentations. We show this leads to the emergence of hierarchical features, if the input data are generated from a hierarchical latent tree model. With the same framework, we also show analytically that in BYOL, the combination of BatchNorm and a predictor network creates an implicit contrastive term, acting as an approximate covariance operator. Additionally, for linear architectures we derive exact solutions for BYOL that provide conceptual insights into how BYOL can learn useful non-collapsed representations without any contrastive terms that separate negative pairs. Extensive ablation studies justify our theoretical findings.

## 1 Introduction

While self-supervised learning (SSL) has achieved great empirical success across multiple domains, including computer vision (He et al., 2020; Goyal et al., 2019; Chen et al., 2020a; Grill et al., 2020; Misra and Maaten, 2020; Caron et al., 2020), natural language processing (Devlin et al., 2018), and speech recognition (Wu et al., 2020; Baevski and Mohamed, 2020; Baevski et al., 2019), its theoretical understanding remains elusive, especially when multi-layer nonlinear deep networks are involved (Bahri et al., 2020). Unlike supervised learning (SL) that deals with labeled data, SSL learns meaningful structures from randomly initialized networks without human-provided labels.

In this paper, we propose a systematic theoretical analysis of SSL with deep ReLU networks. Our analysis imposes no parametric assumptions on the input data distribution and is applicable to state-of-the-art SSL methods that typically involve two parallel (or *dual*) deep ReLU networks during training (e.g., SimCLR (Chen et al., 2020a), BYOL (Grill et al., 2020), etc). We do so by developing an analogy between SSL and a theoretical framework for analyzing supervised learning, namely the student-teacher setting (Tian, 2020; Allen-Zhu and Li, 2020; Lampinen and Ganguli, 2018; Saad and Solla, 1996), which also employs a pair of dual networks. Our results indicate that SimCLR weight updates at every layer are amplified by a fundamental positive semi definite (PSD) *covariance operator* that only captures feature variability across data points that *survive* averages over data augmentation procedures designed in practice to scramble semantically unimportant features (e.g. random image crops, blurring or color distortions (Falcon and Cho, 2020; Kolesnikov et al., 2019; Misra and Maaten, 2020; Purushwalkam and Gupta, 2020)). This covariance operator provides a principled framework to study how SimCLR amplifies initial random selectivity to obtain distinctive features that vary *across* samples after surviving averages over data-augmentations.

Based on the covariance operator, we further show that (1) in a two-layer setting, a top-level covariance operator helps accelerate the learning of low-level features, and (2) when the data are generated by a hierarchical latent tree model, training deep ReLU networks leads to an emergence of the latent variables in its intermediate layers. We also analyze how BYOL might work *without negative pairs*. First we show analytically that an interplay between the zero-mean operation in BatchNorm and the extra predictor in the online network creates an implicit contrastive term, consistent with empirical observations in the recent blog (Fetterman and Albrecht, 2020). Note this analysis does not rule out the possibility that BYOL could work with other normalization techniques that don't introduce contrastive terms, as shown recently (Richemond et al., 2020a). To address this, we also derive exact solutions to BYOL in linear networks without *any* normalization, providing insight into how BYOL can learn without contrastive terms induced either by negative pairs or by BatchNorm. Finally, we also discover that reinitializing the predictor every few epochs doesn't hurt BYOL performance, thereby questioning the hypothesis of an optimal predictor in (Grill et al., 2020).

Figure 1: **(a)** Overview of the two SSL algorithms we study in this paper: SimCLR ($\mathcal{W}_1 = \mathcal{W}_2 = \mathcal{W}$, no predictor, NCE Loss) and BYOL ($\mathcal{W}_1$ has an extra predictor, $\mathcal{W}_2$ is a moving average), **(b)** Detailed Notation.

To the best of our knowledge, we are the first to provide a systematic theoretical analysis of modern SSL methods with deep ReLU networks that elucidates how both data and data augmentation, drive the learning of internal representations across multiple layers.

**Related Work.** Besides SimCLR and BYOL, we briefly mention other concurrent SSL frameworks for vision. MoCo (He et al., 2020; Chen et al., 2020b) keeps a large bank of past representations in a queue as the slow-progressing target to train from. DeepCluster (Caron et al., 2018) and SwAV (Caron et al., 2020) learn the representations by iteratively or implicitly clustering on the current representations and improving representations using the cluster label. (Alwassel et al., 2019) applies similar ideas to multi-modality tasks. In contrast, the literature on the analysis of SSL with dual deep ReLU networks is sparse. (Arora et al., 2019) proposes an interesting analysis of how contrastive learning aids downstream classification tasks, given assumptions about data generation. However, it does not explicitly analyze the learning of representations in deep networks.

## 2   OVERALL FRAMEWORK

**Notation.** Consider an $L$-layer ReLU network obeying $\boldsymbol{f}_l = \psi(\tilde{\boldsymbol{f}}_l)$ and $\tilde{\boldsymbol{f}}_l = W_l \boldsymbol{f}_{l-1}$ for $l = 1, \ldots L$. Here $\tilde{\boldsymbol{f}}_l$ and $\boldsymbol{f}_l$ are $n_l$ dimensional pre-activation and activation vectors in layer $l$, with $\boldsymbol{f}_0 = \boldsymbol{x}$ being the input and $\boldsymbol{f}_L = \tilde{\boldsymbol{f}}_L$ the output (no ReLU at the top layer). $W_l \in \mathbb{R}^{n_l \times n_{l-1}}$ are the weight matrices, and $\psi(\boldsymbol{u}) := \max(\boldsymbol{u}, 0)$ is the element-wise ReLU nonlinearity. We let $\mathcal{W} := \{W_l\}_{l=1}^L$ be all network weights. We also denote the gradient of any loss function with respect to $\boldsymbol{f}_l$ by $\boldsymbol{g}_l \in \mathbb{R}^{n_l}$, and the derivative of the output $\boldsymbol{f}_L$ with respect to an earlier pre-activation $\tilde{\boldsymbol{f}}_l$ by the Jacobian matrix $J_l(\boldsymbol{x}; \mathcal{W}) \in \mathbb{R}^{n_L \times n_l}$, as both play key roles in backpropagation (Fig. 1(b)).

**An analogy between self-supervised and supervised learning: the dual network scenario.** Many recent successful approaches to self-supervised learning (SSL), including SimCLR (Chen et al., 2020a), BYOL (Grill et al., 2020) and MoCo (He et al., 2020), employ a dual "Siamese-like" pair (Koch et al., 2015) of such networks (Fig. 1(b)). Each network has its own set of weights $\mathcal{W}_1$ and $\mathcal{W}_2$, receives respective inputs $\boldsymbol{x}_1$ and $\boldsymbol{x}_2$ and generates outputs $\boldsymbol{f}_{1,L}(\boldsymbol{x}_1; \mathcal{W}_1)$ and $\boldsymbol{f}_{2,L}(\boldsymbol{x}_2; \mathcal{W}_2)$. The pair of inputs $\{\boldsymbol{x}_1, \boldsymbol{x}_2\}$ can be either positive or negative, depending on how they are sampled. For a positive pair, a *single* data point $\boldsymbol{x}$ is drawn from the data distribution $p(\cdot)$, and then two augmented views $\boldsymbol{x}_1$ and $\boldsymbol{x}_2$ are drawn from a conditional augmentation distribution $p_{\mathrm{aug}}(\cdot|\boldsymbol{x})$. Possible image augmentations include random crops, blurs or color distortions, that ideally preserve semantic content useful for downstream tasks. In contrast, for a negative pair, two *different* data points $\boldsymbol{x}, \boldsymbol{x}' \sim p(\cdot)$ are sampled, and then each are augmented independently to generate $\boldsymbol{x}_1 \sim p_{\mathrm{aug}}(\cdot|\boldsymbol{x})$ and $\boldsymbol{x}_2 \sim p_{\mathrm{aug}}(\cdot|\boldsymbol{x}')$. For SimCLR, the dual networks have tied weights with $\mathcal{W}_1 = \mathcal{W}_2$, and a loss function is chosen to encourage the representation of positive (negative) pairs to become similar (dissimilar). In BYOL, only positive pairs are used, and the first network $\mathcal{W}_1$, called the online network, is trained to match the output of the second network $\mathcal{W}_2$ (the target), using an additional layer named *predictor*. The target network ideally provides training targets that can improve the online network's representation and does not contribute a gradient. The improved online network is gradually incorporated into the target network, yielding a bootstrapping procedure.

Our fundamental goal is to analyze the mechanisms governing how SSL methods like SimCLR and BYOL lead to the emergence of meaningful intermediate features, starting from random initializations, and how these features depend on the data distribution $p(\boldsymbol{x})$ and augmentation procedure $p_{\mathrm{aug}}(\cdot|\boldsymbol{x})$. Interestingly, the analysis of *supervised* learning (SL) often employs a similar dual network scenario, called *teacher-student setting* (Tian, 2020; Allen-Zhu and Li, 2020; Lampinen and Ganguli, 2018; Saad and Solla, 1996), where $\mathcal{W}_2$ are the ground truth weights of a *fixed* teacher network, which generates outputs in response to random inputs. These input-output pairs constitute training data for the first network, which is a student network. Only the student network's weights

$\mathcal{W}_1$ are trained to match the target outputs provided by the teacher. This yields an interesting mathematical parallel between SL, in which the teacher is fixed and only the student evolves, and SSL, in which both the teacher and student evolve with potentially different dynamics. This mathematical parallel opens the door to using techniques from SL (e.g., (Tian, 2020)) to analyze SSL.

**Gradient of $\ell_2$ loss for dual deep ReLU networks.** As seen above, the (dis)similarity of representations between a pair of dual networks plays a key role in both SSL and SL. We thus consider minimizing a simple measure of dissimilarity, the squared $\ell_2$ distance $r := \frac{1}{2}\|\boldsymbol{f}_{1,L} - \boldsymbol{f}_{2,L}\|^2$ between the final outputs $\boldsymbol{f}_{1,L}$ and $\boldsymbol{f}_{2,L}$ of two multi-layer ReLU networks with weights $\mathcal{W}_1$ and $\mathcal{W}_2$ and inputs $\boldsymbol{x}_1$ and $\boldsymbol{x}_2$. Without loss of generality, we only analyze the gradient w.r.t $\mathcal{W}_1$. For each layer $l$, we first define the *connection* $K_l(\boldsymbol{x})$, a quantity that connects the bottom-up feature vector $\boldsymbol{f}_{l-1}$ with the top-down Jacobian $J_l$, which both contribute to the gradient at weight layer $l$.

**Definition 1** (The connection $K_l(\boldsymbol{x})$). *The connection $K_l(\boldsymbol{x}; \mathcal{W}) := \boldsymbol{f}_{l-1}(\boldsymbol{x}; \mathcal{W}) \otimes J_l^{\mathsf{T}}(\boldsymbol{x}; \mathcal{W}) \in \mathbb{R}^{n_l n_{l-1} \times n_L}$. Here $\otimes$ is the Kronecker product.*

**Theorem 1** (Squared $\ell_2$ Gradient for dual deep ReLU networks). *The gradient $g_{W_l}$ of $r$ w.r.t. $W_l \in \mathbb{R}^{n_l \times n_{l-1}}$ for a single input pair $\{\boldsymbol{x}_1, \boldsymbol{x}_2\}$ is (here $K_{1,l} := K_l(\boldsymbol{x}_1; \mathcal{W}_1)$ and $K_{2,l} := K_l(\boldsymbol{x}_2; \mathcal{W}_2)$):*

$$g_{W_l} = \text{vec}\left(\partial r / \partial W_{1,l}\right) = K_{1,l}\left[K_{1,l}^{\mathsf{T}}\text{vec}(W_{1,l}) - K_{2,l}^{\mathsf{T}}\text{vec}(W_{2,l})\right]. \tag{1}$$

We used vectorized notation for the gradient $g_{W_l}$ and weights $W_l$ to emphasize certain theoretical properties of SSL learning below. The equivalent matrix form is $\partial r / \partial W_{1,l} = J_{1,l}^{\mathsf{T}}\left[J_{1,l}W_{1,l}\boldsymbol{f}_{1,l-1} - J_{2,l}W_{2,l}\boldsymbol{f}_{2,l-1}\right]\boldsymbol{f}_{1,l-1}^T$. See Appendix for proofs of all theorems in main text.

## 3 ANALYSIS OF SIMCLR

As discussed above, SimCLR (Chen et al., 2020a) employs both positive and negative input pairs, and a symmetric network structure with $\mathcal{W}_1 = \mathcal{W}_2 = \mathcal{W}$. Let $\{\boldsymbol{x}_1, \boldsymbol{x}_+\}$ be a positive input pair from $\boldsymbol{x}$, and let $\{\boldsymbol{x}_1, \boldsymbol{x}_{k-}\}$ for $k = 1, \ldots, H$ be $H$ negative pairs. These input pairs induce corresponding squared $\ell_2$ distances in output space, $r_+ := \frac{1}{2}\|\boldsymbol{f}_{1,L} - \boldsymbol{f}_{+,L}\|_2^2$, and $r_{k-} := \frac{1}{2}\|\boldsymbol{f}_{1,L} - \boldsymbol{f}_{k-,L}\|_2^2$.

We consider three different contrastive losses, **(1)** the simple contrastive loss $L_{\text{simp}} := r_+ - r_-$, **(2)** (soft) Triplet loss $L_{\text{tri}}^{\tau} := \tau\log(1 + e^{(r_+ - r_- + r_0)/\tau})$ (here $r_0 \geq 0$ is the margin). Note that $\lim_{\tau \to 0} L_{\text{tri}}^{\tau} = \max(r_+ - r_- + r_0, 0)$ (Schroff et al., 2015), **(3)** InfoNCE loss (Oord et al., 2018):

$$L_{\text{nce}}^{\tau}(r_+, r_{1-}, r_{2-}, \ldots, r_{H-}) := -\log \frac{e^{-r_+/\tau}}{e^{-r_+/\tau} + \sum_{k=1}^{H} e^{-r_{k-}/\tau}} \tag{2}$$

Note that when $\|\boldsymbol{u}\|_2 = \|\boldsymbol{v}\|_2 = 1$, we have $-\frac{1}{2}\|\boldsymbol{u} - \boldsymbol{v}\|_2^2 = \text{sim}(\boldsymbol{u}, \boldsymbol{v}) - 1$ where $\text{sim}(\boldsymbol{u}, \boldsymbol{v}) = \frac{\boldsymbol{u}^{\mathsf{T}}\boldsymbol{v}}{\|\boldsymbol{u}\|_2\|\boldsymbol{v}\|_2}$, and Eqn. 2 reduces to what the original SimCLR uses (the term $e^{-1/\tau}$ cancels out).

For simplicity, we move the analysis of the final layer $\ell_2$ normalization to Appendix A.2. In Appendix F.6 of BYOL Grill et al. (2020) v3, it shows that even without $\ell_2$ normalization, the algorithm still works despite numerical instabilities. In this case, the goal of our analysis is to show that useful weight components grow exponentially in the gradient updates.

One property of these loss functions is important for our analysis:

**Theorem 2** (Common Property of Contrastive Losses). *For loss functions $L \in \{L_{\text{simp}}, L_{\text{tri}}^{\tau}, L_{\text{nce}}^{\tau}\}$, we have $\frac{\partial L}{\partial r_+} > 0$, $\frac{\partial L}{\partial r_{k-}} < 0$ for $1 \leq k \leq H$ and $\frac{\partial L}{\partial r_+} + \sum_{k=1}^{H}\frac{\partial L}{\partial r_{k-}} = 0$.*

With Theorem 1 and Theorem 2, we now present our first main contribution. The gradient in SimCLR is governed by a positive semi-definite (PSD) *covariance operator* at any layer $l$:

**Theorem 3** (Covariance Operator for $L_{\text{simp}}$). *With large batch limit, $W_l$'s update under $L_{\text{simp}}$ is:*

$$W_l(t+1) = W_l(t) + \alpha\Delta W_l(t), \quad \text{where } \text{vec}(\Delta W_l(t)) = \text{OP}_l^{\text{simp}}(\mathcal{W})\text{vec}(W_l(t)). \tag{3}$$

*where $\text{OP}_l^{\text{simp}}(\mathcal{W}) := \mathbb{V}_{\boldsymbol{x} \sim p(\cdot)}[\bar{K}_l(\boldsymbol{x}; \mathcal{W})] \in \mathbb{R}^{n_l n_{l-1} \times n_l n_{l-1}}$ is the* covariance operator *for $L_{\text{simp}}$, $\bar{K}_l(\boldsymbol{x}; \mathcal{W}) := \mathbb{E}_{\boldsymbol{x}' \sim p_{\text{aug}}(\cdot|\boldsymbol{x})}[K_l(\boldsymbol{x}'; \mathcal{W})]$ is the expected connection under the augmentation distribution, conditional on the datapoint $\boldsymbol{x}$ and $\alpha$ is the learning rate.*

**Theorem 4** (Covariance Operator for $L_{\text{tri}}^{\tau}$ and $L_{\text{nce}}^{\tau}$ ($H = 1$, single negative pair)). *Let $r := \frac{1}{2}\|\boldsymbol{f}_L(\boldsymbol{x}) - \boldsymbol{f}_L(\boldsymbol{x}')\|_2^2$. The covariance operator $\text{OP}_l(\mathcal{W}) = \frac{1}{2}\mathbb{V}_{\boldsymbol{x},\boldsymbol{x}' \sim p(\cdot)}^{\xi}[\bar{K}_l(\boldsymbol{x}) - \bar{K}_l(\boldsymbol{x}')] + \text{corr}$, where $\text{corr} := \mathcal{O}(\mathbb{E}_{\boldsymbol{x},\boldsymbol{x}' \sim p(\cdot)}[\sqrt{r(\boldsymbol{x},\boldsymbol{x}')\text{tr}\mathbb{V}_{\boldsymbol{x}'' \sim p_{\text{aug}}(\cdot|\boldsymbol{x})}[\boldsymbol{f}_L(\boldsymbol{x}'')]}])$. For $L_{\text{tri}}^{\tau}$, $\xi(r) = \frac{e^{-(r-r_0)/\tau}}{1+e^{-(r-r_0)/\tau}}$ (and $\lim_{\tau \to 0} \xi(r) = \mathbb{I}(r \leq r_0)$). For $L_{\text{nce}}^{\tau}$, $\xi(r) = \frac{1}{\tau}\frac{e^{-r/\tau}}{1+e^{-r/\tau}}$. For $L_{\text{simp}}$, $\xi(r) \equiv 1$ and $\text{corr} = 0$.*

Figure 2: Overview of Sec. 4. **(a)** To analyze the functionality of the *covariance operator* $\mathbb{V}_{z_0}\left[\bar{K}_l(z_0)\right]$ (Eqn. 3), we assume that Nature generates the data from a certain generative model with latent variable $z_0$ and $z'$, while data augmentation takes $\boldsymbol{x}(z_0, z')$, changes $z'$ but keeps $z_0$ intact. **(b)** *Sec. 4.1*: one layer one neuron example. **(c)** *Sec. 4.2*: two-layer case where $\mathbb{V}[\bar{K}_1]$ and $\mathbb{V}[\bar{K}_2]$ interplay. **(d)** *Sec. 4.3*: Hierarchical Latent Tree Models and deep ReLU networks trained with SimCLR. A latent variable $z_\mu$, and its corresponding nodes $\mathcal{N}_\mu$ in multi-layer ReLU side, covers a subset of input $\boldsymbol{x}$, resembling local receptive fields in ConvNet.

Above, we use $\mathrm{Cov}^\xi[X, Y] := \mathbb{E}\left[\xi(X,Y)(X - \mathbb{E}[X])(Y - \mathbb{E}[Y])^\intercal\right]$ and $\mathbb{V}^\xi[X] := \mathrm{Cov}^\xi[X, X]$ ($\mathrm{Cov}[X, Y]$ means $\xi(\cdot) \equiv 1$). The *covariance operator* $\mathrm{OP}_l(\mathcal{W})$ is a time-varying PSD matrix over the *entire* training procedure. Therefore, all its eigenvalues are non-negative and at any time $t$, $W_l$ is most amplified along its largest eigenmodes. Intuitively, $\mathrm{OP}_l(\mathcal{W})$ ignores different views of the same sample $\boldsymbol{x}$ by averaging over the augmentation distribution to compute $\bar{K}_l(\boldsymbol{x})$, and then computes the expected covariance of this *augmentation averaged* connection with respect to the data distribution $p(\boldsymbol{x})$. Thus, at all layers, any variability in the connection across different data points, that survives augmentation averages, leads to weight amplification. This amplification of weights by the PSD data covariance of an augmentation averaged connection constitutes a fundamental description of SimCLR learning dynamics for *arbitrary* data and augmentation distributions.

# 4 HOW THE COVARIANCE OPERATOR DRIVES THE EMERGENCE OF FEATURES

To concretely illustrate how the fundamental covariance operator derived in Theorem 3-4 drives feature emergence in SimCLR, we setup the following paradigm for analysis. The input $\boldsymbol{x} = \boldsymbol{x}(z_0, z')$ is assumed to be generated by two groups of latent variables, *class/sample-specific* latents $z_0$ and *nuisance* latents $z'$. We assume data augmentation only changes $z'$ while preserving $z_0$ (Fig. 2(a)). For brevity we use Theorem 3 ($L_{\mathrm{simp}}$), then $\mathrm{OP} = \mathbb{V}_{z_0}[\bar{K}_l(z_0)]$ since $z'$ is averaged out in $\bar{K}_l(z_0)$.

In this setting, we first show that a linear neuron performs PCA within an augmentation preserved subspace. We then consider how nonlinear neurons with local receptive fields (RFs) can learn to detect simple objects. Finally, we extend our analysis to deep ReLU networks exposed to data generated by a hierarchical latent tree model (HLTM), proving that, with sufficient over-parameterization, there exist lucky nodes at initialization whose activation is correlated with latent variables underlying the data, and that SimCLR amplifies these initial lucky representations during learning.

## 4.1 SELF-SUPERVISED LEARNING AND THE SINGLE NEURON: ILLUSTRATIVE EXAMPLES

**A single linear neuron performs PCA in a preserved subspace.** For a single linear neuron ($L = 1, n_L = 1$), the connection in definition 1 is simply $K_1(\boldsymbol{x}) = \boldsymbol{x}$. Now imagine the input space $\boldsymbol{x}$ can be decomposed into the direct sum of a semantically relevant subspace, and its orthogonal complement, which corresponds to a subspace of nuisance features. Furthermore, suppose the augmentation distribution $p_{\mathrm{aug}}(\cdot|\boldsymbol{x})$ is obtained by multiplying $\boldsymbol{x}$ by a random Gaussian matrix that acts *only* in the nuisance subspace, thereby identically preserving the semantic subspace. Then the augmentation averaged connection $\bar{K}_1(\boldsymbol{x}) = Q^s \boldsymbol{x}$ where $Q^s$ is a projection operator onto the semantic subspace. In essence, only the projection of data onto the semantic subspace survives augmentation averaging, as the nuisance subspace is scrambled. Then $\mathrm{OP} = \mathbb{V}_{\boldsymbol{x}}[\bar{K}_1(\boldsymbol{x})] = Q^s \mathbb{V}_{\boldsymbol{x}}[\boldsymbol{x}] Q^{s\intercal}$. Thus the covariance of the data distribution, projected onto the semantic subspace, governs the growth of the weight vector $W_1$, demonstrating SimCLR on a single linear neuron performs PCA within a semantic subspace preserved by data augmentation.

**A single linear neuron cannot detect localized objects**. We now consider a generative model in which data vectors can be thought of as images of objects of the form $\boldsymbol{x}(z_0, z')$ where $z_0$ is an important latent semantic variable denoting object identity, while $z'$ is an unimportant latent variable denoting nuisance features, like object pose or location. The augmentation procedure scrambles pose/position while preserving object identity. Consider a simple concrete example (Fig. 3(a)):

$$\boldsymbol{x}(z_0, z') = \begin{cases} \boldsymbol{e}_{z'} + \boldsymbol{e}_{(z'+1) \bmod d} & z_0 = 1 \\ \boldsymbol{e}_{z'} + \boldsymbol{e}_{(z'+2) \bmod d} & z_0 = 2, \end{cases} \tag{4}$$

Here $0 \le z' \le d - 1$ denotes $d$ discrete translational object positions on a periodic ring and $z_0 \in \{1, 2\}$ denotes two possible objects `11` and `101`. The distribution is uniform both over objects

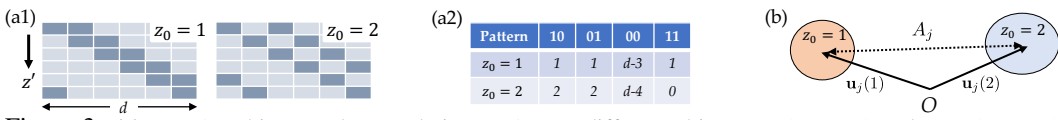

Figure 3: **(a)** Two 1D objects under translation: *(a1)* Two different objects `11` ($z_0 = 1$) and `101` ($z_0 = 2$) located at different locations specified by $z'$. *(a2)* The frequency table for a neuron with local receptive field of size 2. **(b)** In two-layer case (Fig. 2(c)), $\mathbb{V}[\bar{K}_1]$ and $\mathbb{V}[\bar{K}_2]$ interplay in two-cluster data distribution.

and positions: $p(z_0, z') = \frac{1}{2d}$. Augmentation shifts the object to a uniformly random position via $p_{\mathrm{aug}}(z'|z_0) = 1/d$. For a single linear neuron $K_1(\boldsymbol{x}) = \boldsymbol{x}$, and the augmentation-averaged connection is $\bar{K}_1(z_0) = \frac{2}{d}\mathbf{1}$, and is actually independent of object identity $z_0$ (both objects activate two pixels at any location). Thus $\mathrm{OP}_1 = \mathbb{V}_{z_0}\left[\bar{K}_1(z_0)\right] = 0$ and no learning happens.

**A local receptive field (RF) does not help.** In the same generative model, now consider a linear neuron with a local RF of width 2. Within the RF only four patterns can arise: `00`, `01`, `10`, `11`. Taking the expectation over $z'$ given $z_0$ (Fig. 3(a2)) yields $\bar{K}_1(z_0{=}1) = \frac{1}{d}\left[\boldsymbol{x}_{11} + \boldsymbol{x}_{01} + \boldsymbol{x}_{10} + (d-3)\boldsymbol{x}_{00}\right]$ and $\bar{K}_1(z_0{=}2) = \frac{1}{d}\left[2\boldsymbol{x}_{01} + 2\boldsymbol{x}_{10} + (d-4)\boldsymbol{x}_{00}\right]$. Here, $\boldsymbol{x}_{11} \in \mathbb{R}^2$ denotes pattern `11`. This yields

$$\mathrm{OP}_1 = \mathbb{V}_{z_0}\left[\bar{K}_1(z_0)\right] = \frac{1}{4d^2}\boldsymbol{u}\boldsymbol{u}^\mathsf{T} \qquad \text{where } \boldsymbol{u} := \boldsymbol{x}_{11} + \boldsymbol{x}_{00} - \boldsymbol{x}_{01} - \boldsymbol{x}_{10}, \tag{5}$$

and $\mathrm{OP}_1 \in \mathbb{R}^{2\times 2}$ since the RF has width 2. Note that the signed sum of the four pattern vectors in $\boldsymbol{u}$ actually cancel, so that $\boldsymbol{u} = \boldsymbol{0}$, implying $\mathrm{OP}_1 = 0$ and no learning happens. Interestingly, although the conditional distribution of the 4 input patterns depends on the object identity $z_0$ (Fig. 3(a2)), a linear neuron cannot learn to discriminate the objects.

**A nonlinear neuron with local RF can learn to detect object selective features**. With a ReLU neuron with weight vector $\boldsymbol{w}$, from Def. 1, the connection is now $K_1(\boldsymbol{x}, \boldsymbol{w}) = \psi'(\boldsymbol{w}^\mathsf{T}\boldsymbol{x})\boldsymbol{x}$. Suppose $\boldsymbol{w}(t)$ happens to be selective for a *single* pattern $\boldsymbol{x}_p$ (where $p \in \{00, 01, 10, 11\}$), i.e., $\boldsymbol{w}(t)^\mathsf{T}\boldsymbol{x}_p > 0$ and $\boldsymbol{w}(t)^\mathsf{T}\boldsymbol{x}_{p'} < 0$ for $p' \neq p$. The augmentation averaged connection is then $\bar{K}_1(z_0) \propto \boldsymbol{x}_p$ where the proportionality constant depends on object identity $z_0$ and can be read off (Fig. 3(a2)). Since this averaged connection varies with object identity $z_0$ for all $p$, the covariance operator $\mathrm{OP}_1$ is nonzero and is given by $\mathbb{V}_{z_0}\left[\bar{K}_1(z_0)\right] = c_p \boldsymbol{x}_p \boldsymbol{x}_p^\mathsf{T}$ where the constant $c_p > 0$ depends on the selective pattern $p$ and can be computed from Fig. 3(a2). By Theorem 3, the dot product $\boldsymbol{x}_p^\mathsf{T}\boldsymbol{w}(t)$ grows over time:

$$\boldsymbol{x}_p^\mathsf{T}\boldsymbol{w}(t+1) = \boldsymbol{x}_p^\mathsf{T}\left(I_{2\times 2} + \alpha c_p \boldsymbol{x}_p \boldsymbol{x}_p^\mathsf{T}\right)\boldsymbol{w}(t) = \left(1 + \alpha c_p\|\boldsymbol{x}_p\|^2\right)\boldsymbol{x}_p^\mathsf{T}\boldsymbol{w}_j(t) > \boldsymbol{x}_p^\mathsf{T}\boldsymbol{w}_j(t) > 0. \tag{6}$$

Thus the learning dynamics amplifies the initial selectivity to the object selective feature vector $\boldsymbol{x}_p$ in a way that cannot be done with a linear neuron. Note this argument also holds with bias terms and initial selectivity for more than one pattern. Moreover, with a local RF, the probability of weak initial selectivity to local object sensitive features is high, and we may expect amplification of such weak selectivity in real neural network training, as observed in other settings (Williams et al., 2018).

### 4.2 A Two-layer Case with Multiple Hidden Neurons

Now consider a two-layer network ($L = 2$). The hidden layer has $n_1$ ReLU neurons while the output has $n_2$ (Fig. 2(c)). In this case, the augmentation-averaged connection $\bar{K}_1(z_0)$ at the lower layer $l = 1$ can be written as ($d = n_0$ is the input dimension):

$$\bar{K}_1(z_0) = [\boldsymbol{w}_{2,1}\boldsymbol{u}_1^\mathsf{T}(z_0), \boldsymbol{w}_{2,2}\boldsymbol{u}_2^\mathsf{T}(z_0), \ldots, \boldsymbol{w}_{2,n_1}\boldsymbol{u}_{n_1}^\mathsf{T}(z_0)]^\mathsf{T} \in \mathbb{R}^{n_1 d \times n_2} \tag{7}$$

where $\boldsymbol{w}_{1,j} \in \mathbb{R}^d$ and $\boldsymbol{w}_{2,j} \in \mathbb{R}^{n_2}$ are weight vectors into and out of hidden neuron $j$ (Fig. 2(c)), and $\boldsymbol{u}_j(z_0) := \mathbb{E}_{z'|z_0}\left[\boldsymbol{x}(z_0, z')\mathbb{I}(\boldsymbol{w}_{1,j}^\mathsf{T}\boldsymbol{x}(z_0, z') \geq 0)\right] \in \mathbb{R}^d$ is the augmentation average of only those inputs that activate hidden neuron $j$. While the gradient dynamics in SimCLR under $L_{\mathrm{simp}}$ has a close form (Eqn. 65), it is hard to see what happens. Instead, we consider an intuitive sub-case:

**Theorem 5** (Dynamics of two-layer ($W_2$ diagonal)). *If $n_1 = n_2$ and $W_2 = \mathrm{diag}(w_{2,1}, \ldots, w_{2,n_1})$:*

$$\dot{w}_{2,j} = (\boldsymbol{w}_{1,j}^\mathsf{T} A_j \boldsymbol{w}_{1,j})w_{2,j}, \quad \dot{\boldsymbol{w}}_{1,j} = w_{2,j}^2 A_j \boldsymbol{w}_{1,j}, \quad \text{where } A_j := \mathbb{V}_{z_0}[\boldsymbol{u}_j(z_0)]. \tag{8}$$

Note that for ReLU neurons, $A_j$ changes with $\boldsymbol{w}_{1,j}$, while for linear neurons, $A_j$ would be constant, since gating $\psi'(\boldsymbol{w}_{1,j}^\mathsf{T}\boldsymbol{x}) \equiv 1$. It is easy to see that $\mathrm{d}w_{2,j}^2/\mathrm{d}t = \mathrm{d}\|\boldsymbol{w}_{1,j}\|_2^2/\mathrm{d}t$ and thus $w_{2,j}^2 = \|\boldsymbol{w}_{1,j}\|_2^2 + c$ where $c$ is some time-independent constant. Since $A_j$ is always PSD, $\boldsymbol{w}_{1,j}^\mathsf{T} A_j \boldsymbol{w}_{1,j} \geq 0$ and $|w_{2,j}|$ generically increases, which in turn accelerates the dynamics of $\boldsymbol{w}_{1,j}$, which is most amplified at any given time, along the largest eigenvector of $A_j$. This dynamics exhibits *top-down modulation* whereby the top-layer weights accelerate the training of the lower layer.

| Symbol | Definition | Size | Description |
|---|---|---|---|
| $\mathcal{N}_l, \mathcal{Z}_l$ | | | The set of all nodes and all latent variables at layer $l$. |
| $\mathcal{N}_\mu, \mathcal{N}_\mu^{\mathrm{ch}}$ | | | Nodes corresponding to latent variable $z_\mu$. $\mathcal{N}_\mu^{\mathrm{ch}}$ are children under $\mathcal{N}_\mu$. |
| $P_{\mu\nu}$ | $[\mathbb{P}(z_\nu\|z_\mu)]$ | $2 \times 2$ | The top-down transition probability from $z_\mu$ to $z_\nu$. |
| $v_j(z_\mu), \boldsymbol{v}_j$ | $\mathbb{E}_z\left[f_j\|z_\mu\right], [v_j(z_\mu)]$ | scalar, 2 | Expected activation $f_j$ given $z_\mu$ ($z_\mu$'s descendants are marginalized). |
| $\boldsymbol{f}_\mu, \boldsymbol{f}_{\mathcal{N}_\mu^{\mathrm{ch}}}$ | $[f_j]_{j\in\mathcal{N}_\mu}, [f_k]_{k\in\mathcal{N}_\mu^{\mathrm{ch}}}$ | $\|\mathcal{N}_\mu\|, \|\mathcal{N}_\mu^{\mathrm{ch}}\|$ | Activations for all nodes $j \in \mathcal{N}_\mu$ and for the children of $\mathcal{N}_\mu$ |
| $\rho_{\mu\nu}$ | $2\mathbb{P}(z_\nu{=}1\|z_\mu{=}1) - 1$ | scalar in $[-1,1]$ | Polarity of the transitional probability. |
| $\rho_0$ | $\mathbb{P}(z_0 = 1) - \mathbb{P}(z_0 = 0)$ | scalar | Polarity of probability of root latent $z_0$. |
| $s_k$ | $\frac{1}{2}(v_k(1) - v_k(0))$ | scalar | Discrepancy of node $k$ w.r.t its latent variable $z_{\nu(k)}$. |
| $\boldsymbol{a}_\mu$ | $[\rho_{\mu\nu(k)}s_k]_{k\in\mathcal{N}_\mu^{\mathrm{ch}}}$ | $\|\mathcal{N}_\mu^{\mathrm{ch}}\|$ | Child selectivity vector. |

Table 1: Notation for Sec. 4.3 (binary symmetric HLTM).

Previous works (Allen-Zhu and Li, 2020) also mention a similar concept in supervised learning, called "backward feature correction." Here we demonstrate rigorously that a similar behavior can occur in SSL under gradient descent in the 2-layer case when the top layer $W_2$ is diagonal.

As an example, consider a mixture of Gaussians: $\boldsymbol{x} \sim \frac{1}{2}\mathbb{I}(z_0{=}1)N(\boldsymbol{w}_1^*, \sigma^2) + \frac{1}{2}\mathbb{I}(z_0{=}2)N(\boldsymbol{w}_2^*, \sigma^2)$ and let $\Delta\boldsymbol{w}^* := \boldsymbol{w}_1^* - \boldsymbol{w}_2^*$, then in the linear case, $A_j \sim \Delta\boldsymbol{w}^*\Delta\boldsymbol{w}^{*\intercal}$ and $\boldsymbol{w}_{1,j}$ converges to $\pm\Delta\boldsymbol{w}^*$ (Fig. 3(b)). In the nonlinear case with multiple Gaussians, if one of the Gaussians sits at the origin (e.g., background noise), then dependent on initialization, $A_j$ evolves into $\boldsymbol{w}_k^*\boldsymbol{w}_k^{*\intercal}$ for some center $k$, and $\boldsymbol{w}_{1,j} \to \boldsymbol{w}_k^*$. Note this dynamics is insensitive to specific parametric forms of the input data.

## 4.3 DEEP ReLU SSL TRAINING WITH HIERARCHICAL LATENT TREE MODELS (HLTM)

We next study how multi-layer ReLU networks learn from data generated by an HLTM, in which visible leaf variables are sampled via a hierarchical branching diffusion process through a sequence of latent variables starting from a root variable $z_0$ (Fig. 2(d, left)). The HLTM represents a mathematical abstraction of the hierarchical structure real-world objects, which consist of spatially localized parts and subparts, all of which can lie in different configurations or occluded states. See Appendix D.2 for a detailed description and motivation for the HLTM. Simpler versions of the HLTM have been used to mathematically model how both infants and linear neural networks learn hierarchical structure (Saxe et al., 2019). We examine when a multi-layer ReLU network with spatially local RFs can learn the latent generative variables when exposed only to the visible leaf variables (Fig. 2(d, right)).

We define symbols in Tbl. 1. At layer $l$, we have categorical latent variables $\{z_\mu\}$, where $\mu \in \mathcal{Z}_l$ indexes different latent variables. Each $z_\mu$ can take discrete values. The topmost latent variable is $z_0$. Following the tree structure, for $\mu \in \mathcal{Z}_l$ and $\nu_1, \nu_2 \in \mathcal{Z}_{l-1}$, conditional independence holds: $\mathbb{P}(z_{\nu_1}, z_{\nu_2}|z_\mu) = \mathbb{P}(z_{\nu_1}|z_\mu)\mathbb{P}(z_{\nu_2}|z_\mu)$. The final sample $\boldsymbol{x}$ is just the collection of all visible leaf variables (Fig. 2(d)), and thus depends on all latent variables. Corresponding to the hierarchical tree model, each neural network node $j \in \mathcal{N}_l$ maps to a unique $\mu = \mu(j) \in \mathcal{Z}_l$. Let $\mathcal{N}_\mu$ be all nodes that map to $\mu$. For $j \in \mathcal{N}_\mu$, its activation $f_j$ only depends on the value of $z_\mu$ and its descendant latent variables, through input $\boldsymbol{x}$. Define $v_j(z_\mu) := \mathbb{E}_z\left[f_j|z_\mu\right]$ as the expected activation w.r.t $z_\mu$. Given a sample $\boldsymbol{x}$, data augmentation involves *resampling* all $z_\mu$ (which are $z'$ in Fig. 2), fixing the root $z_0$.

**Symmetric Binary HLTM.** Here we consider a symmetric binary case: each $z_\mu \in \{0, 1\}$ and for $\mu \in \mathcal{Z}_l, \nu \in \mathcal{Z}_{l-1}$, $\mathbb{P}(z_\nu=1|z_\mu=1) = \mathbb{P}(z_\nu=0|z_\mu=0) = (1 + \rho_{\mu\nu})/2$, where the *polarity* $\rho_{\mu\nu} \in [-1, 1]$ measures how informative $z_\mu$ is. If $\rho_{\mu\nu} = \pm 1$ then there is no stochasticity in the top-down generation process; $\rho_{\mu\nu} = 0$ means no information in the downstream latents and the posterior of $z_0$ given the observation $\boldsymbol{x}$ can only be uniform. See Appendix for more general cases.

Now we compute covariance operator $\mathrm{OP}_\mu = \mathbb{V}_{z_0}[\bar{K}_\mu(z_0)]$ at different layers, where $\bar{K}_\mu(z_0) = \mathbb{E}_{z'}\left[\boldsymbol{f}_{\mathcal{N}_\mu^{\mathrm{ch}}} \otimes J_\mu^\intercal|z_0\right]$. Here we mainly check the term $\mathbb{E}_{z'}\left[\boldsymbol{f}_{\mathcal{N}_\mu^{\mathrm{ch}}}|z_0\right]$ and assume $J_\mu$ is constant.

**Theorem 6** (Activation covariance in binary HLTM). $\mathbb{V}_{z_0}[\mathbb{E}_{z'}\left[\boldsymbol{f}_{\mathcal{N}_\mu^{\mathrm{ch}}}|z_0\right]] = o_\mu\boldsymbol{a}_\mu\boldsymbol{a}_\mu^\intercal$. *Here* $\boldsymbol{a}_\mu := [\rho_{\mu\nu(k)}s_k]_{k\in\mathcal{N}_\mu^{\mathrm{ch}}}$ *and* $o_\mu := \rho_{0\mu}^2(1 - \rho_{0\mu}^2)$. *If* $\max_{\alpha\beta}|\rho_{\alpha\beta}| < 1$, *then* $\lim_{L\to+\infty}\rho_{0\mu} \to 0$ *for* $\mu \in \mathcal{Z}_l$.

Theorem 6 suggests when $\rho_{0\mu}$ and $\|\boldsymbol{a}_\mu\|$ are large, the covariance $\mathrm{OP}_\mu = o_\mu\boldsymbol{a}_\mu\boldsymbol{a}_\mu^\intercal \otimes J_\mu^\intercal J_\mu$ is large and training is faster. For deep HLTM and deep networks, at lower layers, $\rho_{0\mu} \to 0$ and $P_{0\mu}$ is uniform due to mixing of the Markov Chain, making $\mathrm{OP}_\mu$ small. Thus training in SSL is faster at the top layers where the covariance operators have large magnitude. On the other hand, large $\|\boldsymbol{a}_\mu\|$ implies $s_k := (v_k(1) - v_k(0))/2$ is large, or the expected activation $v_k(z_\nu)$ is *selective* for different values of $z_\nu$ for $\nu \in \mathrm{ch}(\mu)$. Interestingly, this can be achieved by *over-parameterization* ($|\mathcal{N}_\mu| > 1$):

**Theorem 7** (Lucky nodes in deep ReLU networks regarding to binary HLTM at initialization). *Suppose each element of the weights $W_l$ between layer $l + 1$ and $l$ are initialized with* Uniform $\left[-\sigma_w\sqrt{3/|\mathcal{N}_\mu^{\mathrm{ch}}|}, \sigma_w\sqrt{3/|\mathcal{N}_\mu^{\mathrm{ch}}|}\right]$. *There exists $\sigma_l^2$ so that $\mathbb{V}[f_k|z_\nu] \leq \sigma_l^2$ for any $k \in \mathcal{N}_l$. For any $\mu \in \mathcal{Z}_{l+1}$, if $|\mathcal{N}_\mu| = O(\exp(c))$, then with high probability, there exists at least one node $j \in \mathcal{N}_\mu$ so that their* pre-activation gap $\tilde{v}_j(1) - \tilde{v}_j(0) = 2\boldsymbol{w}_j^\mathsf{T}\boldsymbol{a}_\mu > 0$ *and the activations satisfy:*

$$\left|v_j^2(1) - v_j^2(0)\right| \geq 3\sigma_w^2 \left[\frac{1}{4|\mathcal{N}_\mu^{\mathrm{ch}}|} \sum_{k \in \mathcal{N}_\mu^{\mathrm{ch}}} |v_k(1) - v_k(0)|^2 \left(\frac{c+6}{6}\rho_{\mu\nu}^2 - 1\right) - \sigma_l^2\right]. \tag{9}$$

Intuitively, this means that with large polarity $\rho_{\mu\nu}$ (strong top-down signals), randomly initialized over-parameterized ReLU networks yield selective neurons, if the lower layer also contains selective ones. For example, when $\sigma_l = 0$, $c = 9$, if $\rho_{\mu\nu} \geq 63.3\%$ then there is a gap between expected activations $v_j(1)$ and $v_j(0)$, and the gap is larger when the selectivity in the lower layer $l$ is higher. Note that at the lowest layer, $\{v_k\}$ are themselves observable leaf latent variables and are selective by definition. So a bottom-up mathematical induction of latent selectivity will unfold.

If we further assume $J_\mu^\mathsf{T}J_\mu = I$, then after the gradient update, for the "lucky" node $j$ we have:

$$\boldsymbol{a}_\mu^\mathsf{T}\boldsymbol{w}_j(t+1) = \boldsymbol{a}_\mu^\mathsf{T}\left[I + \alpha o_\mu \boldsymbol{a}_\mu \boldsymbol{a}_\mu^\mathsf{T}\right]\boldsymbol{w}_j(t) = (1 + \alpha o_\mu \|\boldsymbol{a}_\mu\|_2^2)\boldsymbol{a}_\mu^\mathsf{T}\boldsymbol{w}_j(t) > \boldsymbol{a}_\mu^\mathsf{T}\boldsymbol{w}_j(t) > 0$$

which means that the pre-activation gap $\tilde{v}_j(1) - \tilde{v}_j(0) = 2\boldsymbol{w}_j^\mathsf{T}\boldsymbol{a}_\mu$ grows over time and the latent variable $z_\mu$ is *learned* (instantiated as $f_j$) during training, even if the network is never supervised with its true value. While here we analyze the simplest case ($J_\mu^\mathsf{T}J_\mu = I$), in practice $J_\mu$ changes over time. Similar to Sec. 4.2, once the top layer starts to have large weights, the magnitude of $J_\mu$ for lower layer becomes larger and training is accelerated.

We implement the HLTM and confirm, as predicted by our theory, that the intermediate layers of deep ReLU networks do indeed learn the latent variables of the HLTM (see Tbl. 2 below).

## 5 ANALYSIS OF INGREDIENTS UNDERLYING BYOL LEARNING

In BYOL, the two networks are no-longer identical and, interestingly, only positive pairs are used for training. The first network with weights $\mathcal{W}_1 = \mathcal{W} := \{\mathcal{W}_{\mathrm{base}}, \mathcal{W}_{\mathrm{pred}}\}$ is an *online* network that is trained to predict the output of the second *target* network with weights $\mathcal{W}_2 = \mathcal{W}'$, using a learnable *predictor* $\mathcal{W}_{\mathrm{pred}}$ to map the online to target outputs (Fig. 1(a) and Fig. 1 in Grill et al. (2020)). In contrast, the target network has $\mathcal{W}' := \{\mathcal{W}'_{\mathrm{base}}\}$, where $\mathcal{W}'_{\mathrm{base}}$ is an exponential moving average (EMA) of $\mathcal{W}_{\mathrm{base}}$: $\mathcal{W}'_{\mathrm{base}}(t+1) = \gamma_{\mathrm{ema}}\mathcal{W}'_{\mathrm{base}}(t) + (1 - \gamma_{\mathrm{ema}})\mathcal{W}_{\mathrm{base}}(t)$. Since BYOL only uses positive pairs, we consider the following loss function:

$$r := \frac{1}{2}\|\boldsymbol{f}_L(\boldsymbol{x}_1; \mathcal{W}) - \boldsymbol{f}_{L'}(\boldsymbol{x}_+; \mathcal{W}')\|_2^2 \tag{10}$$

where the input data are positive pairs: $\boldsymbol{x}_1, \boldsymbol{x}_+ \sim p_{\mathrm{aug}}(\cdot|\boldsymbol{x})$ and $\boldsymbol{x} \sim p(\cdot)$. The two outputs, $\boldsymbol{f}_L(\boldsymbol{x}_1; \mathcal{W})$ and $\boldsymbol{f}_{L'}(\boldsymbol{x}_+; \mathcal{W}')$, are from the online and the target networks, respectively. Note that $L' < L$ due to the presence of an extra predictor on the side of online network ($\mathcal{W}$).

With neither EMA ($\gamma_{\mathrm{ema}} = 0$) *nor* the predictor, $\mathcal{W}' = \mathcal{W}$ and the BYOL update without BN is

$$\mathrm{vec}\left(\Delta W_l\right)_{\mathrm{sym}} = -\mathbb{E}_{\boldsymbol{x}}\left[\mathbb{V}_{\boldsymbol{x}' \sim p_{\mathrm{aug}}(\cdot|\boldsymbol{x})}\left[K_l(\boldsymbol{x}')\right]\right]\mathrm{vec}(W_l) \tag{11}$$

(see App. E.1 for proof). This update only promotes variance minimization in the representations of different augmented views of the same data samples and therefore would yield model collapse.

We now consider the effects played by the extra predictor and BatchNorm (BN) (Ioffe and Szegedy, 2015) in BYOL. Our interest in BatchNorm is motivated by a recent blogpost (Fetterman and Albrecht, 2020). We will see that combining both could yield a sufficient condition to create an implicit contrastive term that could help BYOL learn. As pointed out recently by Richemond et al. (2020a), BYOL can still work using other normalization techniques that do not rely on cross batch statistics (e.g., GroupNorm (Wu and He, 2018), Weight Standardization (Qiao et al., 2019) and careful initialization of affine transform of activations). In Appendix F we derive exact solutions to BYOL for linear architectures without any normalization, to provide conceptual insights into how BYOL can still learn *without* contrastive terms, at least in the linear setting. Here we focus on BatchNorm, leaving an analysis of other normalization techniques in nonlinear BYOL settings for future work.

When adding predictor, Theorem 1 can still be applied by adding identity layers on top of the target network $\mathcal{W}'$ so that the online and the target networks have the same depth. Theorem 5 in (Tian,

2018) demonstrates this version of BN shifts the downward gradients so their mini-batch mean is $\mathbf{0}$:

$$\tilde{\tilde{\boldsymbol{g}}}_l^i := \boldsymbol{g}_l^i - \frac{1}{|B|} \sum_{i \in B} \boldsymbol{g}_l^i = \boldsymbol{g}_l^i - \bar{\boldsymbol{g}}_l \tag{12}$$

Here $\boldsymbol{g}_l^i$ is the $i$-th sample in a batch and $\bar{\boldsymbol{g}}_l$ is the batch average (same for $\bar{\boldsymbol{f}}_l$). Backpropagating through this BN (vs. just subtracting the mean only in the forward pass[1]), leads to a correction term:

**Theorem 8.** *If (1) the network is linear from layer $l$ to the topmost and (2) the downward gradient $\boldsymbol{g}_l$ undergoes Eqn. 12, then with large batch limits, the correction of the update is[2] (for brevity, dependency on $\mathcal{W}$ is omitted, while dependency on $\mathcal{W}'$ is made explicit):*

$$\mathrm{vec}(\delta W_l^{\mathrm{BN}}) = \mathbb{E}_{\boldsymbol{x}} \left[ \bar{K}_l(\boldsymbol{x}) \right] \left\{ \mathbb{E}_{\boldsymbol{x}} \left[ \bar{K}_l^\mathsf{T}(\boldsymbol{x}) \right] \mathrm{vec}(W_l) - \mathbb{E}_{\boldsymbol{x}} \left[ \bar{K}_l^\mathsf{T}(\boldsymbol{x}; \mathcal{W}') \right] \mathrm{vec}(W_l') \right\} \tag{13}$$

*and the corrected weight update is $\widetilde{\Delta W_l} := \Delta W_l + \delta W_l^{\mathrm{BN}}$. Using Eqn. 11, we have:*

$$\mathrm{vec}(\widetilde{\Delta W_l}) = \mathrm{vec}(\Delta W_l)_{\mathrm{sym}} - \mathbb{V}_{\boldsymbol{x}} \left[ \bar{K}_l(\boldsymbol{x}) \right] \mathrm{vec}(W_l) + \mathrm{Cov}_{\boldsymbol{x}} \left[ \bar{K}_l(\boldsymbol{x}), \bar{K}_l(\boldsymbol{x}; \mathcal{W}') \right] \mathrm{vec}(W_l') \tag{14}$$

**Corollary 1** (SimCLR). *For SimCLR with contrastive losses $L_{\mathrm{simp}}$, $L_{\mathrm{tri}}$ and $L_{\mathrm{nce}}$, $\delta W_l^{\mathrm{BN}} = 0$.*

### 5.1 THE CASE WITHOUT EMA ($\gamma_{\mathrm{ema}} = 0$ AND THUS $\mathcal{W}_{\mathrm{base}} = \mathcal{W}'_{\mathrm{base}}$)

In BYOL when the predictor is present, $\mathcal{W}' \neq \mathcal{W}$ and BN is present, from the analysis above we know that $\delta W_l^{\mathrm{BN}} \neq 0$, which provides an implicit contrastive term. Note that $\mathcal{W}' \neq \mathcal{W}$ means there is a predictor, the online network uses EMA, or both. We first discuss the case without EMA.

From Theorem 8, if we further consider a single linear predictor, then the following holds. Here $\bar{K}_{l,\mathrm{base}}(\boldsymbol{x}) := \bar{K}_l(\boldsymbol{x}; \mathcal{W}_{\mathrm{base}})$ and zero-mean expected connection $\hat{K}_l(\boldsymbol{x}) := \bar{K}_l(\boldsymbol{x}) - \mathbb{E}_{\boldsymbol{x}} \left[ \bar{K}_l(\boldsymbol{x}) \right]$.

**Corollary 2** (The role of a predictor in BYOL). *If $\mathcal{W}_{\mathrm{pred}} = \{W_{\mathrm{pred}}\}$ is linear and no EMA, then $\mathrm{vec}(\widetilde{\Delta W_l}) = \mathrm{vec}(\Delta W_l)_{\mathrm{sym}} + \mathbb{E}_{\boldsymbol{x}} \left[ \hat{K}_{l,\mathrm{base}}(\boldsymbol{x}) W_{\mathrm{pred}}^\mathsf{T} (I - W_{\mathrm{pred}}) \hat{K}_{l,\mathrm{base}}^\mathsf{T}(\boldsymbol{x}) \right] \mathrm{vec}(W_l)$. If there is no stop gradient, then $\mathrm{vec}(\widetilde{\Delta W_l}) = 2\mathrm{vec}(\Delta W_l)_{\mathrm{sym}} - \mathbb{V}_{\boldsymbol{x}} \left[ \bar{K}_{l,\mathrm{base}}(\boldsymbol{x})(I - W_{\mathrm{pred}})^\mathsf{T} \right] \mathrm{vec}(W_l)$.*

*The Predictor.* To see why $\mathcal{W}_{\mathrm{pred}}$ plays a critical role, we check some special case. If $W_{\mathrm{pred}} = \beta I_{n_L \times n_L}$ ($W_{\mathrm{pred}}$ has to be a squared matrix), then $\mathrm{vec}(\Delta W_l) = \mathrm{vec}(\Delta W_l)_{\mathrm{sym}} + \beta(1 - \beta)\mathbb{V}_{\boldsymbol{x}} \left[ \bar{K}_{l,\mathrm{base}}(\boldsymbol{x}) \right] \mathrm{vec}(W_l)$. If $0 < \beta < 1$, then $\beta(1 - \beta) > 0$ and the covariance operator appears. In this regime, BYOL works like SimCLR, *except that* it also minimizes variance across different augmented views of the same data sample through $\mathrm{vec}(\Delta W_l)_{\mathrm{sym}}$ (Eqn. 11), the first term in Eqn. 14.

Indeed, the recent blogpost (Fetterman and Albrecht, 2020) as well as our own experiments (Tbl. 3) suggests that standard BYOL without BN fails. In addition, we also initialize the predictor with small positive weights (See Appendix G.4), and reinitialize the predictor weights once in a while (Tbl. 5), and BYOL still works, consistent with our theoretical prediction.

*Stop Gradient.* In BYOL, the target network $\mathcal{W}'$ serves as a target to be learned from, but does not contribute gradients to the current weight $\mathcal{W}$. Without EMA, we might wonder whether the target network should also contribute the gradient or not. Corollary 2 shows that this won't work: no matter what $W_{\mathrm{pred}}$ is, the update always contains a (weighted) negative covariance operator.

### 5.2 DYNAMICS OF EXPONENTIAL MOVING AVERAGE (EMA)

On the other hand, the EMA part might play a different role. Consider the following linear dynamic system, which is a simplified version of Eqn. 14 (we omit $\mathrm{vec}(\Delta W_l)_{\mathrm{sym}}$ and $W_l' = W_{l,\mathrm{ema}}$):

$$\boldsymbol{w}(t+1) - \boldsymbol{w}(t) = \Delta \boldsymbol{w}(t) = \alpha \left[ -\boldsymbol{w}(t) + (1 - \lambda)\boldsymbol{w}_{\mathrm{ema}}(t) \right] \tag{15}$$

**Theorem 9** (EMA dynamics in Eqn. 15). *$\boldsymbol{w}(t) \propto (1 + \kappa)^t$. Here we define $\kappa := \frac{1}{2}(\eta + \alpha) \left( \sqrt{1 + 4\alpha\eta\lambda/(\eta + \alpha)^2} - 1 \right)$ and $\eta := 1 - \gamma_{\mathrm{ema}}$. Moreover, if $\lambda \geq 0$, then $\kappa \leq \lambda/(1/\alpha + 1/\eta)$.*

From the analysis above, when $\beta$ is small, we see that the coefficient before $W_l$ ($\sim \beta^2$) is typically smaller than that before $W_{l,\mathrm{ema}}$ ($\sim \beta$). This means $\lambda > 0$. In this case, $\kappa > 0$ and $\boldsymbol{w}(t)$ grows exponentially and learning happens. Compared to no EMA case ($\gamma_{\mathrm{ema}} = 0$ or $\eta = 1$), with EMA, we have $\eta < 1$ and $\kappa$ becomes smaller. Then the growth is less aggressive and training stabilizes.

---

[1]In PyTorch, the former is `x-x.mean()` and the latter is `x-x.mean().detach()`.

[2]A formal treatment requires Jacobian $J$ to incorporate BatchNorm's contribution and is left for future work.

Table 2: Normalized Correlation between the topmost latent variables in binary HLTM and topmost nodes in deep ReLU networks ($L = 5$) go up when training with SimCLR with NCE loss. We see higher correlations at both initialization and end of training, with more over-parameterization (**Left**: $|\mathcal{N}_\mu| = 2$, **Right**: $|\mathcal{N}_\mu| = 5$).

| $\rho_{\mu\nu}$ | Initial | 1 epoch | 20 epochs | $\rho_{\mu\nu}$ | Initial | 1 epoch | 20 epochs |
|---|---|---|---|---|---|---|---|
| $\sim \text{Uniform}[0.7, 1]$ | 0.51 | 0.69 | 0.76 | $\sim \text{Uniform}[0.7, 1]$ | 0.60 | 0.72 | 0.88 |
| $\sim \text{Uniform}[0.8, 1]$ | 0.65 | 0.76 | 0.79 | $\sim \text{Uniform}[0.8, 1]$ | 0.73 | 0.80 | 0.87 |
| $\sim \text{Uniform}[0.9, 1]$ | 0.81 | 0.85 | 0.86 | $\sim \text{Uniform}[0.9, 1]$ | 0.87 | 0.90 | 0.95 |

Table 3: Top-1 STL performance with different combination of predictor (**P**), EMA and BatchNorm using BYOL. EMA means $\gamma_{\text{ema}} = 0.996$. Batch size is 128 and all experiments run on 5 seeds and 100 epochs.

| - | EMA | BN | EMA, BN | P | P, EMA | P, BN | P, EMA, BN |
|---|---|---|---|---|---|---|---|
| $38.7 \pm 0.6$ | $39.3 \pm 0.9$ | $33.0 \pm 0.3$ | $32.8 \pm 0.5$ | $39.5 \pm 3.1$ | $44.4 \pm 3.2$ | $63.6 \pm 1.06$ | $\mathbf{78.1 \pm 0.3}$ |

Table 4: Top-1 STL performance using different BatchNorm components in the predictor and the projector of BYOL ($\gamma_{\text{ema}} = 0.996$, 100 epochs). There is no affine part. "$\mu$" = zero-mean normalization only, "$\mu, \sigma$" = BN without affine, "$\mu, \sigma^\nmid$" = normalization with mean and std but only backpropagating through mean. All variants with detached zero-mean normalization (in red) yield similar poor performance as no normalization.

| - | $\mu$ | $\sigma$ | $\mu, \sigma$ | $\mu^\nmid$ | $\sigma^\nmid$ | $\mu^\nmid, \sigma$ | $\mu, \sigma^\nmid$ | $\mu^\nmid, \sigma^\nmid$ |
|---|---|---|---|---|---|---|---|---|
| $43.9 \pm 4.2$ | $64.8 \pm 0.6$ | $72.2 \pm 0.9$ | $\mathbf{78.1 \pm 0.3}$ | $\color{red}{44.2 \pm 7.0}$ | $54.2 \pm 0.6$ | $\color{red}{48.3 \pm 2.7}$ | $76.3 \pm 0.4$ | $\color{red}{47.0 \pm 8.1}$ |

Table 5: Top-1 performance of BYOL using reinitialization of the predictor every $T$ epochs.

| | Original BYOL | ReInit $T = 5$ | ReInit $T = 10$ | ReInit $T = 20$ |
|---|---|---|---|---|
| STL-10 (100 epochs) | 78.1 | 78.6 | **79.1** | 79.0 |
| ImageNet (60 epochs) | 60.9 | 61.9 | **62.4** | **62.4** |

## 6 EXPERIMENTS

We test our theoretical findings through experiments on STL-10 (Coates et al., 2011) and ImageNet (Deng et al., 2009). We use a simplified linear evaluation protocol: the linear classifier is trained on frozen representations computed *without* data augmentation. This reuses pre-computed representations and accelerates evaluation by 10x. See Sec. G.3 for detailed setup.

**Hierarchical Latent Tree Model (HLTM)**. We implement HLTM and check whether the intermediate layers of deep ReLU networks learn the corresponding latent variables at the same layer. The degree of learning is measured by the normalized correlations between the ground truth latent variable $z_\mu$ and its best corresponding node $j \in \mathcal{N}_\mu$. Tbl. 2 indicates this measure increases with over-parameterization and learning, consistent with our analysis (Sec. 4.3). More experiments in Sec. G.2.

**Factors underlying BYOL performance**. To test our theory, we perform an ablation study of BYOL on STL-10 by modifying three key components: predictor, EMA and BN. Tbl. 3 shows that BN and predictor are important and EMA further improves the performance. First, without a predictor, neither BN nor EMA give good performance. A predictor without BN still doesn't work. A predictor with BN starts to show good performance (63.6%) and further adding EMA leads to the best performance (78.1%). This is consistent with our theoretical findings in Sec. 5, in which we show that using a predictor with BN yields $\delta W_l^{\text{BN}} \neq 0$ and leads to an implicit contrastive term.

To further test our understanding of the role played by BN, we fractionate BN into several sub-components: subtract by batch mean (mean-norm), divide by batch standard deviation (std-norm) and affine, and do ablation studies (Tbl. 4). Surprisingly, removing affine yields slightly better performance on STL-10 (from 78.1% to 78.7%). We also find that variants of mean-norm performs reasonably, while variants of *detached* mean-norm has similar poor performance as no normalization, supporting that centralizing backpropagated gradient leads to implicit contrastive terms (Sec. 5). Note that std-norm also helps, which we leave for future analysis.

We also check whether the online network requires an "optimal predictor" as suggested by recent version (v3) of BYOL. For this, we reinitialize the predictor (ReInit) every $T$ epochs and compare the final performance under linear evaluation protocol. Interestingly, as shown in Tbl. 5, ReInit actually improves the performance a bit, compared to the original BYOL that keeps training the same predictor, which should therefore be closer to optimal. Moreover, if we shrink the initial weight range of the predictor to make $\text{Cov}_{\boldsymbol{x}}\left[\bar{K}_l(\boldsymbol{x}), \bar{K}_l(\boldsymbol{x}; \mathcal{W}')\right]$ (third term in Eqn. 14) more dominant, and reduce the learning rate, the performance further improves (See Tbl. 10 in Appendix G.4), thereby corroborating our analysis.

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

## A    BACKGROUND AND BASIC SETTING (SECTION 2)

### A.1    LEMMAS

**Definition 2** (reversibility). *A layer $l$ is* reversible *if there is a $G_l(\boldsymbol{x}; \mathcal{W}) \in \mathbb{R}^{n_l \times n_{l-1}}$ so that $\boldsymbol{f}_l(\boldsymbol{x}; \mathcal{W}) = G_l(\boldsymbol{x}; \mathcal{W})\boldsymbol{f}_{l-1}(\boldsymbol{x}; \mathcal{W})$ and $\boldsymbol{g}_{l-1} = G_l^\mathsf{T}(\boldsymbol{x}; \mathcal{W})Q_l\boldsymbol{g}_l$ for some constant PSD matrix $\mathbb{R}^{n_l \times n_l} \ni Q_l \succeq 0$. A network is* reversible *if all layers are.*

Note that many different kinds of layers have this reversible property, including linear layers (MLP and Conv) and (leaky) ReLU nonlinearity. For multi-layer ReLU network, for each layer $l$, we have:

$$G_l(\boldsymbol{x}; \mathcal{W}) = D_l(\boldsymbol{x}; \mathcal{W})W_l, \qquad Q_l \equiv I_{n_l \times n_l} \tag{16}$$

where $D_l \in \mathbb{R}^{n_l \times n_l}$ is a binary diagonal matrix that encodes the gating of each neuron at layer $l$. The gating $D_l(\boldsymbol{x}; \mathcal{W})$ depends on the current input $\boldsymbol{x}$ and current weight $\mathcal{W}$.

In addition to ReLU, other activation function also satisfies this condition, including linear, LeakyReLU and monomial activations. For example, for power activation $\psi(x) = x^p$ where $p > 1$, we have (where $\tilde{\boldsymbol{f}}_l$ is the pre-activation at layer $l$):

$$G_l(\boldsymbol{x}; \mathcal{W}) = \mathrm{diag}^{p-1}(\tilde{\boldsymbol{f}}_l)W_l, \qquad Q_l \equiv pI_{n_l \times n_l} \tag{17}$$

**Remark**. Note that the reversibility is not the same as invertible. Specifically, reversibility only requires the transfer function of a backpropagation gradient is a transpose of the forward function.

**Lemma 1** (Recursive Gradient Update (Extension to Lemma 1 in (Tian, 2020))). *Let (pseudo)-Jacobian matrix $\tilde{J}_L(\boldsymbol{x}) = I_{n_L \times n_L}$, and recursively define $\tilde{J}_{l-1}(\boldsymbol{x}) := \tilde{J}_l(\boldsymbol{x})\sqrt{Q_l}G_l(\boldsymbol{x}) \in \mathbb{R}^{n_l \times n_{l-1}}$. Here $\sqrt{Q_l}$ is the constant PSD matrix so that $\sqrt{Q_l}\sqrt{Q_l} = Q_l \succeq 0$.*

*If **(1)** the network is reversible (Def. 2) and **(2)** $\sqrt{Q_l}$ commutes with $\tilde{J}_l(\boldsymbol{x}_1)^\mathsf{T}\tilde{J}_l(\boldsymbol{x}_1)$ and $\tilde{J}_l(\boldsymbol{x}_1)^\mathsf{T}\tilde{J}_l(\boldsymbol{x}_2)$, then minimizing the $\ell_2$ objective*

$$r(\mathcal{W}_1) := \frac{1}{2}\|\boldsymbol{f}_L(\boldsymbol{x}_1; \mathcal{W}_1) - \boldsymbol{f}_L(\boldsymbol{x}_2; \mathcal{W}_2)\|_2^2 \tag{18}$$

*with respect to weight matrix $W_l$ at layer $l$ yields the following gradient at layer $l$:*

$$\boldsymbol{g}_l = \tilde{J}_l^\mathsf{T}(\boldsymbol{x}_1; \mathcal{W}_1) \left[ \tilde{J}_l(\boldsymbol{x}_1; \mathcal{W}_1)\boldsymbol{f}_l(\boldsymbol{x}_1; \mathcal{W}_1) - \tilde{J}_l(\boldsymbol{x}_2; \mathcal{W}_2)\boldsymbol{f}_l(\boldsymbol{x}_2; \mathcal{W}_2) \right] \tag{19}$$

*Proof.* We prove by induction. Note that our definition of $W_l$ is the transpose of $W_l$ defined in (Tian, 2020). Also our $\boldsymbol{g}_l(\boldsymbol{x})$ is the gradient *before* nonlinearity, while (Tian, 2020) uses the same symbol for the gradient after nonlinearity.

For notation brievity, we let $\boldsymbol{f}_l(\boldsymbol{x}_1) := \boldsymbol{f}_l(\boldsymbol{x}_1; \mathcal{W}_l)$ and $G_l(\boldsymbol{x}_1) := G_l(\boldsymbol{x}_1; \mathcal{W}_l)$. Similar for $\boldsymbol{x}_2$ and $W_2$.

When $l = L$, by the property of $\ell_2$-loss, we know that $\boldsymbol{g}_L = \boldsymbol{f}_L(\boldsymbol{x}_1; \mathcal{W}_1) - \boldsymbol{f}_L(\boldsymbol{x}_2; \mathcal{W}_2)$, by setting $\tilde{J}_L(\boldsymbol{x}_1) = \tilde{J}_L(\boldsymbol{x}_2) = I$, the condition holds. Now suppose for layer $l$, we have:

$$\boldsymbol{g}_l = \tilde{J}_l^\mathsf{T}(\boldsymbol{x}_1) \left[ \tilde{J}_l(\boldsymbol{x}_1)\boldsymbol{f}_l(\boldsymbol{x}_1) - \tilde{J}_l(\boldsymbol{x}_2)\boldsymbol{f}_l(\boldsymbol{x}_2) \right] \tag{20}$$

Then:

$$\boldsymbol{g}_{l-1} = G_l^\mathsf{T}(\boldsymbol{x}_1)Q_l\boldsymbol{g}_l \tag{21}$$

$$= G_l^\mathsf{T}(\boldsymbol{x}_1)Q_l\tilde{J}_l^\mathsf{T}(\boldsymbol{x}_1) \left[ \tilde{J}_l(\boldsymbol{x}_1)\boldsymbol{f}_l(\boldsymbol{x}_1) - \tilde{J}_l(\boldsymbol{x}_2)\boldsymbol{f}_l(\boldsymbol{x}_2) \right] \tag{22}$$

$$= \underbrace{G_l^\mathsf{T}(\boldsymbol{x}_1)\sqrt{Q_l}\tilde{J}_l^\mathsf{T}(\boldsymbol{x}_1)}_{\tilde{J}_{l-1}^\mathsf{T}(\boldsymbol{x}_1)} \cdot \left[ \tilde{J}_l(\boldsymbol{x}_1)\sqrt{Q_l}\boldsymbol{f}_l(\boldsymbol{x}_1) - \tilde{J}_l(\boldsymbol{x}_2)\sqrt{Q_l}\boldsymbol{f}_l(\boldsymbol{x}_2) \right] \tag{23}$$

$$= \tilde{J}_{l-1}^\mathsf{T}(\boldsymbol{x}_1) \left[ \underbrace{\tilde{J}_l(\boldsymbol{x}_1)\sqrt{Q_l}G_l(\boldsymbol{x}_1)}_{\tilde{J}_{l-1}(\boldsymbol{x}_1)} \boldsymbol{f}_{l-1}(\boldsymbol{x}_1) - \underbrace{\tilde{J}_l(\boldsymbol{x}_2)\sqrt{Q_l}G_l(\boldsymbol{x}_2)}_{\tilde{J}_{l-1}(\boldsymbol{x}_2)} \boldsymbol{f}_{l-1}(\boldsymbol{x}_2) \right] \tag{24}$$

$$= \tilde{J}_{l-1}^\mathsf{T}(\boldsymbol{x}_1) \left[ \tilde{J}_{l-1}(\boldsymbol{x}_1)\boldsymbol{f}_{l-1}(\boldsymbol{x}_1) - \tilde{J}_{l-1}(\boldsymbol{x}_2)\boldsymbol{f}_{l-1}(\boldsymbol{x}_2) \right] \tag{25}$$

Note that for multi-layered ReLU network, $G_l(\boldsymbol{x}) = D_l(\boldsymbol{x})W_l$, $Q_l = I$ for each ReLU+Linear layer, if we set $\boldsymbol{x}_1 = \boldsymbol{x}_2 = \boldsymbol{x}$, $\mathcal{W}_1 = \mathcal{W}$, $\mathcal{W}_2 = \mathcal{W}^*$ (teacher weights), then we go back to the original Lemma 1 in (Tian, 2020). □

**Remark on ResNet.** Note that the same structure holds for blocks of ResNet with ReLU activation.

**An alternative form of Lemma 1.** Note that we can alternatively group linear weight with its immediate downward nonlinearity and Lemma 1 still holds. In this case, we will have:

$$\tilde{\boldsymbol{g}}_l = J_l^\mathsf{T}(\boldsymbol{x}_1) \left[ J_l(\boldsymbol{x}_1)\tilde{\boldsymbol{f}}_l(\boldsymbol{x}_1) - J_l(\boldsymbol{x}_2)\tilde{\boldsymbol{f}}_l(\boldsymbol{x}_2) \right] \tag{26}$$

where $J_l(\boldsymbol{x})$ is the (pseudo-)Jacobian: $J_l(\boldsymbol{x}) := \partial \boldsymbol{f}_L / \partial \tilde{\boldsymbol{f}}_l$ (i.e., with respect to the pre-activation $\tilde{\boldsymbol{f}}_l$), as defined in the notation paragraph of Sec. 2, and $\tilde{\boldsymbol{g}}_l$ is the back-propagated gradient *after* the nonlinearity. This will be used in the following Lemma 2. For other reversible layers (e.g., Eqn. 17), the relationship between the pseudo-Jacobian and the real one can differ by a constant (e.g., some power of $\sqrt{p}$).

## A.2 $\ell_2$-NORMALIZATION IN THE TOPMOST LAYER

For $\ell_2$-normalized layer $\boldsymbol{f}_l := \boldsymbol{f}_{l-1}/\|\boldsymbol{f}_{l-1}\|_2$, we have $G_l := 1/\|\boldsymbol{f}_{l-1}\|_2 \cdot I_{n_l \times n_l}$ and due to the following identity (here $\tilde{\boldsymbol{y}} := \boldsymbol{y}/\|\boldsymbol{y}\|_2$):

$$\frac{\partial \tilde{\boldsymbol{y}}}{\partial \boldsymbol{y}} = \frac{1}{\|\boldsymbol{y}\|_2} \left( I - \tilde{\boldsymbol{y}}\tilde{\boldsymbol{y}}^\mathsf{T} \right) \tag{27}$$

Therefore we have $\partial \boldsymbol{f}_l / \partial \boldsymbol{f}_{l-1} = (I_{n_l \times n_l} - \boldsymbol{f}_l \boldsymbol{f}_l^\mathsf{T})G_l$ and we could set $Q_l := I - \boldsymbol{f}_l \boldsymbol{f}_l^\mathsf{T}$, which is a projection matrix and thus PSD. Furthermore, since the normalization layer is at the topmost, $\tilde{J}_l = I$ and $Q_l$ trivially commutes with $\tilde{J}_l^\mathsf{T} \tilde{J}_l$.

The *only issue* is that $Q_l$ is not a constant matrix and can change over training. Therefore Lemma 1 doesn't apply exactly to such a layer but can be regarded as an approximate way to model.

## A.3 THEOREM 1

Now we prove Theorem 1 in a more general setting where the network is reversible (note that deep ReLU networks are included and its $Q_l$ is a simple identity matrix):

**Lemma 2** (Squared $\ell_2$ Gradient for dual deep reversible networks). *The gradient $g_{W_l}$ of the squared loss $r$ with respect to $W_l \in \mathbb{R}^{n_l \times n_{l-1}}$ for a single input pair $\{\boldsymbol{x}_1, \boldsymbol{x}_2\}$ is:*

$$g_{W_l} = \mathrm{vec}\left(\partial r / \partial W_{1,l}\right) = K_{1,l} \left[ K_{1,l}^\mathsf{T}\mathrm{vec}(W_{1,l}) - K_{2,l}^\mathsf{T}\mathrm{vec}(W_{2,l}) \right]. \tag{28}$$

*Here $K_l(\boldsymbol{x}; \mathcal{W}) := \boldsymbol{f}_{l-1}(\boldsymbol{x}; \mathcal{W}) \otimes J_l^\mathsf{T}(\boldsymbol{x}; \mathcal{W})$, $K_{1,l} := K_l(\boldsymbol{x}_1; \mathcal{W}_1)$ and $K_{2,l} := K_l(\boldsymbol{x}_2; \mathcal{W}_2)$.*

*Proof.* We consider more general case where the two towers have different parameters, namely $\mathcal{W}_1$ and $\mathcal{W}_2$. Applying Lemma 1 for the branch with input $\boldsymbol{x}_1$ at the linear layer $l$, and we have (See Eqn. 26):

$$\tilde{\boldsymbol{g}}_{1,l} = J_{1,l}^\mathsf{T}[J_{1,l}W_{1,l}\boldsymbol{f}_{1,l-1} - J_{2,l}W_{2,l}\boldsymbol{f}_{2,l-1}] \tag{29}$$

where $\boldsymbol{f}_{1,l-1} := \boldsymbol{f}_{l-1}(\boldsymbol{x}_1; \mathcal{W}_1)$ is the activation of layer $l-1$ just below the linear layer at tower 1 (similar for other symbols), and $\tilde{\boldsymbol{g}}_{1,l}$ is the back-propagated gradient *after* the nonlinearity.

In this case, the gradient (and the weight update, according to gradient descent) of the weight $W_l$ between layer $l$ and layer $l-1$ is:

$$\frac{\partial r}{\partial W_{1,l}} = \tilde{\boldsymbol{g}}_{1,l}\boldsymbol{f}_{1,l-1}^\mathsf{T} \tag{30}$$

$$= J_{1,l}^\mathsf{T}J_{1,l}W_{1,l}\boldsymbol{f}_{1,l-1}\boldsymbol{f}_{1,l-1}^\mathsf{T} - J_{1,l}^\mathsf{T}J_{2,l}W_{2,l}\boldsymbol{f}_{2,l-1}\boldsymbol{f}_{1,l-1}^\mathsf{T} \tag{31}$$

Using $\mathrm{vec}(AXB) = (B^\mathsf{T} \otimes A)\mathrm{vec}(X)$ (where $\otimes$ is the Kronecker product), we have:

$$\mathrm{vec}\left(\frac{\partial r}{\partial W_{1,l}}\right) = \left(\boldsymbol{f}_{1,l-1}\boldsymbol{f}_{1,l-1}^\mathsf{T} \otimes J_{1,l}^\mathsf{T}J_{1,l}\right)\mathrm{vec}(W_{1,l}) - \left(\boldsymbol{f}_{1,l-1}\boldsymbol{f}_{2,l-1}^\mathsf{T} \otimes J_{1,l}^\mathsf{T}J_{2,l}\right)\mathrm{vec}(W_{2,l}) \tag{32}$$

Let

$$K_l(\boldsymbol{x}; W) := \boldsymbol{f}_{l-1}(\boldsymbol{x}; \mathcal{W}) \otimes J_l^{\mathsf{T}}(\boldsymbol{x}; \mathcal{W}) \in \mathbb{R}^{n_l n_{l-1} \times n_L} \tag{33}$$

Note that $K_l(\boldsymbol{x}; \mathcal{W})$ is a function of the current weight $W$, which includes weights at all layers. By the mixed-product property of Kronecker product $(A \otimes B)(C \otimes D) = AC \otimes BD$, we have:

$$\mathrm{vec}\left(\frac{\partial r}{\partial W_{1,l}}\right) = K_l(\boldsymbol{x}_1) K_l(\boldsymbol{x}_1)^{\mathsf{T}} \mathrm{vec}(W_{1,l}) - K_l(\boldsymbol{x}_1) K_l(\boldsymbol{x}_2)^{\mathsf{T}} \mathrm{vec}(W_{2,l}) \tag{34}$$

$$= K_l(\boldsymbol{x}_1) \left[ K_l(\boldsymbol{x}_1)^{\mathsf{T}} \mathrm{vec}(W_{1,l}) - K_l(\boldsymbol{x}_2)^{\mathsf{T}} \mathrm{vec}(W_{2,l}) \right] \tag{35}$$

where $K_l(\boldsymbol{x}_1) = K_l(\boldsymbol{x}_1; \mathcal{W}_1)$ and $K_l(\boldsymbol{x}_2) = K_l(\boldsymbol{x}_2; \mathcal{W}_2)$.

In SimCLR case, we have $\mathcal{W}_1 = \mathcal{W}_2 = \mathcal{W}$ so

$$\mathrm{vec}\left(\frac{\partial r}{\partial W_l}\right) = K_l(\boldsymbol{x}_1) \left[ K_l(\boldsymbol{x}_1) - K_l(\boldsymbol{x}_2) \right]^{\mathsf{T}} \mathrm{vec}(W_l) \tag{36}$$

$\square$

# B   ANALYSIS OF SIMCLR USING TEACHER-STUDENT SETTING (SECTION 3)

## B.1   THEOREM 2

*Proof.* For $L_{\mathrm{simp}}$ and $L_{\mathrm{tri}}$ the derivation is obvious. For $L_{\mathrm{nce}}$, we have:

$$\frac{\partial L}{\partial r_+} = \frac{1}{\tau}\left(1 - \frac{e^{-r_+/\tau}}{e^{-r_+/\tau} + \sum_{k'=1}^{H} e^{-r_{k'-}/\tau}}\right) > 0 \tag{37}$$

$$\frac{\partial L}{\partial r_{k-}} = -\frac{1}{\tau}\left(\frac{e^{-r_{k-}/\tau}}{e^{-r_+/\tau} + \sum_{k'=1}^{H} e^{-r_{k'-}/\tau}}\right) < 0, \qquad k = 1, \ldots, H \tag{38}$$

and obviously we have:

$$\frac{\partial L}{\partial r_+} + \sum_{k=1}^{H} \frac{\partial L}{\partial r_{k-}} = 0 \tag{39}$$

$\square$

## B.2   THE COVARIANCE OPERATOR UNDER DIFFERENT LOSS FUNCTIONS

**Lemma 3.** *For a loss function $L$ that satisfies Theorem 2, with a batch of size one with samples $\mathcal{X} := \{\boldsymbol{x}_1, \boldsymbol{x}_+, \boldsymbol{x}_{1-}, \boldsymbol{x}_{2-}, \ldots, \boldsymbol{x}_{H-}\}$, where $\boldsymbol{x}_1, \boldsymbol{x}_+ \sim p_{\mathrm{aug}}(\cdot|\boldsymbol{x})$ are augmentation from the same sample $\boldsymbol{x}$, and $\boldsymbol{x}_{k-} \sim p_{\mathrm{aug}}(\cdot|\boldsymbol{x}_k')$ are augmentations from independent samples $\boldsymbol{x}_k' \sim p(\cdot)$. We have:*

$$\mathrm{vec}(g_{W_l}) = K_l(\boldsymbol{x}_1) \sum_{k=1}^{H} \left[ \left.\frac{\partial L}{\partial r_{k-}}\right|_{\mathcal{X}} \cdot (K_l(\boldsymbol{x}_+) - K_l(\boldsymbol{x}_{k-}))^{\mathsf{T}} \right] \mathrm{vec}(W_l) \tag{40}$$

*Proof.* First we have:

$$\mathrm{vec}(g_{W_l}) = \frac{\partial L}{\partial W_l} = \frac{\partial L}{\partial r_+}\frac{\partial r_+}{\partial W_l} + \sum_{k=1}^{H} \frac{\partial L}{\partial r_{k-}}\frac{\partial r_{k-}}{\partial W_l} \tag{41}$$

Then we compute each terms. Using Theorem 1, we know that:

$$\frac{\partial r_+}{\partial W_l} = K_l(\boldsymbol{x}_1)(K_l(\boldsymbol{x}_1) - K_l(\boldsymbol{x}_+))^{\mathsf{T}} \mathrm{vec}(W_l) \tag{42}$$

$$\frac{\partial r_{k-}}{\partial W_l} = K_l(\boldsymbol{x}_1)(K_l(\boldsymbol{x}_1) - K_l(\boldsymbol{x}_{k-}))^{\mathsf{T}} \mathrm{vec}(W_l), \quad k = 1, \ldots, n \tag{43}$$

Since Eqn. 39 holds, $K_l(\boldsymbol{x}_1)K_l^{\mathsf{T}}(\boldsymbol{x}_1)$ will be cancelled out and we have:

$$\mathrm{vec}(g_{W_l}) = K_l(\boldsymbol{x}_1) \sum_{k=1}^{H} \left[ \frac{\partial L}{\partial r_{k-}}(K_l(\boldsymbol{x}_+) - K_l(\boldsymbol{x}_{k-}))^{\mathsf{T}} \right] \mathrm{vec}(W_l) \tag{44}$$

$\square$

## B.3 THEOREM 3

*Proof.* For $L_{\text{simp}} := r_+ - r_-$, we have $H = 1$ and $\frac{\partial L}{\partial r_-} \equiv -1$. Therefore using Lemma 3, we have:

$$\text{vec}(g_{W_l}) = -K_l(\boldsymbol{x}_1)\left[K_l^{\mathsf{T}}(\boldsymbol{x}_+) - K_l^{\mathsf{T}}(\boldsymbol{x}_-)\right]\text{vec}(W_l) \tag{45}$$

Taking large batch limits, we know that $\mathbb{E}\left[K_l(\boldsymbol{x}_1)K_l^{\mathsf{T}}(\boldsymbol{x}_+)\right] = \mathbb{E}_{\boldsymbol{x}}\left[\bar{K}_l(\boldsymbol{x})\bar{K}_l^{\mathsf{T}}(\boldsymbol{x})\right]$ since $\boldsymbol{x}_1, \boldsymbol{x}_+ \sim p_{\text{aug}}(\cdot|\boldsymbol{x})$ are all augmented data points from a common sample $\boldsymbol{x}$. On the other hand, $\mathbb{E}\left[K_l(\boldsymbol{x}_1)K_l^{\mathsf{T}}(\boldsymbol{x}_{k-})\right] = \mathbb{E}_{\boldsymbol{x}}\left[\bar{K}_l(\boldsymbol{x})\right]\mathbb{E}_{\boldsymbol{x}}\left[\bar{K}_l^{\mathsf{T}}(\boldsymbol{x})\right]$ since $\boldsymbol{x}_1 \sim p_{\text{aug}}(\cdot|\boldsymbol{x})$ and $\boldsymbol{x}_{k-} \sim p_{\text{aug}}(\cdot|\boldsymbol{x}_k')$ are generated from independent samples $\boldsymbol{x}$ and $\boldsymbol{x}_k'$ and independent data augmentation. Therefore,

$$\text{vec}(g_{W_l}) = -\left\{\mathbb{E}_{\boldsymbol{x}}\left[\bar{K}_l(\boldsymbol{x})\bar{K}_l^{\mathsf{T}}(\boldsymbol{x})\right] - \mathbb{E}_{\boldsymbol{x}}\left[\bar{K}_l(\boldsymbol{x})\right]\mathbb{E}_{\boldsymbol{x}}\left[\bar{K}_l^{\mathsf{T}}(\boldsymbol{x})\right]\right\}\text{vec}(W_l) \tag{46}$$

$$= -\mathbb{V}_{\boldsymbol{x}}\left[\bar{K}_l(\boldsymbol{x})\right]\text{vec}(W_l) \tag{47}$$

The conclusion follows since gradient descent is used and $\Delta W_l = -g_{W_l}$. □

## B.4 THEOREM 4

*Proof.* When $\partial L/\partial r_{k-}$ is no longer constant, we consider its expansion with respect to un-augmented data point $\mathcal{X}_0 = \{\boldsymbol{x}, \boldsymbol{x}_1', \ldots, \boldsymbol{x}_k'\}$. Here $\mathcal{X} = \{\boldsymbol{x}_1, \boldsymbol{x}_+, \boldsymbol{x}_{1-}, \ldots, \boldsymbol{x}_{H-}\}$ is one data sample that includes both positive and negative pairs. Note that $\boldsymbol{x}_1, \boldsymbol{x}_+ \sim p_{\text{aug}}(\cdot|\boldsymbol{x})$ and $\boldsymbol{x}_{k-} \sim p_{\text{aug}}(\cdot|\boldsymbol{x}_k')$ for $1 \le k \le H$.

$$\left.\frac{\partial L}{\partial r_{k-}}\right|_{\mathcal{X}} = \left.\frac{\partial L}{\partial r_{k-}}\right|_{\mathcal{X}_0} + \epsilon \tag{48}$$

where $\epsilon$ is a bounded quantity for $L_{\text{tri}}$ ($|\epsilon| \le 1$) and $L_{\text{nce}}$ ($|\epsilon| \le 2/\tau$).

We consider $H = 1$ where there is only a single negative pair (and $r_-$). In this case $\mathcal{X}_0 = \{\boldsymbol{x}, \boldsymbol{x}'\}$. Let $r := \frac{1}{2}\|\boldsymbol{f}_L(\boldsymbol{x}) - \boldsymbol{f}_L(\boldsymbol{x}')\|_2^2$ and $\xi(\boldsymbol{x}, \boldsymbol{x}') := -\left.\frac{\partial L}{\partial r_{k-}}\right|_{\mathcal{X}_0}$.

Note that for $L_{\text{tri}}$, it is not differentiable, so we could use its soft version: $L_{\text{tri}}^{\tau}(r_+, r_-) = \tau \log(1 + e^{(r_+ - r_- + r_0)/\tau})$. It is easy to see that $\lim_{\tau \to 0} L_{\text{tri}}^{\tau}(r_+, r_-) \to \max(r_+ - r_- + r_0, 0)$.

For the two losses:

- For $L_{\text{tri}}^{\tau}$, we have

$$\xi(\boldsymbol{x}, \boldsymbol{x}') = \xi(r) = \frac{e^{-r/\tau}}{e^{-r_0/\tau} + e^{-r/\tau}}. \tag{49}$$

- For $L_{\text{nce}}$, we have

$$\xi(\boldsymbol{x}, \boldsymbol{x}') = \xi(r) = \frac{1}{\tau}\frac{e^{-r/\tau}}{1 + e^{-r/\tau}}. \tag{50}$$

Note that for $L_{\text{tri}}^{\tau}$ we have $\lim_{\tau \to 0} \xi(r) = \mathbb{I}(r \le r_0)$. for $L_{\text{nce}}$, since it is differentiable, by Taylor expansion we have $\epsilon = O(\|\boldsymbol{x}_1 - \boldsymbol{x}\|_2, \|\boldsymbol{x}_+ - \boldsymbol{x}\|_2, \|\boldsymbol{x}_- - \boldsymbol{x}'\|_2)$, which will be used later.

**The constant term $\xi$ with respect to data augmentation.** In the following, we first consider the term $\xi$, which only depends on un-augmented data points $\mathcal{X}_0$. From Lemma 3, we now have a term in the gradient:

$$\boldsymbol{g}_l(\mathcal{X}) := -K_l(\boldsymbol{x}_1)\left[K_l^{\mathsf{T}}(\boldsymbol{x}_+) - K_l^{\mathsf{T}}(\boldsymbol{x}_-)\right]\xi(\boldsymbol{x}, \boldsymbol{x}')\text{vec}(W_l) \tag{51}$$

Under the large batch limit, taking expectation with respect to data augmentation $p_{\text{aug}}$ and notice that all augmentations are done independently, given un-augmented data $\boldsymbol{x}$ and $\boldsymbol{x}'$, we have:

$$\boldsymbol{g}_l(\boldsymbol{x}, \boldsymbol{x}') := \mathbb{E}_{p_{\text{aug}}}[\boldsymbol{g}_l(\mathcal{X})] = -\bar{K}_l(\boldsymbol{x})\left[\bar{K}_l^{\mathsf{T}}(\boldsymbol{x}) - \bar{K}_l^{\mathsf{T}}(\boldsymbol{x}')\right]\xi(\boldsymbol{x}, \boldsymbol{x}')\text{vec}(W_l) \tag{52}$$

Symmetrically, if we swap $\boldsymbol{x}$ and $\boldsymbol{x}'$ since both are sampled from the same distribution $p(\cdot)$, we have:

$$\boldsymbol{g}_l(\boldsymbol{x}', \boldsymbol{x}) = -\bar{K}_l(\boldsymbol{x}')\left[\bar{K}_l^{\mathsf{T}}(\boldsymbol{x}') - \bar{K}_l^{\mathsf{T}}(\boldsymbol{x})\right]\xi(\boldsymbol{x}', \boldsymbol{x})\text{vec}(W_l) \tag{53}$$

since $\xi(\boldsymbol{x}', \boldsymbol{x})$ only depends on the squared $\ell_2$ distance $r$ (Eqn. 49 and Eqn. 50), we have $\xi(\boldsymbol{x}', \boldsymbol{x}) = \xi(\boldsymbol{x}, \boldsymbol{x}') = \xi(r)$ and thus:

$$
\begin{aligned}
\boldsymbol{g}_l(\boldsymbol{x}, \boldsymbol{x}') + \boldsymbol{g}_l(\boldsymbol{x}', \boldsymbol{x}) &= -\left[\bar{K}_l(\boldsymbol{x})\bar{K}_l^\mathsf{T}(\boldsymbol{x}) - \bar{K}_l(\boldsymbol{x})\bar{K}_l^\mathsf{T}(\boldsymbol{x}') + \bar{K}_l(\boldsymbol{x}')\bar{K}_l^\mathsf{T}(\boldsymbol{x}') - \bar{K}_l(\boldsymbol{x}')\bar{K}_l^\mathsf{T}(\boldsymbol{x})\right]\xi(r)\mathrm{vec}(W_l) \\
&= -\xi(r)(\bar{K}_l(\boldsymbol{x}) - \bar{K}_l(\boldsymbol{x}'))(\bar{K}_l(\boldsymbol{x}) - \bar{K}_l(\boldsymbol{x}'))^\mathsf{T}\mathrm{vec}(W_l)
\end{aligned} \tag{54}
$$

Therefore, we have:

$$
\begin{aligned}
\mathbb{E}_{\boldsymbol{x},\boldsymbol{x}'\sim p}\left[\boldsymbol{g}_l(\boldsymbol{x}, \boldsymbol{x}')\right] &= -\frac{1}{2}\mathbb{E}_{\boldsymbol{x},\boldsymbol{x}'\sim p}\left[\xi(r)(\bar{K}_l(\boldsymbol{x}) - \bar{K}_l(\boldsymbol{x}'))(\bar{K}_l(\boldsymbol{x}) - \bar{K}_l(\boldsymbol{x}'))^\mathsf{T}\right]\mathrm{vec}(W_l) \tag{55} \\
&= -\frac{1}{2}\mathbb{V}_{\boldsymbol{x},\boldsymbol{x}'\sim p}^{\xi}\left[\bar{K}_l(\boldsymbol{x}) - \bar{K}_l(\boldsymbol{x}')\right]\mathrm{vec}(W_l) \tag{56}
\end{aligned}
$$

**Bound the error**. For $L_{\mathrm{nce}}$, let $F := -\frac{\partial L}{\partial r_-}$, then we can compute their partial derivatives:

$$
\frac{\partial F}{\partial r_+} = -F(1/\tau - F), \qquad \frac{\partial F}{\partial r_-} = F(1/\tau - F) \tag{57}
$$

Note that $|F(1/\tau - F)| \le 1/\tau^2$ is always bounded. From Taylor expansion, we have:

$$
\epsilon = -\frac{\partial F}{\partial r_+}\bigg|_{\{\tilde{r}_+, \tilde{r}_-\}}(r_+ - r_+^0) - \frac{\partial F}{\partial r_-}\bigg|_{\{\tilde{r}_+, \tilde{r}_-\}}(r_- - r_-^0) \tag{58}
$$

for derivatives evaluated at some point $\{\tilde{r}_+, \tilde{r}_-\}$ at the line connecting $(\boldsymbol{x}, \boldsymbol{x}, \boldsymbol{x}')$ and $(\boldsymbol{x}_1, \boldsymbol{x}_+, \boldsymbol{x}_-)$. $r_+^0$ and $r_-^0$ are squared $\ell^2$ distances evaluated at $(\boldsymbol{x}, \boldsymbol{x}, \boldsymbol{x}')$, therefore, $r_+^0 \equiv 0$ and $r_-^0 = \frac{1}{2}\|\boldsymbol{f}(\boldsymbol{x}) - \boldsymbol{f}(\boldsymbol{x}')\|_2^2$ (note that here we just use $\boldsymbol{f} := \boldsymbol{f}_L$ for brevity).

Therefore, we have $r_+ - r_+^0 = \frac{1}{2}\|\boldsymbol{f}(\boldsymbol{x}_1) - \boldsymbol{f}(\boldsymbol{x}_+)\|_2^2$ and

$$
\begin{aligned}
r_- - r_-^0 &= [\boldsymbol{f}(\boldsymbol{x}) - \boldsymbol{f}(\boldsymbol{x}')]^\mathsf{T}[(\boldsymbol{f}(\boldsymbol{x}_1) - \boldsymbol{f}(\boldsymbol{x})) - (\boldsymbol{f}(\boldsymbol{x}_-) - \boldsymbol{f}(\boldsymbol{x}'))] \\
&+ \frac{1}{2}\|(\boldsymbol{f}(\boldsymbol{x}_1) - \boldsymbol{f}(\boldsymbol{x})) - (\boldsymbol{f}(\boldsymbol{x}_-) - \boldsymbol{f}(\boldsymbol{x}'))\|^2
\end{aligned} \tag{59}
$$

Therefore, we have:

$$
\begin{aligned}
\mathbb{E}_{p_{\mathrm{aug}}}\left[\left|\frac{\partial F}{\partial r_+}(r_+ - r_+^0)\right|\right] &\le \frac{1}{\tau^2}\cdot\frac{1}{2}\int\|\boldsymbol{f}(\boldsymbol{x}_1) - \boldsymbol{f}(\boldsymbol{x}_+)\|_2^2 p_{\mathrm{aug}}(\boldsymbol{x}_1|\boldsymbol{x})p_{\mathrm{aug}}(\boldsymbol{x}_+|\boldsymbol{x})\mathrm{d}\boldsymbol{x}_1\mathrm{d}\boldsymbol{x}_+ \\
&= \frac{1}{\tau^2}\mathrm{tr}\mathbb{V}_{\mathrm{aug}}[\boldsymbol{f}|\boldsymbol{x}] \tag{60}
\end{aligned}
$$

where $\mathrm{tr}\mathbb{V}_{\mathrm{aug}}[\boldsymbol{f}|\boldsymbol{x}] := \mathrm{tr}\mathbb{V}_{\boldsymbol{x}'\sim p_{\mathrm{aug}}(\cdot|\boldsymbol{x})}[\boldsymbol{f}(\boldsymbol{x}')]$ is a scalar. Similarly, using Lemma 4, we have the following (using $\|\boldsymbol{a}\|_2^2 + \|\boldsymbol{b}\|_2^2 \ge \frac{1}{2}\|\boldsymbol{a} - \boldsymbol{b}\|_2^2$). Here $c_0 := \max_{\boldsymbol{x}}\|\boldsymbol{f}(\boldsymbol{x}) - \mathbb{E}_{\boldsymbol{x}'\sim p_{\mathrm{aug}}(\cdot|\boldsymbol{x})}[\boldsymbol{f}(\boldsymbol{x}')]\|_2^2$:

$$
\begin{aligned}
&\mathbb{E}_{p_{\mathrm{aug}}}\left[\left|\frac{\partial F}{\partial r_-}(r_- - r_-^0)\right|\right] \tag{61} \\
&\le \frac{1}{\tau^2}\left\{\|\boldsymbol{f}(\boldsymbol{x}) - \boldsymbol{f}(\boldsymbol{x}')\|\left(\sqrt{\mathrm{tr}\mathbb{V}_{\mathrm{aug}}[\boldsymbol{f}|\boldsymbol{x}]} + \sqrt{\mathrm{tr}\mathbb{V}_{\mathrm{aug}}[\boldsymbol{f}|\boldsymbol{x}']} + 2c_0\right) + \mathrm{tr}\mathbb{V}_{\mathrm{aug}}[\boldsymbol{f}|\boldsymbol{x}] + \mathrm{tr}\mathbb{V}_{\mathrm{aug}}[\boldsymbol{f}|\boldsymbol{x}']\right\}
\end{aligned}
$$

Let $M_K := \max_{\boldsymbol{x}}\|K_l(\boldsymbol{x})\|$ so finally we have:

$$
\begin{aligned}
&|\mathbb{E}_{\boldsymbol{x},\boldsymbol{x}',\mathrm{aug}}[\epsilon K_l(\boldsymbol{x}_1)(K_l^\mathsf{T}(\boldsymbol{x}_+) - K_l^\mathsf{T}(\boldsymbol{x}_-))]| \\
&\le \mathbb{E}_{\boldsymbol{x},\boldsymbol{x}',\mathrm{aug}}[|\epsilon K_l(\boldsymbol{x}_1)(K_l^\mathsf{T}(\boldsymbol{x}_+) - K_l^\mathsf{T}(\boldsymbol{x}_-))|] \\
&\le \frac{2M_K^2}{\tau^2}\left\{2\mathbb{E}_{\boldsymbol{x},\boldsymbol{x}'\sim p(\cdot)}\left[\|\boldsymbol{f}(\boldsymbol{x}) - \boldsymbol{f}(\boldsymbol{x}')\|\left(\sqrt{\mathrm{tr}\mathbb{V}_{\mathrm{aug}}[\boldsymbol{f}|\boldsymbol{x}]} + c_0\right)\right] + 3\mathrm{tr}\mathbb{E}_{\boldsymbol{x}}[\mathbb{V}_{\mathrm{aug}}[\boldsymbol{f}|\boldsymbol{x}]]\right\} \tag{62}
\end{aligned}
$$

Note that if there is no augmentation (i.e., $p_{\mathrm{aug}}(\boldsymbol{x}_1|\boldsymbol{x}) = \delta(\boldsymbol{x}_1 - \boldsymbol{x})$), then $c_0 = 0$, $\mathbb{V}_{\mathrm{aug}}[\boldsymbol{f}|\boldsymbol{x}] \equiv 0$ and the error (Eqn. 62) is also zero. A small range of augmentation yields tight bound.

For $L_{\mathrm{tri}}^\tau$, the derivation is similar. The only difference is that we have $1/\tau$ rather than $1/\tau^2$ in Eqn. 62. Note that this didn't change the order of the bound since $\xi(r)$ (and thus the covariance operator) has one less $1/\tau$ as well. We could also see that for hard loss $L_{\mathrm{tri}}$, since $\tau \to 0$ this bound will be very loose. We leave a more tight bound as the future work. $\qquad\square$

**Remarks for $H > 1$.** Note that for $H > 1$, $L_{\mathrm{nce}}$ has multiple negative pairs and $\partial L / \partial r_{k-} = e^{-r_{k-}/\tau}/Z(\mathcal{X})$ where $Z(\mathcal{X}) := e^{-r_+/\tau} + \sum_{k=1}^{H} e^{-r_{k-}/\tau}$. While the nominator $e^{-r_{k-}/\tau}$ still only depends on the distance between $\boldsymbol{x}_1$ and $\boldsymbol{x}_{k-}$ (which is good), the normalization constant $Z(\mathcal{X})$ depends on $H + 1$ distance pairs simultaneously. This leads to

$$\xi_k = \frac{\partial L}{\partial r_{k-}}\Big|_{\mathcal{X}_0} = \frac{e^{-\|\boldsymbol{x}-\boldsymbol{x}_k'\|_2^2/\tau}}{1 + \sum_{k=1}^{H} e^{-\|\boldsymbol{x}-\boldsymbol{x}_k'\|_2^2/\tau}} \tag{63}$$

which causes issues with the symmetry trick (Eqn. 54), because the denominator involves many negative pairs at the same time.

However, if we think given one pair of distinct data point $(\boldsymbol{x}, \boldsymbol{x}')$, the normalized constant $Z$ averaged over data augmentation is approximately constant due to homogeneity of the dataset and data augmentation, then Eqn. 54 can still be applied and similar conclusion follows.

## C  THE DYNAMICS OF TWO-LAYER RELU NETWORK AND THE INTERPLAYS OF COVARIANCE OPERATORS BETWEEN NEARBY LAYERS (SECTION 4.2)

### C.1  THEOREM 5

*Proof.* For convenience, we define the centralized version of $\boldsymbol{u}_j(z_0)$: $\hat{\boldsymbol{u}}_j(z_0) = \boldsymbol{u}_j(z_0) - \mathbb{E}_{z_0}[\boldsymbol{u}_j(z_0)] \in \mathbb{R}^d$ and the matrices $A_{jk} := \mathrm{Cov}_{z_0}[\boldsymbol{u}_j(z_0), \boldsymbol{u}_k(z_0)] = \mathbb{E}_{z_0}[\hat{\boldsymbol{u}}_j(z_0)\hat{\boldsymbol{u}}_k^\mathsf{T}(z_0)] \in \mathbb{R}^{d \times d}$. Here both $j$ and $k$ run from 1 to $n_1$.

At layer $l = 1$ the covariance operator is $\mathbb{V}_{z_0}[\bar{K}_1(z_0)] = [\boldsymbol{w}_{2,j}^\mathsf{T} \boldsymbol{w}_{2,k} A_{jk}] \in \mathbb{R}^{n_1 d \times n_1 d}$.

On the other hand, if we check the second layer $l = 2$, we could compute $\bar{K}_2(z_0) \in \mathbb{R}^{n_1 n_2 \times n_2}$. Note that for input $j$ of the second layer, we can compute its expectation with respect to $z|z_0$ as $\boldsymbol{w}_{1,j}^\mathsf{T} \boldsymbol{u}_j(z_0)$. On the other hand, since the last layer doesn't have ReLU nonlinearity, the Jacobian $J_2 = I_{n_2 \times n_2}$ which is independent of the input. So we have:

$$\bar{K}_2(z_0) = \begin{bmatrix} \boldsymbol{w}_{1,1}^\mathsf{T} \boldsymbol{u}_1(z_0) \\ \boldsymbol{w}_{1,2}^\mathsf{T} \boldsymbol{u}_2(z_0) \\ \cdots \\ \boldsymbol{w}_{1,n_1}^\mathsf{T} \boldsymbol{u}_{n_1}(z_0) \end{bmatrix} \otimes I_{n_2 \times n_2} \in \mathbb{R}^{n_1 n_2 \times n_2} \tag{64}$$

So at layer $l = 2$ we can compute the covariance operator $\mathbb{V}_{z_0}[\bar{K}_2(z_0)] = [\boldsymbol{w}_{1,j}^\mathsf{T} A_{jk} \boldsymbol{w}_{1,k}] \otimes I_{n_2 \times n_2} \in \mathbb{R}^{n_1 n_2 \times n_1 n_2}$. Here $[\boldsymbol{w}_{1,j}^\mathsf{T} A_{jk} \boldsymbol{w}_{1,k}] \in \mathbb{R}^{n_1 \times n_1}$.

Using these two covariance operators, we are able to write down the weight update in SimCLR setting with simple contrastive loss (here $Q_j := \sum_k A_{jk} \boldsymbol{w}_{1,k} \boldsymbol{w}_{2,k}^\mathsf{T} \in \mathbb{R}^{d \times n_2}$):

$$\dot{\boldsymbol{w}}_{1,j} = Q_j \boldsymbol{w}_{2,j}, \quad \dot{\boldsymbol{w}}_{2,j} = Q_j^\mathsf{T} \boldsymbol{w}_{1,j} \tag{65}$$

The dynamics of Eqn. 65 can be quite general and hard to solve. In the following, we talk about some special cases.

**Diagonal $W_2$.** We consider the case where $W_2$ is a diagonal and square matrix, so $n_1 = n_2$ and $W_2 = \mathrm{diag}(w_{2,1}, w_{2,2}, \ldots, w_{2,n_1})$ and remains such a structure throughout the training. Note that this also means there is no bias term for all output nodes.

In this case, we could simplify Eqn. 65 due to the fact that now $\boldsymbol{w}_{2,k}^\mathsf{T}(t)\boldsymbol{w}_{2,j}(t) = 0$ for $j \neq k$ at any time step $t$ (again all biases are zero in the top-layer, otherwise the orthogonal condition do not hold):

$$\dot{\boldsymbol{w}}_{1,j} = w_{2,j}^2 A_{jj} \boldsymbol{w}_{1,j} \tag{66}$$

$$\dot{w}_{2,j} = (\boldsymbol{w}_{1,j}^\mathsf{T} A_{jj} \boldsymbol{w}_{1,j}) w_{2,j} \tag{67}$$

Note that if we multiply $\boldsymbol{w}_{1,j}$ to Eqn. 66 and multiply $w_{2,j}$ to Eqn. 67, we arrive at:

$$\frac{1}{2} \frac{\mathrm{d}\|\boldsymbol{w}_{1,j}\|_2^2}{\mathrm{d}t} = w_{2,j}^2 (\boldsymbol{w}_{1,j}^\mathsf{T} A_{jj} \boldsymbol{w}_{1,j}) \tag{68}$$

$$\frac{1}{2} \frac{\mathrm{d}w_{2,j}^2}{\mathrm{d}t} = (\boldsymbol{w}_{1,j}^\mathsf{T} A_{jj} \boldsymbol{w}_{1,j}) w_{2,j}^2 \tag{69}$$

Therefore, $\mathrm{d}\|\boldsymbol{w}_{1,j}\|_2^2/\mathrm{d}t = \mathrm{d}w_{2,j}^2/\mathrm{d}t$ and thus $\|\boldsymbol{w}_{1,j}\|_2^2 = w_{2,j}^2 + c$ with some time-independent constant $c$.

$\square$

## D    HIERARCHICAL LATENT TREE MODELS (SECTION 4.3)

### D.1   LEMMAS

**Lemma 4** (Variance Squashing). *Suppose a function $\phi : \mathbb{R} \mapsto \mathbb{R}$ is L-Lipschitz continuous: $|\phi(x) - \phi(y)| \leq L|x - y|$, then for $x \sim p(\cdot)$, we have:*

$$\mathbb{V}_p[\phi(x)] \leq L^2 \mathbb{V}_p[x] \tag{70}$$

*Proof.* Suppose $x, y \sim p(\cdot)$ are independent samples and $\mu_\phi := \mathbb{E}[\phi(x)]$. Note that $\mathbb{V}[\phi(x)]$ can be written as the following:

$$
\begin{aligned}
\mathbb{E}\left[|\phi(x) - \phi(y)|^2\right] &= \frac{1}{2}\mathbb{E}\left[|(\phi(x) - \mu_\phi) - (\phi(y) - \mu_\phi)|^2\right] \\
&= \mathbb{E}\left[|\phi(x) - \mu_\phi|^2\right] + \mathbb{E}\left[|\phi(y) - \mu_\phi|^2\right] - 2\mathbb{E}\left[(\phi(x) - \mu_\phi)(\phi(y) - \mu_\phi)\right] \\
&= 2\mathbb{V}_p[\phi(x)] \tag{71}
\end{aligned}
$$

Therefore we have:

$$\mathbb{V}_p[\phi(x)] = \frac{1}{2}\mathbb{E}\left[|\phi(x) - \phi(y)|^2\right] \leq \frac{L^2}{2}\mathbb{E}\left[|x - y|^2\right] = L^2 \mathbb{V}_p[x] \tag{72}$$

$\square$

**Lemma 5** (Sharpened Jensen's inequality (Liao and Berg, 2018)). *If function $\phi$ is twice differentiable, and $x \sim p(\cdot)$, then we have:*

$$\frac{1}{2}\mathbb{V}[x]\inf \phi'' \leq \mathbb{E}[\phi(x)] - \phi(\mathbb{E}[x]) \leq \frac{1}{2}\mathbb{V}[x]\sup \phi'' \tag{73}$$

**Lemma 6** (Sharpened Jensen's inequality for ReLU activation). *For ReLU activation $\psi(x) := \max(x, 0)$ and $x \sim p(\cdot)$, we have:*

$$0 \leq \mathbb{E}[\psi(x)] - \psi(\mathbb{E}[x]) \leq \sqrt{\mathbb{V}_p[x]} \tag{74}$$

*Proof.* Since $\psi$ is a convex function, by Jensen's inequality we have $\mathbb{E}[\psi(x)] - \psi(\mathbb{E}[x]) \geq 0$. For the other side, let $\mu := \mathbb{E}_p[x]$ and we have (note that for ReLU, $\psi(x) - \psi(\mu) \leq |x - \mu|$):

$$
\begin{aligned}
\mathbb{E}[\psi(x)] - \psi(\mathbb{E}[x]) &= \int (\psi(x) - \psi(\mu))p(x)\mathrm{d}x \tag{75} \\
&\leq \int |x - \mu|p(x)\mathrm{d}x \tag{76} \\
&\leq \left(\int |x - \mu|^2 p(x)\mathrm{d}x\right)^{1/2}\left(\int p(x)\mathrm{d}x\right)^{1/2} \tag{77} \\
&= \sqrt{\mathbb{V}_p[x]} \tag{78}
\end{aligned}
$$

where the last inequality is due to Cauchy-Schwarz. $\square$

### D.2   MOTIVATION AND DESCRIPTION OF A GENERAL HLTM

Here we describe a general Hierarchical Latent Tree Model (HLTM) of data, and the structure of a multilayer neural network that learns from this data. The structure of the HLTM is motivated by the hierarchical structure of our world in which objects may consist of parts, which in turn may consist of subparts. Moreover the parts and subparts may be in different configurations in relation to each

| Symbol | Definition | Size | Description |
|---|---|---|---|
| $\mathcal{Z}_l$ | | | The set of all latent variables at layer $l$ of the generative model. |
| $\mathcal{N}_l$ | | | The set of all neurons at layer $l$ of the neural network. |
| $\mathcal{N}_\mu$ | | | The set of neurons that corresponds to $z_\mu$. |
| $\mathcal{N}_\mu^{\text{ch}}$ | $\bigcup_{\nu \in \text{ch}(\mu)} \mathcal{N}_\nu$ | | The set of neurons that corresponds to children of latent $z_\mu$. |
| $m_\mu$ | | | Number of possible categorical values taken by $z_\mu \in \{0, \ldots, m_\mu - 1\}$. |
| $\mathbf{0}_\mu, \mathbf{1}_\mu$ | | $m_\mu$ | All-one and all-zero vectors. |
| $P_{\mu\nu}$ | $[\mathbb{P}(z_\nu \| z_\mu)]$ | $m_\mu \times m_\nu$ | The top-down transition probability from $z_\mu$ to $z_\nu$. |
| $\rho_{\mu\nu}$ | $2\mathbb{P}(z_\nu{=}1 \| z_\mu{=}1) - 1$ | scalar in $[-1, 1]$ | Polarity of the transitional probability in the binary case. |
| $P_0$ | $\text{diag}[\mathbb{P}(z_0)]$ | $m_0 \times m_0$ | The diagonal matrix of probability of $z_0$ taking different values. |
| $v_j(z_\mu)$ | $\mathbb{E}_z[f_j \| z_\mu]$ | scalar | Expectation of activation $f_j$ given $z_\mu$ ($z_\mu$'s descendants are marginalized). |
| $\boldsymbol{v}_j$ | $[v_j(z_\mu)]$ | $m_\mu$ | Vector form of $v_j(z_\mu)$. |
| $\boldsymbol{f}_\mu, \boldsymbol{f}_{\mathcal{N}_\mu^{\text{ch}}}$ | $[f_j]_{j \in \mathcal{N}_\mu}, [f_k]_{k \in \mathcal{N}_\mu^{\text{ch}}}$ | $\|\mathcal{N}_\mu\|, \|\mathcal{N}_\mu^{\text{ch}}\|$ | Activations for all nodes $j \in \mathcal{N}_\mu$ and for the children of $\mathcal{N}_\mu$ |
| $\boldsymbol{v}_{0k}, V_{0,\mathcal{N}_\mu^{\text{ch}}}$ | $[\mathbb{E}_z[f_k\|z_0]], [\boldsymbol{v}_{0k}]_{k \in \mathcal{N}_\mu^{\text{ch}}}$ | $m_0, m_0 \times \|\mathcal{N}_\mu^{\text{ch}}\|$ | Expected activation conditioned on $z_0$ |
| $s_k$ | $\frac{1}{2}(v_k(1) - v_k(0))$ | scalar | Discrepancy of node $k$ w.r.t its latent variable $z_{\nu(k)}$. |
| $\boldsymbol{a}_\mu$ | $[\rho_{\mu\nu(k)} s_k]_{k \in \mathcal{N}_\mu^{\text{ch}}}$ | $\|\mathcal{N}_\mu^{\text{ch}}\|$ | Child selectivity vector in the binary case. |

Table 6: Extended notation in HLTM.

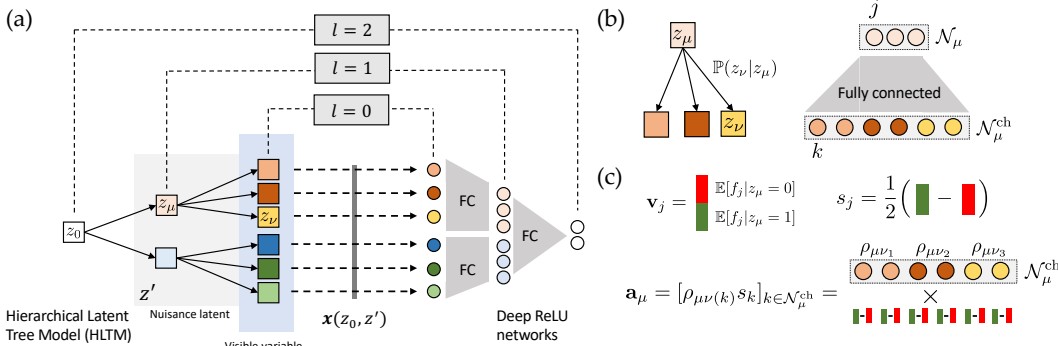

Figure 4: **(a)** The Hierarchical Latent Tree Model (HLTM). **(b)** Correspondence between latent variables and nodes in the intermediate layer of neural networks. **(c)** Definition of $\boldsymbol{v}_j$, $s_j$ and $\boldsymbol{a}_\mu$ in Table. 6.

other in any given instantiation of the object, or any given subpart could be occluded in any given view of an object.

The `HLTM` is a very simple toy model that represents a highly abstract mathematical version of this much more realistic scenario. It consists of a tree structured generative model of data (see Fig. 2(d) and Fig. 4). Simpler versions of this generative model have been used to mathematically model how both infants and linear neural networks learn hierarchically structured data (Saxe et al., 2019). At the top of the tree (i.e. level $L$), a single categorical latent variable $z_0$ takes one of $m_0$ possible integer values values in $\{0 \ldots, m_0 - 1\}$, with a prior distribution $\mathbb{P}(z_0)$. One can roughly think of the value of $z_0$ as denoting the identity of one of $m_0$ possible objects. At level $L - 1$ there is a set of latent variables $\mathcal{Z}_{L-1}$. This set is indexed by $\mu$ and each latent variable $z_\mu$ is itself a categorical variable that takes one of $m_\mu$ values in $\{0, \ldots, m_\mu - 1\}$. Roughly we can think of each latent variable $z_\mu$ as corresponding to a part, and the different values of $z_\mu$ reflect different configurations or occlusion states of that part. The conditional probability distributions $\mathbb{P}(z_\mu | z_0)$, or $m_0$ by $m_\mu$ transition probability matrices, can roughly be thought of as collectively reflecting the distribution over the presence or absence, as well as configurational and occlusional states of each part $\mu$, conditioned on object identity $z_0$. This process can continue onwards to describe subparts of parts, etc...

### D.3 A TWO-LAYER EXAMPLE TO DEMONSTRATE NOTATION

For simplicity, here we demonstrate the notation in a two-layer generative model, and two-layer network with $L = 2$. Thus the top, middle, and leaf levels of the generative model are labelled by $l = 2, 1, 0$, respectively, with corresponding latents $z_0$, $z_\mu$ and $z_\nu$, and the corresponding input, hidden, and final layers of the neural network are labelled by $l = 0, 1, 2$ respectively.

As shown in Fig. 4, at the leaf level $l = 0$ there are a set of visible variables $\mathcal{Z}_0$. This set is indexed by $\nu$ and each visible variable $z_\nu$ is itself a categorical variable that takes one of $m_\nu$ values in the set $\{0 \ldots, m_\nu - 1\}$. Roughly we can think of each $z_\nu$ as a pixel, or more generally, some visible feature. For simplicity, we assume that each part $z_\mu$ at level 1 affects a distinct subset of pixels or visible features $z_\nu$. In essence, we assume each visible variable $z_\nu$ is a child of a unique level 1 latent variable $z_\mu$ in the generative tree process (Fig. 2(d)). In the rough analogy to objects and parts, in this abstraction, each part $\mu$ controls the appearance of a subset of spatially localized nearby pixels or visual features $\nu$ that are all children of part $\mu$. Conversely, each such local cluster of pixels or feature values is influenced by the state of a single part. The conditional probability distribution $\mathbb{P}(z_\nu | z_\mu)$, or $m_\mu$ by $m_\nu$ transition probability matrix, then describes the distribution over pixel or visual feature values $z_\nu$ of each child pixel $\nu$, conditioned on state $z_\mu$ of the parent part $\mu$.

We next consider the two-layer ReLU network that learns from data generated from the two-layer `HLTM` (right hand side of Fig. 2(d)). The neural network has a set of input neurons that are in one to one correspondence with the pixels or visible variables $z_\nu$ that arise at the leaves of the `HLTM`, where $l = 0$. For any given object $z_0$ at layer $l = 2$, and its associated parts states $z_\mu$ at layer $l = 1$, and visible feature values $z_\nu$ at layer $l = 0$, the input neurons of the neural network receive *only* the visible feature values $z_\nu$ as real analog inputs. Thus the neural network **does not have direct access to** the latent variables $z_0$ and $z_\mu$ that generate these visible variables.

**Over-parameterization**. While in the pixel level, there is a one to one correspondence between the children $\nu$ of a subpart $\mu$ and the pixel, in the hidden layer, more than one neuron could correspond to $z_\mu$ (i.e., pool from pixels whose values are influenced by part $\mu$), which is a form of *over-parameterization*. We thus let $\mathcal{N}_\mu$ denote the set of such hidden neurons, and we let $\mathcal{N}_l$ denote the set of all neurons in layer $l$ of the network. Thus $\mathcal{N}_\mu$ is a subset of $\mathcal{N}_1$. We further let $\mathcal{N}_\mu^{\text{ch}}$ denote the subset of neurons in layer 0 that provide input to the hidden layer neurons in $\mathcal{N}_\mu$. Thus $\mathcal{N}_\mu^{\text{ch}}$ is a subset of $\mathcal{N}_0$. Each neuron in the subset $\mathcal{N}_\mu^{\text{ch}}$ is in one to one correspondence with the children (i.e., some $z_\nu$) of latent variable $z_\mu$ in the generative tree (see Fig. 2(d)).

In applying SSL in this setup, each object is specified by a set of values for $z_0$ (object identity), $z_\mu$ (configurational and occlusional states of parts), and $z_\nu$ (pixel values). Given any such object and its realization of configurational and occlusional states of parts and the resulting pixel values, we assume that the process of data augmentation corresponds to resampling $z_\mu$ and $z_\nu$ from the conditional distributions $\mathbb{P}(z_\mu | z_0)$ and $\mathbb{P}(z_\nu | z_\mu)$, while fixing object identity $z_0$. This augmentation process then roughly corresponds to being able to sample the same object under different parts configurations and views.

The key question of interest that we wish to address is, under this generative model of data and model of data augmentation, what do the hidden units of the neural network learn? In particular, can they invert the generative model to convert pixels values at neural layer $l = 0$ into hidden representations at neural layer $l = 1$ that reflect the existence of parts, with their associated states $z_\mu$? More precisely, can the network learn hidden units whose activation across all data points correlates well with the values a latent variable $z_\mu$ takes across all data points? We address this question next, first introducing further simplifying technical assumptions on the generative model and further notation.

## D.4   A GENERAL STRUCTURE OF CONDITIONAL DISTRIBUTIONS IN THE HLTM

For convenience, we define the following symbols for $k \in \mathcal{N}_\mu^{\text{ch}}$ (note that $|\mathcal{N}_\mu^{\text{ch}}| = N_\mu^{\text{ch}}$ is the number of the children of the node set $\mathcal{N}_\mu$):

$$
\begin{align}
\boldsymbol{v}_{\mu k} \quad &:= \quad \mathbb{E}_z\left[f_k | z_\mu\right] = P_{\mu\nu(k)} \boldsymbol{v}_k \in \mathbb{R}^{m_\mu} \tag{79} \\
V_{\mu, \mathcal{N}_\mu^{\text{ch}}} \quad &:= \quad [\boldsymbol{v}_{\mu k}]_{k \in \mathcal{N}_\mu^{\text{ch}}} \tag{80} \\
\tilde{\boldsymbol{v}}_j \quad &:= \quad \left[\mathbb{E}_z\left[\tilde{f}_j | z_\mu\right]\right] = V_{\mu, \mathcal{N}_\mu^{\text{ch}}} \boldsymbol{w}_j \in \mathbb{R}^{m_\mu} \tag{81}
\end{align}
$$

As an extension of binary symmetric $\texttt{HLTM}$, we make an assumption for the transitional probability:

**Assumption 1.** *For $\mu \in \mathcal{Z}_l$ and $\nu \in \mathcal{Z}_{l-1}$, the transitional probability matrix $P_{\mu\nu} := [\mathbb{P}(z_\nu | z_\mu)]$ has decomposition $P_{\mu\nu} = \frac{1}{m_\nu} \mathbf{1}_\mu \mathbf{1}_\nu^{\mathsf{T}} + C_{\mu\nu}$ where $C_{\mu\nu} \mathbf{1}_\nu = \mathbf{0}_\mu$ and $\mathbf{1}_\mu^{\mathsf{T}} C_{\mu\nu} = \mathbf{0}_\nu$.*

Note that $C_{\mu\nu} \mathbf{1} = \mathbf{0}$ is obvious due to the property of conditional probability. The real condition is $\mathbf{1}_\mu^{\mathsf{T}} C_{\mu\nu} = \mathbf{0}_\nu$. If $m_\mu = m_\nu$, then $P_{\mu\nu}$ is a square matrix and Assumption 1 is equivalent to $P_{\mu\nu}$ is double-stochastic. Assumption 1 makes computation of $P_{\mu\nu}$ easy for any $z_\mu$ and $z_\nu$.

**Lemma 7** (Transition Probability). *If Assumption 1 holds, then for $\mu \in \mathcal{Z}_l$, $\nu \in \mathcal{Z}_{l-1}$ and $\alpha \in \mathcal{Z}_{l-2}$, we have:*

$$
P_{\mu\alpha} = P_{\mu\nu} P_{\nu\alpha} = \frac{1}{m_\alpha} \mathbf{1}_\mu \mathbf{1}_\alpha^{\mathsf{T}} + C_{\mu\nu} C_{\nu\alpha} \tag{82}
$$

*In general, for any $\mu \in \mathcal{N}_{l_1}$ and $\alpha \in \mathcal{N}_{l_2}$ with $l_1 > l_2$, we have:*

$$
P_{\mu\alpha} = \frac{1}{m_\alpha} \mathbf{1}_\mu \mathbf{1}_\alpha^{\mathsf{T}} + \prod_{\mu, \dots, \xi, \zeta, \dots, \alpha} C_{\xi\zeta} \tag{83}
$$

*Proof.* Using Assumption 1, we have

$$
\begin{align}
P_{\mu\alpha} \quad &= \quad P_{\mu\nu} P_{\nu\alpha} \tag{84} \\
&= \quad \left(\frac{1}{m_\nu} \mathbf{1}_\mu \mathbf{1}_\nu^{\mathsf{T}} + C_{\mu\nu}\right) \left(\frac{1}{m_\alpha} \mathbf{1}_\nu \mathbf{1}_\alpha^{\mathsf{T}} + C_{\nu\alpha}\right) \tag{85}
\end{align}
$$

since $\mathbf{1}_\nu^{\mathsf{T}} \mathbf{1}_\nu = m_\nu$, $C_{\mu\nu} \mathbf{1}_\nu = \mathbf{0}_\nu$ and $\mathbf{1}_\nu^{\mathsf{T}} C_{\nu\alpha} = \mathbf{0}_\alpha$, the conclusion follows. $\qquad\square$

**Remark.** In the symmetric binary $\texttt{HLTM}$ mentioned in the main text, all $C_{\mu\nu}$ can be parameterized as (here $\boldsymbol{q} := [-1, 1]^{\mathsf{T}}$):

$$
C_{\mu\nu} = C_{\mu\nu}(\rho_{\mu\nu}) = \frac{1}{2} \begin{bmatrix} \rho_{\mu\nu} & -\rho_{\mu\nu} \\ -\rho_{\mu\nu} & \rho_{\mu\nu} \end{bmatrix} = \frac{1}{2} \rho_{\mu\nu} \boldsymbol{q} \boldsymbol{q}^{\mathsf{T}} \tag{86}
$$

This is because $\mathbf{1}_2^{\mathsf{T}} C_{\mu\nu} = \mathbf{0}_2$ and $C_{\mu\nu} \mathbf{1}_2 = \mathbf{0}_2$ provides 4 linear constraints (1 redundant), leaving 1 free parameter, which is the polarity $\rho_{\mu\nu} \in [-1, 1]$ of latent variable $z_\nu$ given its parent $z_\mu$. Moreover, since $\boldsymbol{q}^{\mathsf{T}} \boldsymbol{q} = 2$, the parameterization is close under multiplication:

$$
C(\rho_{\mu\nu}) C(\rho_{\nu\alpha}) = \frac{1}{4} \boldsymbol{q} \boldsymbol{q}^{\mathsf{T}} \boldsymbol{q} \boldsymbol{q}^{\mathsf{T}} \rho_{\mu\nu} \rho_{\nu\alpha} = \frac{1}{2} \boldsymbol{q} \boldsymbol{q}^{\mathsf{T}} \rho_{\mu\nu} \rho_{\nu\alpha} = C(\rho_{\mu\nu} \rho_{\nu\alpha}) \tag{87}
$$

## D.5 THEOREM 6

*Proof.* First note that for each node $k \in \mathcal{N}_\mu^{\text{ch}}$:

$$\boldsymbol{v}_{0k} \quad := \quad [\mathbb{E}_z \left[f_k|z_0\right]] = \sum_{z_\nu} \mathbb{E}_z \left[f_k|z_\nu\right] \mathbb{P}(z_\nu|z_0) = P_{0\nu}\boldsymbol{v}_k \tag{88}$$

$$= \quad \left(\frac{1}{m_\nu}\mathbf{1}_0\mathbf{1}_\nu^\mathsf{T} + C_{0\nu}\right)\boldsymbol{v}_k \tag{89}$$

$$= \quad \frac{1}{m_\nu}\mathbf{1}_0\mathbf{1}_\nu^\mathsf{T}\boldsymbol{v}_k + C_{0\nu}\boldsymbol{v}_k \tag{90}$$

Note that $\mathbf{1}_0\mathbf{1}_\nu^\mathsf{T}\boldsymbol{v}_k$ is a constant regarding to change of $z_0$. So we could remove it when computing covariance operator. On the other hand, for a categorical distribution

$$(\mathbb{P}(z_0 = 0), u(0)), \quad (\mathbb{P}(z_0 = 1), u(1)), \quad \ldots, \quad (\mathbb{P}(z_0 = m_0 - 1), u(m_0 - 1))$$

With $P_0 := \text{diag}[\mathbb{P}(z_0)]$, the mean is $\mathbb{E}_{z_0}\left[u\right] = \mathbf{1}^\mathsf{T}P_0\boldsymbol{u}$ and its covariance can be written as (here $\mathbf{1} = \mathbf{1}_0$):

$$\mathbb{V}_{z_0}[u] = (\boldsymbol{u} - \mathbf{1}\mathbf{1}^\mathsf{T}P_0\boldsymbol{u})^\mathsf{T}P_0(\boldsymbol{u} - \mathbf{1}\mathbf{1}^\mathsf{T}P_0\boldsymbol{u}) = \boldsymbol{u}^\mathsf{T}(P_0 - P_0\mathbf{1}\mathbf{1}^\mathsf{T}P_0)\boldsymbol{u} \tag{91}$$

Note that each column of $V_{0,\mathcal{N}_\mu^{\text{ch}}}$ is $\boldsymbol{v}_{0k}$. Setting $\boldsymbol{u} = \boldsymbol{v}_{0k}$ and we have:

$$\mathbb{V}_{z_0}[\mathbb{E}_z\left[\boldsymbol{f}_{\mathcal{N}_\mu^{\text{ch}}}|z_0\right]] = V_{0,\mathcal{N}_\mu^{\text{ch}}}^\mathsf{T}(P_0 - P_0^\mathsf{T}\mathbf{1}\mathbf{1}^\mathsf{T}P_0)V_{0,\mathcal{N}_\mu^{\text{ch}}} \tag{92}$$

Note that $\mathbb{E}_{z_0}\left[\boldsymbol{v}_{0k}\right] = \frac{1}{m_\nu}\mathbf{1}_\nu^\mathsf{T}\boldsymbol{v}_k + \mathbb{E}_{z_0}\left[C_{0\nu}\boldsymbol{v}_k\right]$, since $\mathbf{1}^\mathsf{T}P_0\mathbf{1} = 1$. With some computation, we could see $\text{Cov}_{z_0}[\boldsymbol{v}_{0k}, \boldsymbol{v}_{0k'}] = \text{Cov}_{z_0}[C_{0\nu(k)}\boldsymbol{v}_k, C_{0\nu(k')}\boldsymbol{v}_{k'}]$.

The equation above can be applied for any cardinality of latent variables. In the binary symmetric case, we have (note here we define $\rho_0 := \mathbb{P}(z_0 = 1) - \mathbb{P}(z_0 = 0)$, and $\boldsymbol{q} := [-1, 1]^\mathsf{T}$):

$$P_0 - P_0\mathbf{1}\mathbf{1}^\mathsf{T}P_0 = \frac{1}{4}(1 - \rho_0^2)\boldsymbol{q}\boldsymbol{q}^\mathsf{T} \tag{93}$$

Note that in the binary symmetric case, according to remarks in Lemma 7, all $C_{\mu\nu} = \frac{1}{2}\rho_{\mu\nu}\boldsymbol{q}\boldsymbol{q}^\mathsf{T}$ and we could compute $C_{0\nu}\boldsymbol{v}_k$:

$$C_{0\nu}\boldsymbol{v}_k = \frac{1}{2}\rho_{0\nu}\boldsymbol{q}\boldsymbol{q}^\mathsf{T}\boldsymbol{v}_k = \rho_{0\nu}\frac{1}{2}(v_k(1) - v_k(0))\boldsymbol{q} = \rho_{0\nu}s_k\boldsymbol{q} \tag{94}$$

where according to Eqn. 87, we have:

$$\rho_{0\nu} := \prod_{0,\ldots,\alpha,\beta,\ldots,\nu} \rho_{\alpha\beta} \tag{95}$$

and the covariance between node $k$ and $k'$ can be computed as:

$$\text{Cov}_{z_0}[\boldsymbol{v}_{0k}, \boldsymbol{v}_{0k'}] \quad = \quad \text{Cov}_{z_0}[C_{0\nu(k)}\boldsymbol{v}_k, C_{0\nu(k')}\boldsymbol{v}_{k'}] \tag{96}$$

$$= \quad \rho_{0\nu(k)}\rho_{0\nu(k')}s_k s_{k'}\frac{1}{4}\boldsymbol{q}^\mathsf{T}\boldsymbol{q}\boldsymbol{q}^\mathsf{T}\boldsymbol{q}(1 - \rho_0^2) \tag{97}$$

$$= \quad \rho_{0\nu(k)}\rho_{0\nu(k')}s_k s_{k'}(1 - \rho_0^2) \tag{98}$$

$$= \quad \rho_{0\mu}^2\rho_{\mu\nu(k)}\rho_{\mu\nu(k')}s_k s_{k'}(1 - \rho_0^2) \tag{99}$$

The last equality is due to the fact that due to tree structure, the path from $z_0$ to all child nodes in $\mathcal{N}_\mu^{\text{ch}}$ must pass $z_\mu$.

Therefore we can compute the covariance operator:

$$\mathbb{V}_{z_0}[\mathbb{E}_z\left[\boldsymbol{f}_{\mathcal{N}_\mu^{\text{ch}}}|z_0\right]] = \rho_{0\mu}^2(1 - \rho_0^2)\boldsymbol{a}_\mu\boldsymbol{a}_\mu^\mathsf{T} \tag{100}$$

When $L \to +\infty$, we have:

$$\rho_{0\nu} := \prod_{0,\ldots,\alpha,\beta,\ldots,\nu} \rho_{\alpha\beta} \to 0 \tag{101}$$

and thus the covariance becomes zero as well. □

### D.6 THEOREM 7

*Proof.* According to our setting, for each node $k \in \mathcal{N}_\mu$, there exists a unique latent variable $z_\nu$ with $\nu = \nu(k)$ that corresponds to it. In the following we omit its dependency on $k$ for brevity.

Since we are dealing with binary case, we define the following for convenience:

$$v_k^+ := v_k(1) \tag{102}$$

$$v_k^- := v_k(0) \tag{103}$$

$$\bar{v}_k := \frac{1}{2}(v_k^+ + v_k^-) = \frac{1}{2}\left(\mathbb{E}_z\left[f_k|z_\nu = 1\right] + \mathbb{E}_z\left[f_k|z_\nu = 0\right]\right) \tag{104}$$

$$\bar{\boldsymbol{v}}_{\mathcal{N}_\mu^{\mathrm{ch}}} := [\bar{v}_k]_{k \in \mathcal{N}_\mu^{\mathrm{ch}}} \in \mathbb{R}^{|\mathcal{N}_\mu^{\mathrm{ch}}|} \tag{105}$$

$$s_k := \frac{1}{2}(v_k^+ - v_k^-) = \frac{1}{2}\left(\mathbb{E}_z\left[f_k|z_\nu = 1\right] - \mathbb{E}_z\left[f_k|z_\nu = 0\right]\right) \tag{106}$$

$$\boldsymbol{s}_{\mathcal{N}_\mu^{\mathrm{ch}}} := [s_k]_{k \in \mathcal{N}_\mu^{\mathrm{ch}}} \in \mathbb{R}^{|\mathcal{N}_\mu^{\mathrm{ch}}|} \tag{107}$$

We also define the *sensitivity* of node $k$ to be $\lambda_k := |(v_k^+)^2 - (v_k^-)^2|$. Intuitively, a large $\lambda_k$ means that the node $k$ is sensitive for changes of latent variable $z_\nu$. If $\lambda_k = 0$, then the node $k$ is invariant to latent variable $z_\nu$.

We first consider pre-activation $\tilde{f}_j := \sum_k w_{jk} f_k$ and its expectation with respect to latent variable $z$:

$$\tilde{v}_j^+ := \mathbb{E}_z\left[\tilde{f}_j \Big| z_\mu = 1\right], \quad \tilde{v}_j^- := \mathbb{E}_z\left[\tilde{f}_j \Big| z_\mu = 0\right] \tag{108}$$

Note that for each node $k \in \mathcal{N}_\mu^{\mathrm{ch}}$ we have:

$$v_{\mu k}^+ = \bar{v}_k + \rho_{\mu\nu} s_k, \quad v_{\mu k}^- = \bar{v}_k - \rho_{\mu\nu} s_k \tag{109}$$

Let $\boldsymbol{a}_\mu := [a_k]_{k \in \mathcal{N}_\mu^{\mathrm{ch}}} := [\rho_{\mu\nu} s_k]_{k \in \mathcal{N}_\mu^{\mathrm{ch}}}$ and

$$\boldsymbol{u}_{\mathcal{N}_\mu^{\mathrm{ch}}}^+ := V_{\mu,\mathcal{N}_\mu^{\mathrm{ch}}}^+ := \left[\mathbb{E}\left[f_k|z_\mu = 1\right]\right] = \bar{\boldsymbol{v}}_{\mathcal{N}_\mu^{\mathrm{ch}}} + \boldsymbol{a}_\mu \tag{110}$$

$$\boldsymbol{u}_{\mathcal{N}_\mu^{\mathrm{ch}}}^- := V_{\mu,\mathcal{N}_\mu^{\mathrm{ch}}}^- := \left[\mathbb{E}\left[f_k|z_\mu = 0\right]\right] = \bar{\boldsymbol{v}}_{\mathcal{N}_\mu^{\mathrm{ch}}} - \boldsymbol{a}_\mu \tag{111}$$

Then we have $\tilde{v}_j^+ = \boldsymbol{w}_j^\mathsf{T} \boldsymbol{u}_{\mathcal{N}_\mu^{\mathrm{ch}}}^+$ and $\tilde{v}_j^- = \boldsymbol{w}_j^\mathsf{T} \boldsymbol{u}_{\mathcal{N}_\mu^{\mathrm{ch}}}^-$.

Note that $\boldsymbol{w}_j$ is a random variable with each entry $w_{jk} \sim \mathrm{Uniform}\left[-\sigma_w \sqrt{\frac{3}{|\mathcal{N}_\mu^{\mathrm{ch}}|}}, \sigma_w \sqrt{\frac{3}{|\mathcal{N}_\mu^{\mathrm{ch}}|}}\right]$. It is easy to verify that $\mathbb{E}\left[w_{jk}\right] = 0$ and $\mathbb{V}[w_{jk}] = \sigma_w^2/|\mathcal{N}_\mu^{\mathrm{ch}}|$. Therefore, for two dimensional vector $\tilde{\boldsymbol{v}}_j = [\tilde{v}_j^+, \tilde{v}_j^-]^\mathsf{T}$, we can compute its first and second order moments: $\mathbb{E}_w\left[\tilde{\boldsymbol{v}}_j\right] = \boldsymbol{0}$ and $\mathbb{V}_w[\tilde{\boldsymbol{v}}_j] = \frac{\sigma_w^2}{|\mathcal{N}_\mu^{\mathrm{ch}}|} V_{\mu,\mathcal{N}_\mu^{\mathrm{ch}}} V_{\mu,\mathcal{N}_\mu^{\mathrm{ch}}}^\mathsf{T} = \frac{\sigma_w^2}{|\mathcal{N}_\mu^{\mathrm{ch}}|}[\boldsymbol{u}_{\mathcal{N}_\mu^{\mathrm{ch}}}^+, \boldsymbol{u}_{\mathcal{N}_\mu^{\mathrm{ch}}}^-]^\mathsf{T}[\boldsymbol{u}_{\mathcal{N}_\mu^{\mathrm{ch}}}^+, \boldsymbol{u}_{\mathcal{N}_\mu^{\mathrm{ch}}}^-]$.

Define the positive and negative set (note that $a_k := \rho_{\mu\nu} s_k$):

$$A_+ = \{k : a_k \geq 0\}, \quad A_- = \{k : a_k < 0\} \tag{112}$$

Without loss of generality, assume that $\sum_{k \in A_+} a_k^2 \geq \sum_{k \in A_-} a_k^2$. In the following, we show there exists $j$ with $\lambda_j$ is greater than some positive threshold. Otherwise the proof is symmetric and we can show $\lambda_j$ is lower than some negative threshold.

When $|\mathcal{N}_\mu^{\mathrm{ch}}|$ is large, by Central Limit Theorem, $\boldsymbol{v}$ can be regarded as zero-mean 2D Gaussian distribution and we have for some $c > 0$:

$$\mathbb{P}\left(\tilde{v}_j^+ \geq \frac{\sqrt{c}\sigma_w}{\sqrt{|\mathcal{N}_\mu^{\mathrm{ch}}|}}\|\boldsymbol{u}_{\mathcal{N}_\mu^{\mathrm{ch}}}^+\|\right) = \frac{1 - \mathrm{erf}(\sqrt{c}/2)}{2} \tag{113}$$

Moreover, if $\boldsymbol{a}_l \neq \boldsymbol{0}$, then the following probability is also not small :

$$\mathbb{P}\left(\tilde{v}_j^+ \geq \frac{\sqrt{c}\sigma_w}{\sqrt{|\mathcal{N}_\mu^{\mathrm{ch}}|}}\|\boldsymbol{u}_{\mathcal{N}_\mu^{\mathrm{ch}}}^+\| \quad \text{and} \quad \tilde{v}_j^- < 0\right) \tag{114}$$

Therefore, when $|\mathcal{N}_\mu| = O(\exp(c))$, with high probability, there exists $\boldsymbol{w}_j$ so that

$$\tilde{v}_j^+ = \boldsymbol{w}_j^\mathsf{T} \boldsymbol{u}_{\mathcal{N}_\mu^{\mathrm{ch}}}^+ \geq \frac{\sqrt{c}\sigma_w}{\sqrt{|\mathcal{N}_\mu^{\mathrm{ch}}|}} \|\boldsymbol{u}_{\mathcal{N}_\mu^{\mathrm{ch}}}^+\|, \qquad \tilde{v}_j^- = \boldsymbol{w}_j^\mathsf{T} \boldsymbol{u}_{\mathcal{N}_\mu^{\mathrm{ch}}}^- < 0 \tag{115}$$

Since $\bar{\boldsymbol{v}}_{\mathcal{N}_\mu^{\mathrm{ch}}} \geq 0$ (all $f_k$ are after ReLU and non-negative), this leads to:

$$\tilde{v}_j^+ \geq \frac{\sqrt{c}\sigma_w}{\sqrt{|\mathcal{N}_\mu^{\mathrm{ch}}|}} \|\boldsymbol{u}_{\mathcal{N}_\mu^{\mathrm{ch}}}^+\| \geq \frac{\sqrt{c}\sigma_w}{\sqrt{|\mathcal{N}_\mu^{\mathrm{ch}}|}} \sqrt{\sum_{k \in A_+} a_k^2} \geq \sigma_w \sqrt{\frac{c}{2|\mathcal{N}_\mu^{\mathrm{ch}}|} \sum_{k \in \mathcal{N}_\mu^{\mathrm{ch}}} \rho_{\mu\nu}^2 s_k^2} \tag{116}$$

By Jensen's inequality, we have (note that $\psi(x) := \max(x, 0)$ is the ReLU activation):

$$v_j^+ = \mathbb{E}_z\left[f_j | z_\mu = 1\right] = \mathbb{E}_z\left[\psi(\tilde{f}_j) | z_\mu = 1\right] \tag{117}$$

$$\geq \psi\left(\mathbb{E}_z\left[\tilde{f}_j \Big| z_\mu = 1\right]\right) = \psi(\tilde{v}_j^+) \geq \sigma_w \sqrt{\frac{c}{2|\mathcal{N}_\mu^{\mathrm{ch}}|} \sum_{k \in \mathcal{N}_\mu^{\mathrm{ch}}} \rho_{\mu\nu}^2 s_k^2} \tag{118}$$

On the other hand, we also want to compute $v_j^- := \mathbb{E}_z\left[f_j | z_\mu = 0\right]$ using sharpened Jensen's inequality (Lemma 6). For this we need to compute the conditional covariance $\mathbb{V}_z[\tilde{f}_j | z_\mu]$:

$$\mathbb{V}_z[\tilde{f}_j | z_\mu] \overset{\text{②}}{=} \sum_k w_{jk}^2 \mathbb{V}_z[f_k | z_\mu] \overset{\text{③}}{\leq} \frac{3\sigma_w^2}{|\mathcal{N}_\mu^{\mathrm{ch}}|} \sum_k \mathbb{V}_z[f_k | z_\mu] \tag{119}$$

$$= \frac{3\sigma_w^2}{|\mathcal{N}_\mu^{\mathrm{ch}}|} \sum_k \left(\mathbb{E}_{z_\nu | z_\mu}\left[\mathbb{V}[f_k | z_\nu]\right] + \mathbb{V}_{z_\nu | z_\mu}\left[\mathbb{E}_z\left[f_k | z_\nu\right]\right]\right) \tag{120}$$

$$\leq 3\sigma_w^2\left(\sigma_l^2 + \frac{1}{|\mathcal{N}_\mu^{\mathrm{ch}}|} \sum_k \mathbb{V}_{z_\nu | z_\mu}\left[\mathbb{E}_z\left[f_k | z_\nu\right]\right]\right) \tag{121}$$

Note that ② is due to conditional independence: $f_k$ as the computed activation, only depends on latent variable $z_\nu$ and its descendants. Given $z_\mu$, all $z_\nu$ and their respective descendants are independent of each other and so does $f_k$. ③ is due to the fact that each $w_{jk}$ are sampled from uniform distribution and $|w_{jk}| \leq \sigma_w \sqrt{\frac{3}{|\mathcal{N}_\mu^{\mathrm{ch}}|}}$.

Here $\mathbb{V}_{z_\nu | z_\mu}\left[\mathbb{E}_z\left[f_k | z_\nu\right]\right] = s_k^2(1 - \rho_{\mu\nu}^2)$ can be computed analytically. It is the variance of a binomial distribution: with probability $\frac{1}{2}(1 + \rho_{\mu\nu})$ we get $v_k^+$ otherwise get $v_k^-$. Therefore, we finally have:

$$\mathbb{V}_z[\tilde{f}_j | z_\mu] \leq 3\sigma_w^2\left(\sigma_l^2 + \frac{1}{|\mathcal{N}_\mu^{\mathrm{ch}}|} \sum_k s_k^2(1 - \rho_{\mu\nu}^2)\right) \tag{122}$$

As a side note, using Lemma 4, since ReLU function $\psi$ has Lipschitz constant $\leq 1$ (empirically it is smaller), we know that:

$$\mathbb{V}_z[f_j | z_\mu] \leq 3\sigma_w^2\left(\sigma_l^2 + \frac{1}{|\mathcal{N}_\mu^{\mathrm{ch}}|} \sum_k s_k^2(1 - \rho_{\mu\nu}^2)\right) \tag{123}$$

Finally using Lemma 6 and $\tilde{v}_j^- < 0$, we have:

$$v_j^- = \mathbb{E}_z\left[f_j | z_\mu = 0\right] = \mathbb{E}_z\left[\psi(\tilde{f}_j) | z_\mu = 0\right] \tag{124}$$

$$\leq \psi\left(\mathbb{E}_z\left[\tilde{f}_j \Big| z_\mu = 0\right]\right) + \sqrt{\mathbb{V}_z[\tilde{f}_j | z_\mu = 0]} \tag{125}$$

$$= \sqrt{\mathbb{V}_z[\tilde{f}_j | z_\mu = 0]} \tag{126}$$

$$\leq \sigma_w \sqrt{3\sigma_l^2 + \frac{3}{|\mathcal{N}_\mu^{\mathrm{ch}}|} \sum_k s_k^2(1 - \rho_{\mu\nu}^2)} \tag{127}$$

Combining Eqn. 118 and Eqn. 127, we have a bound for $\lambda_j$:

$$\lambda_j = (v_j^+)^2 - (v_j^-)^2 \geq 3\sigma_w^2 \left[ \frac{1}{|\mathcal{N}_\mu^{\mathrm{ch}}|} \sum_k s_k^2 \left( \frac{c+6}{6} \rho_{\mu\nu}^2 - 1 \right) - \sigma_l^2 \right] \tag{128}$$

$\square$

# E    THE ANALYSIS OF BYOL IN SEC. 5

## E.1    DERIVATION OF BYOL GRADIENT

Note that for BYOL, we have:

$$\mathrm{vec}\left( \frac{\partial r}{\partial W_l} \right) = K_l(\boldsymbol{x}_1; \mathcal{W}) \left[ K_l^\mathsf{T}(\boldsymbol{x}_1; \mathcal{W})\mathrm{vec}(W_l) - K_l^\mathsf{T}(\boldsymbol{x}_2; \mathcal{W}')\mathrm{vec}(W_l') \right] \tag{129}$$

under large batchsize, we have (note that we omit $\mathcal{W}$ for any term that depends on $\mathcal{W}$, but make dependence of $\mathcal{W}'$ explicit in the math expression):

$$\mathrm{vec}\left( \frac{\partial r}{\partial W_l} \right) = \mathbb{E}_{\boldsymbol{x}\sim p(\cdot)} \left[ \mathbb{E}_{\boldsymbol{x}'\sim p_{\mathrm{aug}}(\cdot|\boldsymbol{x})} \left[ K_l(\boldsymbol{x}')K_l^\mathsf{T}(\boldsymbol{x}') \right] \mathrm{vec}(W_l) - \bar{K}_l(\boldsymbol{x})\bar{K}_l^\mathsf{T}(\boldsymbol{x}; \mathcal{W}')\mathrm{vec}(W_l') \right]$$

For brevity, we write $\mathbb{E}_{\boldsymbol{x}}\left[ \cdot \right] := \mathbb{E}_{\boldsymbol{x}\sim p(\cdot)}\left[ \cdot \right]$ and $\mathbb{E}_{\boldsymbol{x}'}\left[ \cdot \right] := \mathbb{E}_{\boldsymbol{x}'\sim p_{\mathrm{aug}}(\cdot|\boldsymbol{x})}\left[ \cdot \right]$. Similar for $\mathbb{V}$. And the equation above can be written as:

$$\mathrm{vec}\left( \frac{\partial r}{\partial W_l} \right) = \mathbb{E}_{\boldsymbol{x}} \left\{ \mathbb{V}_{\boldsymbol{x}'}[K_l(\boldsymbol{x}')] \right\} \mathrm{vec}(W_l) \tag{130}$$

$$+ \mathbb{E}_{\boldsymbol{x}} \left\{ \bar{K}_l(\boldsymbol{x}) \left[ \bar{K}_l^\mathsf{T}(\boldsymbol{x})\mathrm{vec}(W_l) - \bar{K}_l^\mathsf{T}(\boldsymbol{x}; \mathcal{W}')\mathrm{vec}(W_l') \right] \right\} \tag{131}$$

In terms of weight update by gradient descent, since $\Delta W_l = -\frac{\partial r}{\partial W_l}$, we have:

$$\mathrm{vec}\left( \Delta W_l \right) = -\mathbb{E}_{\boldsymbol{x}} \left\{ \mathbb{V}_{\boldsymbol{x}'}[K_l(\boldsymbol{x}')] \right\} \mathrm{vec}(W_l) \tag{132}$$

$$- \mathbb{E}_{\boldsymbol{x}} \left\{ \bar{K}_l(\boldsymbol{x}) \left[ \bar{K}_l^\mathsf{T}(\boldsymbol{x})\mathrm{vec}(W_l) - \bar{K}_l^\mathsf{T}(\boldsymbol{x}; \mathcal{W}')\mathrm{vec}(W_l') \right] \right\} \tag{133}$$

If we consider the special case $\mathcal{W} = \mathcal{W}'$, then the last two terms cancelled out, yielding:

$$\mathrm{vec}(\Delta W_l)_{\mathrm{sym}} = -\mathbb{E}_{\boldsymbol{x}} \left\{ \mathbb{V}_{\boldsymbol{x}'}[K_l(\boldsymbol{x}')] \right\} \mathrm{vec}(W_l) \tag{134}$$

And the general update (Eqn. 136) can be written as:

$$\mathrm{vec}\left( \Delta W_l \right) = \mathrm{vec}(\Delta W_l)_{\mathrm{sym}} \tag{135}$$

$$- \mathbb{E}_{\boldsymbol{x}} \left\{ \bar{K}_l(\boldsymbol{x}) \left[ \bar{K}_l^\mathsf{T}(\boldsymbol{x})\mathrm{vec}(W_l) - \bar{K}_l^\mathsf{T}(\boldsymbol{x}; \mathcal{W}')\mathrm{vec}(W_l') \right] \right\} \tag{136}$$

## E.2    THEOREM 8

*Proof.* When BN is present, Eqn. 129 needs to be corrected with an additional term, $\widetilde{\frac{\partial r}{\partial W_l}} := \frac{\partial r}{\partial W_l} - \delta W_l^{\mathrm{BN}}$, where $\delta W_l^{\mathrm{BN}}$ is defined as follows:

$$\delta W_l^{\mathrm{BN}} := \frac{1}{|B|} \sum_{i \in B} D_l^i \bar{\boldsymbol{g}}_l \boldsymbol{f}_{l-1}^{i\mathsf{T}} \tag{137}$$

From the proof of Theorem 1 (see Eqn. 26), we know that for each sample $i \in B$ (note that by definition, the back-propagated gradient *after* nonlinearity $\tilde{\boldsymbol{g}}_l^i$ equals to $D_l^i \boldsymbol{g}_l^i$, where $\boldsymbol{g}_l^i$ is the back-propagated gradient *before* nonlinearity):

$$D_l^i \boldsymbol{g}_l^i = J_l^{i\mathsf{T}}[J_l^i W_l \boldsymbol{f}_{l-1}^i - J_l^i(\mathcal{W}')W_l' \boldsymbol{f}_{l-1}^i(\mathcal{W}')] \tag{138}$$

Since the network is linear from layer $l$ to the topmost layer $L$, we have $D_l^i = \bar{D}_l$. Since the only input dependent part in $J_l^i$ is the gating function between the current layer $l$ and the topmost layer $L$, for linear network the gating is always 1 and thus $\bar{J}_l = J_l^i$ and is independent of input data. We

now have (note that we omit $\mathcal{W}$ for any terms that are dependent on $\mathcal{W}$, but will write $\mathcal{W}'$ explicitly for terms that are depend on $\mathcal{W}'$):

$$
\delta W_l^{\mathrm{BN}} \quad := \quad \frac{1}{|B|} \sum_{i \in B} D_l^i \bar{\boldsymbol{g}}_l \boldsymbol{f}_{l-1}^{i\mathsf{T}} = -\bar{D}_l \bar{\boldsymbol{g}}_l \bar{\boldsymbol{f}}_{l-1}^{\mathsf{T}} \tag{139}
$$

$$
= \quad \bar{J}_l^{\mathsf{T}} [\bar{J}_l W_l \bar{\boldsymbol{f}}_{l-1} - \bar{J}_l(\mathcal{W}') W_l' \bar{\boldsymbol{f}}_{l-1}(\mathcal{W}')] \bar{\boldsymbol{f}}_{l-1}^{\mathsf{T}} \tag{140}
$$

Therefore we have:

$$
\mathrm{vec}(\delta W_l^{\mathrm{BN}}) = (\bar{\boldsymbol{f}}_{l-1} \otimes \bar{J}_l^{\mathsf{T}}) \left[ (\bar{\boldsymbol{f}}_{l-1} \otimes \bar{J}_l^{\mathsf{T}}) \mathrm{vec}(W_l) - (\bar{\boldsymbol{f}}_{l-1}(\mathcal{W}') \otimes \bar{J}_l^{\mathsf{T}}(\mathcal{W}')) \mathrm{vec}(W_l') \right] \tag{141}
$$

Note that by assumption, since $\bar{J}_l$ doesn't depend on the input data, we have

$$
\bar{\boldsymbol{f}}_{l-1} \otimes \bar{J}_l^{\mathsf{T}} = \mathbb{E}_B \left[ \boldsymbol{f}_{l-1} \right] \otimes \bar{J}_l^{\mathsf{T}} = \mathbb{E}_B \left[ \boldsymbol{f}_{l-1} \otimes \bar{J}_l^{\mathsf{T}} \right] \tag{142}
$$

Taking large batchsize limits and notice that the batch $B$ could contain any augmented data generated from independent samples from $p(\cdot)$, we have:

$$
\mathrm{vec}(\delta W_l^{\mathrm{BN}}) \quad = \quad \mathbb{E}_{\boldsymbol{x}, \boldsymbol{x}'} \left[ K_l(\boldsymbol{x}') \right] \mathbb{E}_{\boldsymbol{x}, \boldsymbol{x}'} \left[ K_l^{\mathsf{T}}(\boldsymbol{x}') \right] \mathrm{vec}(W_l) \tag{143}
$$

$$
- \quad \mathbb{E}_{\boldsymbol{x}, \boldsymbol{x}'} \left[ K_l(\boldsymbol{x}') \right] \mathbb{E}_{\boldsymbol{x}, \boldsymbol{x}'} \left[ K_l^{\mathsf{T}}(\boldsymbol{x}'; \mathcal{W}') \right] \mathrm{vec}(W_l') \tag{144}
$$

An important thing is that the expectation is taking over $\boldsymbol{x} \sim p(\boldsymbol{x})$ and $\boldsymbol{x}' \sim p_{\mathrm{aug}}(\cdot|\boldsymbol{x})$. Intuitively, this is because $\bar{\boldsymbol{f}}_{l-1}$ and $\bar{\boldsymbol{g}}_l$ are averages over the entire batch, which has both intra-sample and inter-sample variation.

With augment-mean connection $\bar{K}_l(\boldsymbol{x})$ we could write:

$$
\mathrm{vec}(\delta W_l^{\mathrm{BN}}) \quad = \quad \mathbb{E}_{\boldsymbol{x}} \left[ \bar{K}_l(\boldsymbol{x}) \right] \mathbb{E}_{\boldsymbol{x}} \left[ \bar{K}_l^{\mathsf{T}}(\boldsymbol{x}) \right] \mathrm{vec}(W_l) - \mathbb{E}_{\boldsymbol{x}} \left[ \bar{K}_l(\boldsymbol{x}) \right] \mathbb{E}_{\boldsymbol{x}} \left[ \bar{K}_l^{\mathsf{T}}(\boldsymbol{x}; \mathcal{W}') \right] \mathrm{vec}(W_l')
$$

$$
= \quad \mathbb{E}_{\boldsymbol{x}} \left[ \bar{K}_l(\boldsymbol{x}) \right] \left\{ \mathbb{E}_{\boldsymbol{x}} \left[ \bar{K}_l^{\mathsf{T}}(\boldsymbol{x}) \right] \mathrm{vec}(W_l) - \mathbb{E}_{\boldsymbol{x}} \left[ \bar{K}_l^{\mathsf{T}}(\boldsymbol{x}; \mathcal{W}') \right] \mathrm{vec}(W_l') \right\} \tag{145}
$$

Plug in $\delta W_{l,BN}$ into Eqn. 136 and we have corrected gradient for BYOL:

$$
\mathrm{vec}\left( \widetilde{\frac{\partial r}{\partial W_l}} \right) \quad = \quad \mathrm{vec}\left( \frac{\partial r}{\partial W_l} \right) - \mathrm{vec}\left( \delta W_l^{\mathrm{BN}} \right) \tag{146}
$$

$$
= \quad \mathbb{E}_{\boldsymbol{x}} \left[ \mathbb{V}_{\boldsymbol{x}' \sim p_{\mathrm{aug}}(\cdot|\boldsymbol{x})} \left[ K_l(\boldsymbol{x}') \right] \right] \mathrm{vec}(W_l) + \mathbb{V}_{\boldsymbol{x}} \left[ \bar{K}_l(\boldsymbol{x}) \right] \mathrm{vec}(W_l) \tag{147}
$$

$$
- \quad \mathrm{Cov}_{\boldsymbol{x}} \left[ \bar{K}_l(\boldsymbol{x}), \bar{K}_l(\boldsymbol{x}; \mathcal{W}') \right] \mathrm{vec}(W_l') \tag{148}
$$

And the weight update $\widetilde{\Delta W_l} = \Delta W_l + \delta W_l^{\mathrm{BN}}$ is:

$$
\mathrm{vec}\left( \widetilde{\Delta W_l} \right) \quad = \quad -\mathbb{E}_{\boldsymbol{x}} \left[ \mathbb{V}_{\boldsymbol{x}' \sim p_{\mathrm{aug}}(\cdot|\boldsymbol{x})} \left[ K_l(\boldsymbol{x}') \right] \right] \mathrm{vec}(W_l) - \mathbb{V}_{\boldsymbol{x}} \left[ \bar{K}_l(\boldsymbol{x}) \right] \mathrm{vec}(W_l) \tag{149}
$$

$$
+ \quad \mathrm{Cov}_{\boldsymbol{x}} \left[ \bar{K}_l(\boldsymbol{x}), \bar{K}_l(\boldsymbol{x}; \mathcal{W}') \right] \mathrm{vec}(W_l') \tag{150}
$$

Using Eqn. 134, we have:

$$
\mathrm{vec}\left( \widetilde{\Delta W_l} \right) \quad = \quad \mathrm{vec}(\Delta W_l) + \mathrm{vec}(\delta W_l^{\mathrm{BN}}) \tag{151}
$$

$$
= \quad \mathrm{vec}(\Delta W_l)_{\mathrm{sym}} \tag{152}
$$

$$
- \quad \mathbb{V}_{\boldsymbol{x}} \left[ \bar{K}_l(\boldsymbol{x}) \right] \mathrm{vec}(W_l) + \mathrm{Cov}_{\boldsymbol{x}} \left[ \bar{K}_l(\boldsymbol{x}), \bar{K}_l(\boldsymbol{x}; \mathcal{W}') \right] \mathrm{vec}(W_l') \tag{153}
$$

$$\square$$

### E.3  COROLLARY 1

*Proof.* In this case, both the target and online networks use the same weight and there is no predictor. This means $\mathcal{W}' = \mathcal{W}$. Therefore, in Eqn. 145, all $W_l = W_l'$ and $\delta W_l^{\mathrm{BN}} = 0$.

Note that for SimCLR, the loss function contains both positive pair squared distance $r_+$ and negative pair squared distance $r_-$. The argument above shows that $\delta W_l^{\mathrm{BN}} = 0$ for positive pair distance $r_+$. For negative pair distance $r_-$, with the same logic in Theorem. 8, we will see $\delta W_l^{\mathrm{BN}}$ takes the same form as Eqn. 145 and thus is zero as well. $\square$

**Remarks**. Note that BatchNorm does not matter in terms of gradient update, modulo its benefit during optimization. This is justified in the recent blogpost (Fetterman and Albrecht, 2020).

### E.4 COROLLARY 2

*Proof.* By our condition, we consider the case that the extra predictor is a linear layer: $\mathcal{W}_{\text{pred}} = \{W_{\text{pred}}\}$. Note that $W_{\text{pred}} \in \mathbb{R}^{n_L \times n_L}$ is a squared matrix, otherwise we cannot compute the loss function between the output $\boldsymbol{f}_{L'}$ from the online network with the output $\boldsymbol{f}_L$ from the target network.

In this case, for connection $K_l(\boldsymbol{x})$ in the common part of the network (in $\mathcal{W}_{\text{base}}$), we have:

$$\begin{align}
K_l(\boldsymbol{x}) &= \boldsymbol{f}_{l-1}(\boldsymbol{x}) \otimes J_l^\mathsf{T}(\boldsymbol{x}) = \boldsymbol{f}_{l-1}(\boldsymbol{x}) \otimes J_{l,\text{base}}^\mathsf{T}(\boldsymbol{x}) W_{\text{pred}}^\mathsf{T} \tag{154} \\
&= (\boldsymbol{f}_{l-1}(\boldsymbol{x}) \otimes J_{l,\text{base}}^\mathsf{T}(\boldsymbol{x})) W_{\text{pred}}^\mathsf{T} \tag{155}
\end{align}$$

Here $J_{l,\text{base}}(\boldsymbol{x})$ is the Jacobian from the current layer $l$ to the layer right before the extra predictor. The last equality is due to the fact that $\boldsymbol{f}_{l-1}$ is a vector. Therefore, for augment-mean $\bar{K}_l(\boldsymbol{x})$, since $W_{\text{pred}}$ doesn't depend on the input data distribution, we have:

$$\bar{K}_l(\boldsymbol{x}) = \bar{K}_{l,\text{base}}(\boldsymbol{x}) W_{\text{pred}}^\mathsf{T} \tag{156}$$

where $\bar{K}_{l,\text{base}}(\boldsymbol{x}) := \bar{K}_l(\boldsymbol{x}; \mathcal{W}_{\text{base}})$. To make things concise, let $\hat{K}_l(\boldsymbol{x}) := \bar{K}_l(\boldsymbol{x}) - \mathbb{E}_{\boldsymbol{x}}\left[\bar{K}_l(\boldsymbol{x})\right]$. Obviously we have $\hat{K}_l(\boldsymbol{x}) = \hat{K}_{l,\text{base}}(\boldsymbol{x}) W_{\text{pred}}^\mathsf{T}$. And the covariance operator becomes:

$$\mathbb{V}_{\boldsymbol{x}}[\bar{K}_l(\boldsymbol{x}; \mathcal{W})] = \mathbb{E}_{\boldsymbol{x}}\left[\hat{K}_{l,\text{base}}(\boldsymbol{x}) W_{\text{pred}}^\mathsf{T} W_{\text{pred}} \hat{K}_{l,\text{base}}^\mathsf{T}(\boldsymbol{x})\right] \tag{157}$$

Now let $\widehat{\Delta W_l}$ be the last two terms in Eqn. 14:

$$\text{vec}(\widehat{\Delta W_l}) = -\mathbb{V}_{\boldsymbol{x}}\left[\bar{K}_l(\boldsymbol{x}; \mathcal{W})\right] \text{vec}(W_l) + \text{Cov}_{\boldsymbol{x}}\left[\bar{K}_l(\boldsymbol{x}; \mathcal{W}), \bar{K}_l(\boldsymbol{x}; \mathcal{W}')\right] \text{vec}(W_l') \tag{158}$$

Since there is no EMA, $\mathcal{W}_{\text{base}} = \mathcal{W}_{\text{base}}'$ and we have:

$$\begin{align}
\text{vec}(\widehat{\Delta W_l}) &= \left\{-\mathbb{V}_{\boldsymbol{x}}\left[\bar{K}_l(\boldsymbol{x}; \mathcal{W})\right] + \text{Cov}_{\boldsymbol{x}}\left[\bar{K}_l(\boldsymbol{x}; \mathcal{W}), \bar{K}_l(\boldsymbol{x}; \mathcal{W}')\right]\right\} \text{vec}(W_l) \tag{159} \\
&= \mathbb{E}_{\boldsymbol{x}}\left[\hat{K}_{l,\text{base}}(\boldsymbol{x}) W_{\text{pred}}^\mathsf{T}(I - W_{\text{pred}}) \hat{K}_{l,\text{base}}^\mathsf{T}(\boldsymbol{x})\right] \text{vec}(W_l) \tag{160}
\end{align}$$

Therefore, the final expression of $\text{vec}(\widetilde{\Delta W_l})$ is the following:

$$\begin{align}
\text{vec}(\widetilde{\Delta W_l}) &= \text{vec}(\Delta W_l) + \text{vec}(\delta W_l^{\text{BN}}) \\
&= \left\{-\mathbb{E}_{\boldsymbol{x}}\left[\mathbb{V}_{\boldsymbol{x}' \sim p_{\text{aug}}(\cdot|\boldsymbol{x})}\left[K_l(\boldsymbol{x}')\right]\right] + \mathbb{E}_{\boldsymbol{x}}\left[\hat{K}_{l,\text{base}}(\boldsymbol{x}) W_{\text{pred}}^\mathsf{T}(I - W_{\text{pred}}) \hat{K}_{l,\text{base}}^\mathsf{T}(\boldsymbol{x})\right]\right\} \text{vec}(W_l)
\end{align}$$

If there is no stop gradient on the target network side, and we receive gradient from both the online and the target network, then for any common layer $l$, the weight update $\text{vec}(\widetilde{\widetilde{\Delta W_l}})$ becomes symmetric (note that this can be derived by swapping $\mathcal{W}'$ with $\mathcal{W}$ and add the two terms together):

$$\begin{align}
\text{vec}(\widetilde{\widetilde{\Delta W_l}}) &= 2\text{vec}(\Delta W_l)_{\text{sym}} \tag{161} \\
&\quad - \left(\mathbb{V}_{\boldsymbol{x}}\left[\bar{K}_l(\boldsymbol{x}; \mathcal{W})\right] + \mathbb{V}_{\boldsymbol{x}}\left[\bar{K}_l(\boldsymbol{x}; \mathcal{W}')\right]\right) \text{vec}(W_l) \tag{162} \\
&\quad + \text{Cov}_{\boldsymbol{x}}\left[\bar{K}_l(\boldsymbol{x}; \mathcal{W}), \bar{K}_l(\boldsymbol{x}; \mathcal{W}')\right] \text{vec}(W_l) \tag{163} \\
&\quad + \text{Cov}_{\boldsymbol{x}}\left[\bar{K}_l(\boldsymbol{x}; \mathcal{W}'), \bar{K}_l(\boldsymbol{x}; \mathcal{W})\right] \text{vec}(W_l) \tag{164}
\end{align}$$

which gives:

$$\begin{align}
\text{vec}(\widetilde{\widetilde{\Delta W_l}}) &= 2\text{vec}(\Delta W_l)_{\text{sym}} - \mathbb{E}_{\boldsymbol{x}}\left[\hat{K}_{l,\text{base}}(\boldsymbol{x})(I - W_{\text{pred}})^\mathsf{T}(I - W_{\text{pred}}) \hat{K}_{l,\text{base}}^\mathsf{T}(\boldsymbol{x})\right] \text{vec}(W_l) \\
&= 2\text{vec}(\Delta W_l)_{\text{sym}} - \mathbb{V}_{\boldsymbol{x}}\left[\bar{K}_{l,\text{base}}(\boldsymbol{x})(I - W_{\text{pred}})^\mathsf{T}\right] \text{vec}(W_l) \tag{165}
\end{align}$$

$\square$

### E.5   Theorem 9

*Proof.* Consider the following discrete dynamics of a weight vector $\boldsymbol{w}(t)$:

$$\boldsymbol{w}(t+1) - \boldsymbol{w}(t) = \alpha \left[ -\boldsymbol{w}(t) + (1-\lambda)\boldsymbol{w}_{\text{ema}}(t) \right] \tag{166}$$

where $\alpha$ is the learning rate, $\boldsymbol{w}_{\text{ema}}(t+1) = \gamma_{\text{ema}}\boldsymbol{w}_{\text{ema}}(t) + (1-\gamma_{\text{ema}})\boldsymbol{w}(t)$ is the exponential moving average of $\boldsymbol{w}(t)$. For convenience, we use $\eta := 1 - \gamma_{\text{ema}}$.

Since it is a recurrence equation, we apply *z-transform* on the temporal domain, where $\boldsymbol{w}(z) := \mathbb{Z}[\boldsymbol{w}(t)] = \sum_{t=0}^{+\infty} \boldsymbol{w}(t) z^{-t}$. This leads to:

$$z(\boldsymbol{w}(z) - \boldsymbol{w}(0)) = \boldsymbol{w}(z) - \alpha \left( \boldsymbol{w}(z) - \mathbb{Z}[\boldsymbol{w}_{\text{ema}}(t)](1-\lambda) \right) \tag{167}$$

Note that for $\boldsymbol{w}_{\text{ema}}(t)$ we have:

$$z(\boldsymbol{w}_{\text{ema}}(z) - \boldsymbol{w}_{\text{ema}}(0)) = (1-\eta)\boldsymbol{w}_{\text{ema}}(z) + \eta\boldsymbol{w}(z) \tag{168}$$

If we set $\boldsymbol{w}_{\text{ema}}(0) = 0$, i.e., the target network is all zero at the beginning, then it gives $\boldsymbol{w}_{\text{ema}}(z) = \frac{\eta}{z-1+\eta}\boldsymbol{w}(z)$. Plugging it back to Eqn. 167 and we have:

$$z(\boldsymbol{w}(z) - \boldsymbol{w}(0)) = \boldsymbol{w}(z) - \alpha\boldsymbol{w}(z)\left( 1 - \frac{\eta}{z-1+\eta}(1-\lambda) \right) \tag{169}$$

And then we could solve $\boldsymbol{w}(z)$:

$$\boldsymbol{w}(z) = \frac{z(z-1+\eta)}{(z-1)^2 + (\eta+\alpha)(z-1) + \alpha\eta\lambda}\boldsymbol{w}(0) \tag{170}$$

Note that the denominator has two roots $z_1$ and $z_2$:

$$z_{1,2} = 1 - \frac{1}{2}\left( \eta+\alpha \pm \sqrt{(\eta+\alpha)^2 - 4\alpha\eta\lambda} \right) \tag{171}$$

and $\boldsymbol{w}(z)$ can be written as

$$\boldsymbol{w}(z) = \frac{z(z-1+\eta)}{(z-z_1)(z-z_2)}\boldsymbol{w}(0) \tag{172}$$

Without loss of generality, let $z_1 < z_2$. The larger root $z_2 > 1$ when $\lambda < 0$, so the zero ($z = 1-\eta = \gamma_{\text{ema}}$) in the nominator won't cancel out the pole at $z_2$. And we have:

$$\frac{z}{(z-z_1)(z-z_2)} = \frac{z}{z_2-z_1} \frac{(z-z_1) - (z-z_2)}{(z-z_1)(z-z_2)} \tag{173}$$

$$= \frac{z}{z_2-z_1}\left( \frac{1}{z-z_2} - \frac{1}{z-z_1} \right) \tag{174}$$

$$= \frac{1}{z_2-z_1}\left( \frac{1}{1-z_2 z^{-1}} - \frac{1}{1-z_1 z^{-1}} \right) \tag{175}$$

where $1/(1-z_2 z^{-1})$ corresponds to a power series $z_2^t$ in the temporal domain. Therefore, we could see $\boldsymbol{w}(t)$ has exponential growth due to $z_2 > 1$.

Now let us check how $z_2$ changes over $\eta$, i.e., how the parameter $\gamma_{\text{ema}} := 1 - \eta$ of EMA affects the learning process. We have:

$$z_2 = 1 + \frac{\eta+\alpha}{2}\left( \sqrt{1 + \frac{4\alpha\eta\lambda}{(\eta+\alpha)^2}} - 1 \right) \tag{176}$$

Use the fact that $(1+x)^{1/2} \le 1 + \frac{1}{2}x$ for $x \ge 0$, we have:

$$z_2 - 1 \le \frac{\eta+\alpha}{4} \frac{4\alpha\eta\lambda}{(\eta+\alpha)^2} = \frac{\lambda}{\frac{1}{\alpha} + \frac{1}{\eta}} \tag{177}$$

Compared to no EMA case (i.e., $\gamma_{\text{ema}} = 0$ or $\eta = 1$), with a $\gamma_{\text{ema}} < 1$ but close to 1 (or equivalently, $\eta$ is close to 0), the upper bound of $z_2$ becomes smaller but still greater than 1, and the exponential growth is less aggressive, which stabilizes the training. Note that if $\gamma_{\text{ema}} = 1$ (or $\eta = 0$), then $\boldsymbol{w}_{\text{ema}}(t) \equiv \boldsymbol{w}_{\text{ema}}(0) = \boldsymbol{0}$ and learning also doesn't happen. $\qquad\square$

## F   EXACT SOLUTIONS TO BYOL WITH LINEAR ARCHITECTURES WITHOUT BATCHNORM

An interesting property of BYOL is that it finds useful non-collapsed solutions for the online network and target network, despite the fact that it does not employ contrastive terms to separate the representations of negative pairs. While BatchNorm can implicitly introduce contrastive terms in BYOL, as discussed in the main paper, recent work (Richemond et al., 2020b) has shown that other normalization methods which do not introduce contrastive terms, nevertheless enable BYOL to work well. We therefore analyze BYOL in a simple linear setting to obtain insight into why it does not lead to collapsed solutions, even without BatchNorm. We first derive exact fixed point solutions to BYOL learning dynamics in this setting, and discuss their stability. We then discuss specific models for data distributions and augmentation procedures, and show how the fixed point solutions of BYOL learning dynamics depend on both data and augmentation distributions. We then discuss how our theory reveals a fundamental role for the predictor in avoiding collapse in BYOL solutions. Finally, we derive a highly reduced three dimensional description of BYOL learning dynamics that provide considerable insights into dynamical mechanisms enabling BYOL to avoid collapsed solutions without negative pairs to force apart representations of different objects.

### F.1   THE FIXED POINT STRUCTURE OF BYOL LEARNING DYNAMICS.

We consider a single linear layer online network with weights $W_1 \in \mathbb{R}^{n_1 \times n_0}$ and a single layer target network with weights $\Theta \in \mathbb{R}^{n_1 \times n_0}$. Additionally, the online network has a predictor layer with weights $W_2 \in \mathbb{R}^{n_1 \times n_1}$, that maps the output of the online network to the output space of the target network. BYOL only uses positive pairs in which a single data point $\boldsymbol{x}$ is drawn from the data distribution $p(\cdot)$, and then two augmented views $\boldsymbol{x}_1$ and $\boldsymbol{x}_2$ are drawn from a conditional augmentation distribution $p_{\mathrm{aug}}(\cdot|\boldsymbol{x})$. The loss function driving the learning dynamics of the online weights $W_1$ and predictor weights $W_2$ given a single positive pair $\{\boldsymbol{x}_1, \boldsymbol{x}_2\}$ and a given target network $\Theta$ is then given by

$$L = \|W_2 W_1 \boldsymbol{x}_1 - \Theta \boldsymbol{x}_2\|_2^2. \tag{178}$$

In contrast, the dynamics of the target network weights $\Theta$ follows that of the online weights $W_1$ through an exponential moving average. In the limit of large batch sizes and slow learning rates, the combined learning dynamics is then well approximated by the continuous time ordinary differential equations (see e.g. Saxe et al. (2014) for analogous equations in the setting of supervised learning):

$$\tau_o \frac{dW_2}{dt} = \left[\Theta \Sigma^d - W_2 W_1 \Sigma^s\right] W_1^T \tag{179}$$

$$\tau_p \frac{dW_1}{dt} = W_2^T \left[\Theta \Sigma^d - W_2 W_1 \Sigma^s\right] \tag{180}$$

$$\tau_t \frac{d\Theta}{dt} = -\Theta + W_1, \tag{181}$$

where

$$\Sigma^s \equiv \mathbb{E}_{\boldsymbol{x}_1} \left[\boldsymbol{x}_1 \boldsymbol{x}_1^T\right] \tag{182}$$

$$\Sigma^d \equiv \mathbb{E}_{\boldsymbol{x}_1, \boldsymbol{x}_2} \left[\boldsymbol{x}_1 \boldsymbol{x}_2^T\right] = \mathbb{E}_{\boldsymbol{x} \sim p(\cdot)} \left[\bar{K}(\boldsymbol{x}) \bar{K}(\boldsymbol{x})^T\right], \tag{183}$$

and

$$\bar{K}(\boldsymbol{x}) \equiv \mathbb{E}_{\boldsymbol{x}_1 \sim p_{\mathrm{aug}}(\cdot|\boldsymbol{x})} \left[\boldsymbol{x}_1\right]. \tag{184}$$

Here $\Sigma^s$ is the correlation matrix of a single augmented view $\boldsymbol{x}_1$ of the data $\boldsymbol{x}$, while $\Sigma^d$ is the correlation matrix between two augmented views of the same data point, or equivalently, the correlation matrix of the augmentation averaged vector $\bar{K}(\boldsymbol{x})$. Additionally, we have retained the possibility of having three different learning rates for the online, predictor, and target networks, represented by the time constants $\tau_o$, $\tau_p$, and $\tau_t$ respectively.

Because of the linearity of the networks, the final outcome of learning depends on the data and augmentation procedures only through the two correlation matrices $\Sigma^s$ and $\Sigma^d$. Examining equation 179-equation 181, we find sufficient conditions for a fixed point given by $W_2 W_1 \Sigma^s = \Theta \Sigma^d$ and $W_1 = \Theta$. Inserting the second equation into the first and right multiplying both sides by $[\Sigma^s]^{-1}$

(assuming $\Sigma^s$ is invertible), yields a manifold of fixed point solutions in $W_1$ and $W_2$ satisfying the nonlinear equation

$$W_2 W_1 = W_1 \Sigma^d [\Sigma^s]^{-1}. \tag{185}$$

This constitutes a set of $n_1 \times n_2$ nonlinear equations in $(n_1 \times n_2) + (n_2 \times n_2)$ unknowns, yielding generically a nonlinear manifold of solutions in $W_1$ and $W_2$ of dimensionality $n_2 \times n_2$ corresponding to the number of predictor parameters. For concreteness, we will assume that $n_2 \leq n_1$, so that the online and target networks perform dimensionality reduction. Then a special class of solutions to equation 185 can be obtained by assuming the $n_2$ rows of $W_1$ correspond to $n_2$ left-eigenvectors of $\Sigma^d [\Sigma^s]^{-1}$ and $W_2$ is a diagonal matrix with the corresponding eigenvalues. This special class of solutions can then be generalized by a transformation $W_2 \to S W_2 S^{-1}$ and $W_1 \to S W_1$ where $S$ is any invertible $n_2$ by $n_2$ matrix. Indeed this transformation is a symmetry of equation 185, which defines the solution manifold.

In addition to these families of solutions, the collapsed solution $W_1 = W_2 = \Theta = 0$ also exists, and a natural question is, why doesn't BYOL generically converge to this collapsed solution? This question can be addressed by analyzing the stability of both the collapsed solution and the families of solutions presented above. The basic calculation involves computing the Jacobian of the vector field defining the dynamics of equation 179 through equation 181. A fixed point solution is stable if and only if all eigenvalues of the Jacobian evaluated at a fixed point solution are negative. Using methods similar to that of Baldi and Hornik (1989), which carried out a similar stability analysis for learning dynamics in two weight layer linear networks in the supervised setting, it is possible to show that all of the above fixed point solutions are unstable *except* for those derived from the special solutions where the $n_2$ rows of $W_1$ correspond to the top $n_2$ principal eigenmodes of $\Sigma^d [\Sigma^s]^{-1}$.

Thus this analysis sketch provides conceptual insights into why BYOL, at least in this simple setting, learns nontrivial, and potentially useful representations with only positive examples, and does not converge to the naive collapsed solution. Basically, the collapsed solution, as well as other subdominant solutions, are unstable, while solutions corresponding to the principal eigenmodes of $\Sigma^d [\Sigma^s]^{-1}$ are stable. Thus, from generic initial conditions, one would expect that the row space of the online network would converge to the span of the top $n_2$ principal eigenmodes of $\Sigma^d [\Sigma^s]^{-1}$.

### F.2    ILLUSTRATIVE MODELS FOR DATA AND DATA AUGMENTATION

While the above section suggests that BYOL converges to the top eigenmodes of $\Sigma^d [\Sigma^s]^{-1}$, here we make this result more concrete by giving illustrative examples of data distributions and data augmentation procedures, and the resulting properties of $\Sigma^d [\Sigma^s]^{-1}$.

**Multiplicative scrambling.**   Consider for example a multiplicative subspace scrambling model, used in the illustration of SIMCLR in Sec. 4.1. In this model, data augmentation scrambles a subspace by multiplying by a random Gaussian matrix, while identically preserving the orthogonal complement of the subspace. In applications, the scrambled subspace could correspond to a space of nuisance features, while the preserved subspace could correspond to semantically important features.

More precisely, we consider a random scrambling operator $A$ which only scrambles data vectors $\boldsymbol{x}$ within a fixed $k$ dimensional subspace spanned by the orthonormal columns of the $n_0 \times k$ matrix $U$. Within this subspace, data vectors are scrambled by a random Gaussian $k \times k$ matrix $B$. Thus $A$ takes the form $A = P^c + U B U^T$ where $P^c = I - U U^T$ is a projection operator onto the $n_0 - k$ dimensional conserved, semantically important, subspace orthogonal to the span of the columns of $U$, and the elements of $B$ are i.i.d. zero mean unit variance Gaussian random variables so that $\mathbb{E}\left[B_{ij} B_{kl}\right] = \delta_{ik}\delta_{jl}$. Under this simple model, the augmentation average $\bar{K}(\boldsymbol{x})$ in equation 184 becomes $\bar{K}(\boldsymbol{x}) = P^c \boldsymbol{x}$. Thus, intuitively, under multiplicative subspace scrambling, the only aspect of a data vector that survives averaging over augmentations is the projection of this data vector onto the preserved subspace. Then the correlation matrix of two different augmented views is $\Sigma^d = P^c \Sigma^x P^c$ while the correlation matrix of two identical views is $\Sigma^s = \Sigma^x$ where $\Sigma^x \equiv \mathbb{E}_{\boldsymbol{x} \sim p(\cdot)}\left[\boldsymbol{x}\boldsymbol{x}^T\right]$ is the correlation matrix of the data distribution. Thus BYOL learns the principal eigenmodes of $\Sigma^d [\Sigma^s]^{-1} = P^c \Sigma^x P_c [\Sigma^x]^{-1}$. In the special case in which $P^c$ commutes with $\Sigma^x$, we have the simple result that $\Sigma^d [\Sigma^s]^{-1} = P^c$, which is completely independent of the data correlation matrix $\Sigma^x$. Thus in this simple setting BYOL learns the subspace of features that are identically conserved under data augmentation, independent of how much data variance there is in the different dimensions of this conserved subspace.

It is interesting to compare to SimCLR in the same setting, which learns the principal eigenmodes of $P^c \Sigma^x P^c$ as described in Sec. 4.1. Thus SimCLR also projects to the conserved subspace, but is further influenced by the correlation matrix of the data within this subspace. In actual applications, which performs better will depend on whether or not features of high variance within the conserved subspace are important for downstream tasks; SimCLR (BYOL) should perform better if conserved features of high variance are (are not) important.

**Additive scrambling.** We also consider, as an illustrative example, data augmentation procedures which simply add Gaussian noise with a prescribed noise covariance matrix $\Sigma^n$. Under this model, we have $\Sigma^s = \Sigma^x + \Sigma^n$ while $\Sigma^d = \Sigma^x$. Thus in this setting, BYOL learns principal eigenmodes of $\Sigma^d [\Sigma^s]^{-1} = \Sigma^x [\Sigma^x + \Sigma^n]^{-1}$. Thus intuitively, dimensions with larger noise variance are attenuated in learned BYOL representations. On the otherhand, correlations in the data that are not attenuated by noise are preferentially learned, but the degree to which they are learned is not strongly influenced by the magnitude of the data correlation (i.e. consider dimensions that lie along small eigenvalues of $\Sigma^n$).

### F.3 THE IMPORTANCE OF THE PREDICTOR IN BYOL.

Here we note that our theory explains why the predictor plays a crucial role in BYOL learning in this simple setting, as is observed empirically in more complex settings. To see this, we can model the removal of the predictor by simply setting $W_2 = I$ in all the above equations. The fixed point solutions then obey $W_1 = W_1 \Sigma^d [\Sigma^s]^{-1}$. This will only have nontrivial, non-collapsed solutions if $\Sigma^d [\Sigma^s]^{-1}$ has eigenvectors with eigenvalue 1. Rows of $W_1$ consisting of linear combinations of these eigenvectors will then constitute solutions.

This constraint of eigenvalue 1 yields a much more restrictive condition on data distributions and augmentation procedures for BYOL to have non-collapsed solutions. It can however be satisfied in multiplicative scrambling if an eigenvector of the data matrix $\Sigma^x$ lies in the column space of the projection operator $P^c$ (in which case it is an eigenvector of eigenvalue 1 of $\Sigma^d [\Sigma^s]^{-1} = P^c \Sigma^x P_c [\Sigma^x]^{-1}$. This condition cannot however be generically satisfied for additive scrambling case, in which generically all the eigenvalues of $\Sigma^d [\Sigma^s]^{-1} = \Sigma^x [\Sigma^x + \Sigma^n]^{-1}$ are less than 1. In this case, without a predictor, it can be checked that the collapsed solution $W_1 = \Theta = 0$ is stable. In contrast, with a predictor, the collapsed solution can be checked to be unstable, and therefore it will not be found from generic initial conditions.

Thus overall, in this simple setting, our theory provides conceptual insight into how the introduction of a predictor is crucial for creating new non-collapsed solutions for BYOL, whose existence destabilizes the collapsed solutions.

### F.4 REDUCTION OF BYOL LEARNING DYNAMICS TO LOW DIMENSIONS

The full learning dynamics in equation 179 to equation 181 constitutes a set of high dimensional nonlinear ODEs which are difficult to solve from arbitrary initial conditions. However, there is a special class of *decoupled* initial conditions which permits additional insight. Consider the special case in which $\Sigma^s$ and $\Sigma^d$ commute, and so are simultaneously diagonalizable and share a common set of eigenvectors, which we denote by $\boldsymbol{u}^\alpha \in \mathbb{R}^{n_0}$. Consider also a special set of initial conditions where each row of $W_1$ and the corresponding row of $\Theta$ are both proportional to one of the eigenmodes $\boldsymbol{u}^\alpha$, with scalar proportionality constants $w_1^\alpha$ and $\theta^\alpha$ respectively, and $W_2$ is diagonal, with the corresponding diagonal element given by $w_2^\alpha$. Then it is straightforward to see that under the dynamics in equation 179 to equation 181, that the structure of this initial condition will remain the same, with only the scalars $w_1^\alpha$, $\theta^\alpha$ and $w_2^\alpha$ changing over time. Moreover, the scalars decouple across the different indices $\alpha$, and the dynamics are driven by the eigenvalues $\lambda_s^\alpha$ and $\lambda_d^\alpha$ of $\Sigma_s$ and $\Sigma_d$ respectively. Inserting this special class of initial conditions into the dynamics in equation 179 to equation 181, and dropping the $\alpha$ index, we find the dynamics of the triplet of scalars is given by

$$\tau_o \frac{dw_2}{dt} = [\theta \lambda_d - w_2 w_1 \lambda_s] w_1 \tag{186}$$

$$\tau_p \frac{dw_1}{dt} = w_2 [\theta \lambda_d - w_2 w_1 \lambda_s] \tag{187}$$

$$\tau_t \frac{d\theta}{dt} = -\theta + w_1. \tag{188}$$

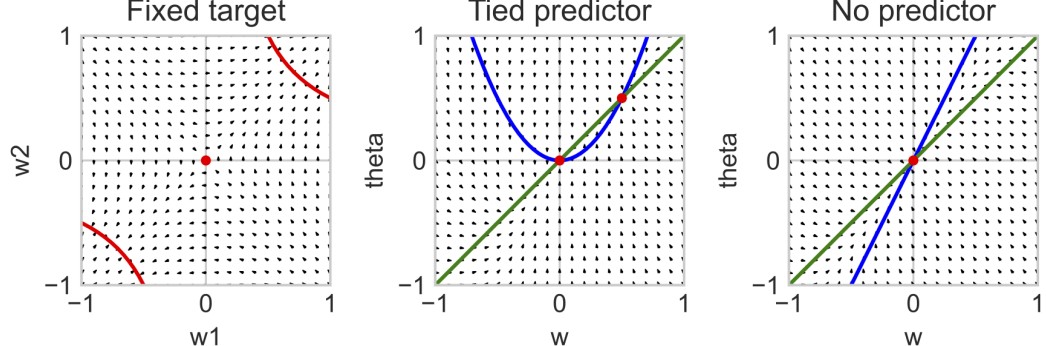

Figure 5: A visualization of BYOL dynamics in low dimensions. **Left**: Black arrows denote the vector field of the flow in the $w_1$ and $w_2$ of plane online and predictor weights in Eqns. 186 and 187 when the target network weight $\theta$ is fixed to 1. For all 3 panels, $\lambda_s = 1$, $\lambda_d = 1/2$, and $\tau_o = \tau_p = \tau_t = 1$, and all vectors are normalized to unit length to indicate direction of flow alone. The red curve shows the hyperobolic manifold of stable fixed points $w_2 w_1 = \theta \lambda_d \lambda_s^{-1}$, while the red point at the origin is an unstable fixed point. For a fixed target network, the online and predictor weights will cooperatively amplify each other to escape the collapsed solution at the origin. **Middle**: A visualization of the full low dimensional BYOL dynamics in Eqns 186-188 when the online and predictor weights are tied so that $w_1 = w_2 = w$. The green curve shows the nullcline $\theta = w$ corresponding to $\frac{d\theta}{dt} = 0$ and the blue curve shows part of the nullcline $\frac{dw}{dt} = 0$ corresponding to $w^2 = \theta \lambda_d \lambda_s^{-1}$. The intersection of these two nullclines yields two fixed points (red dots): an unstable collapsed solution at the origin $w = \theta = 0$, and a stable nontrivial solution with $\theta = w$ and $w = \lambda_d \lambda_s^{-1}$. **Right**: A visualization of dynamics in Eqns 186-188 when the the predictor is removed, so that $w_2$ is fixed to 1. The resulting two dimensional flow field on $w = w_1$ and $\theta$ is shown (black arrows). The green curve shows the nullcline $w = \theta$ corresponding to $\frac{d\theta}{dt} = 0$, while the blue curve shows the nullcline $w = \theta \lambda_d \lambda_s^{-1}$. The slope of this nullcline is $\lambda_s \lambda_d^{-1} > 1$. The resulting nullcline structure yields a single fixed point at the origin which is stable. Thus there only exists a collapsed solution. In the special case where $\lambda_s \lambda_d^{-1} = 1$, the two nullclines coincide, yielding a one dimensional manifold of solutions.

Alternatively, this low dimensional dynamics can be obtained from equation 179 to equation 181 not only by considering a special class of decoupled initial conditions, but also by considering the special case where every matrix is simply a 1 by 1 matrix, making the scalar replacements $W_1 \to w_1$, $W_2 \to w_2$, $\Theta \to \theta$, $\Sigma^s \to \lambda_s$, and $\Sigma^d \to \lambda_d$.

The fixed point conditions of this dynamics are given by $\theta = w_1$ and $w_2 w_1 = \theta \lambda_d \lambda_s^{-1}$. Thus the collapsed point $w_1 = w_2 = \theta = 0$ is a solution. Additionally $w_2 = \lambda_d \lambda_s^{-1}$ and $w_1 = \theta$ taking any value is also a family of non-collapsed solutions. We can understand the three dimensional dynamics intuitively as follows when $\tau_t \gg \tau_o$ and $\tau_o = \tau_p$. In this case, the target network evolves very slowly compared to the online network, as is done in practice, and for simplicity we use the same learning rate for the predictor as we do for the online network. In this situation, we can treat $\theta$ as approximately constant on the fast time scale of $\tau_o$ on which the online and predictor weights $w_1$ and $w_2$ evolve. Then the joint dynamics in equation 186 and equation 187 obeys gradient descent on the error function

$$E = \frac{\lambda_s}{2}(\theta \lambda_d \lambda_s^{-1} - w_2 w_1)^2. \tag{189}$$

Iso-contours of constant error are hyperbolas in the $w_1$ by $w_2$ plane, and for fixed $\theta$, the origin $w_1 = w_2 = 0$ is a saddle point, yielding an unstable fixed point (see Fig. 5 (left)). From generic initial conditions, $w_1$ and $w_2$ will then cooperatively amplify each other to rapidly escape the collapsed solution at the origin, and approach the zero error hyperbolic contour $w_2 w_1 = \theta \lambda_d \lambda_s^{-1}$ where $\theta$ is close to its initial value. Then the slower target network $\theta$ will adjust, slowly moving this contour until $\theta = w_1$. The more rapid dynamics of $w_1$ and $w_2$ will hug the moving contour $w_2 w_1 = \theta \lambda_d \lambda_s^{-1}$ as $\theta$ slowly adjusts. In this fashion, the joint fast dynamics of $w_1$ and $w_2$, combined with the slow dynamics of $\theta$, lead to a nonzero fixed point for all 3 values, despite the existence of a collapsed fixed point at the origin. Moreover, the larger the ratio $\lambda_d \lambda_s^{-1}$, which is determined by the data, the larger the final values of both $w_1$ and $w_2$ will tend to be.

We can obtain further insight by noting that the submanifold $w_1 = w_2$, in which the online and predictor weights are tied, constitutes an invariant submanifold of the dynamics in Eqns. 186 to 188; if $w_1 = w_2$ at any instant of time, then this condition holds for all future time. Therefore we

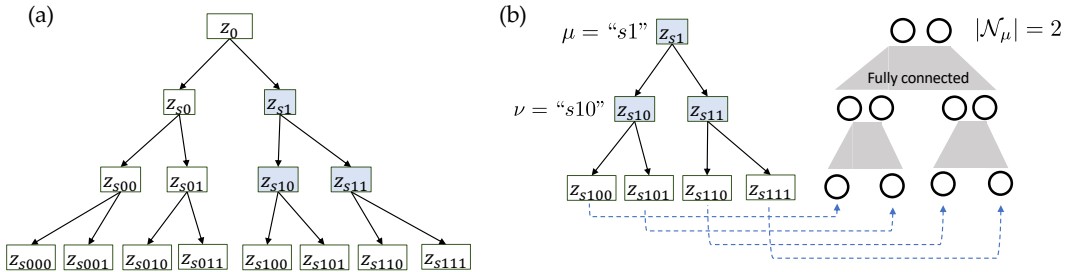

Figure 6: The Hierarchical Latent Tree Model (HLTM) used in our experiments (Sec. G.2 and Sec. 6).

can both analyze and visualize the dynamics on this two dimensional invariant submanifold, with coordinates $w = w_1 = w_2$ and $\theta$ (Fig. 5 (middle)). This analysis clearly shows an unstable collapsed solution at the origin, with $w = \theta = 0$, and a stable non-collapsed solution at $w = \theta = \lambda_d \lambda_s^{-1}$.

We note again, that the generic existence of these non-collapsed solutions in Fig. 5 depends critically on the presence of a predictor with adjustable weights $w_2$. Removing the predictor corresponds to forcing $w_2 = 1$, and non-collapsed solutions cannot exist unless $\lambda_d = \lambda_s$, as demonstrated in Fig. 5 (right). Thus, remarkably, in BYOL in this simple setting, the introduction of a predictor network plays a crucial role, even though it neither adds to the expressive capacity of the online network, nor improves its ability to match the target network. Instead, it plays a crucial role by dramatically modifying the learning dynamics (compare e.g. Fig 5 middle and right panels), thereby enabling convergence to noncollapsed solutions through a dynamical mechanism whereby the online and predictor network cooperatively amplify each others' weights to escape collapsed solutions ( Fig. 5 (left)).

Overall, this analysis of BYOL learning dynamics provides considerable insight into the dynamical mechanisms enabling BYOL to avoid collapsed solutions, without negative pairs to force apart representations, in what is likely to be the simplest nontrivial setting.

## G   Additional Experiments

### G.1   Experiments on Two-layer network

We conduct experiments to verify our theoretical reasoning in Sec. 4.2. We follow the setting in Theorem 5, i.e., two-layer ReLU network ($L = 2$) that contains the same number of hidden nodes as the number of output nodes ($n_1 = n_2$). The top-layer weight $W_2$ is a diagonal matrix with no bias (note that "no bias" is important here, otherwise we won't have $\boldsymbol{w}_{2,j}^\top \boldsymbol{w}_{2,k} = 0$ for $j \neq k$).

We use $L_{\mathrm{nce}}^\tau$ ($\tau = 0.05$ and $H = 1$) and $L_{\mathrm{simp}}$ and use SGD optimizer with learning rate of 0.01. All experiments run with 5000 minibatches with batchsize 128. We test cases with and without $\ell_2$ normalization at the output layer. For each setting, i.e., (loss, normalization), we run 30 random seeds. The data are generated by a mixture of 10 Gaussian distributions (with uniform prior on each mixture), with mean $\mu_k \sim \mathcal{N}(0, I)$ and a covariance matrix $\Sigma_k = 0.1I$. We set the first cluster to be zero-mean.

**Without $\ell_2$ normalization**. Fig. 7 shows the weight growth of the top ($W_2$) and bottom ($W_1$) layer. As predicted by Theorem 5, the weight quickly grows to infinity. Note that the $y$-axis is log scale so exponential growth is shown as linear. From the figure, it is clear that with $L_{\mathrm{simp}}$, their growth is super exponential due to the inter-plays between the top and the bottom layers. On the other hand, with $L_{\mathrm{nce}}$, the growth slows down due to the fact that its associated covariance operator has a weight which decays exponentially when the representations of two distinct samples become far apart.

**With $\ell_2$ normalization**. With the normalization, the weights will not grow substantially and we focus ourselves more on the meaning of each intermediate nodes after training. From the theoretical reasoning in Sec. 4.2, in the ReLU case, we should see each node gradually moves towards (or *specializes* to) one cluster after training. Fig. 8 shows that a node $j$ that is only active for 1 or 2 cluster centers (out of 10) have much higher $|w_{2,j}|$ than some other node that is active on many cluster centers. This shows that those "specialized" nodes has undergone learning and their fan-out weight magnitude $|w_{2,j}|$ becomes (or remains) high, according to the dynamics in Theorem 5.

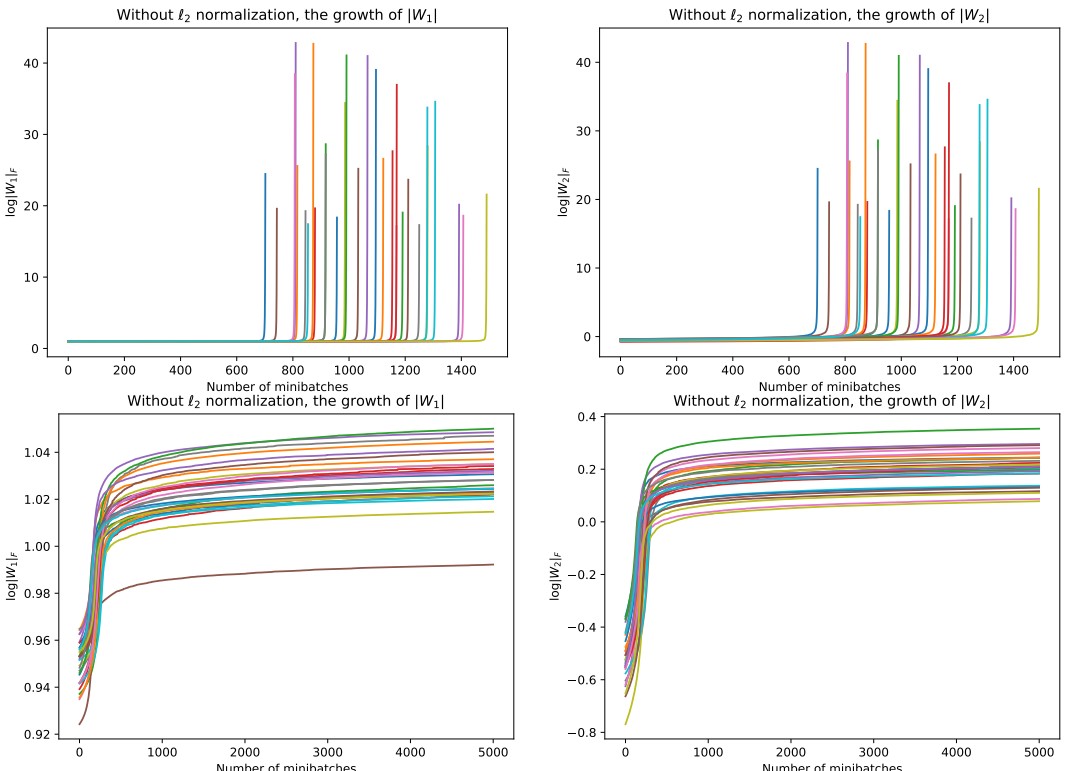

Figure 7: **Top row**: Without $\ell_2$ normalization, training with SimCLR and $L_{\mathrm{simp}}$ leads to fast growth of the weight magnitude over time (each curve is one training curve out of 30 trials with different random seeds). Furthermore, this growth is super exponential due to the interplay between top and bottom layers, as suggested by the dynamics in Eqn. 8. Note that the $y$-axis is in log scale. **Bottom row**: Without $\ell_2$ normalization, $L_{\mathrm{nce}}$ has more stable weight magnitude over training since its covariance operator is weighted (See Theorem 4).

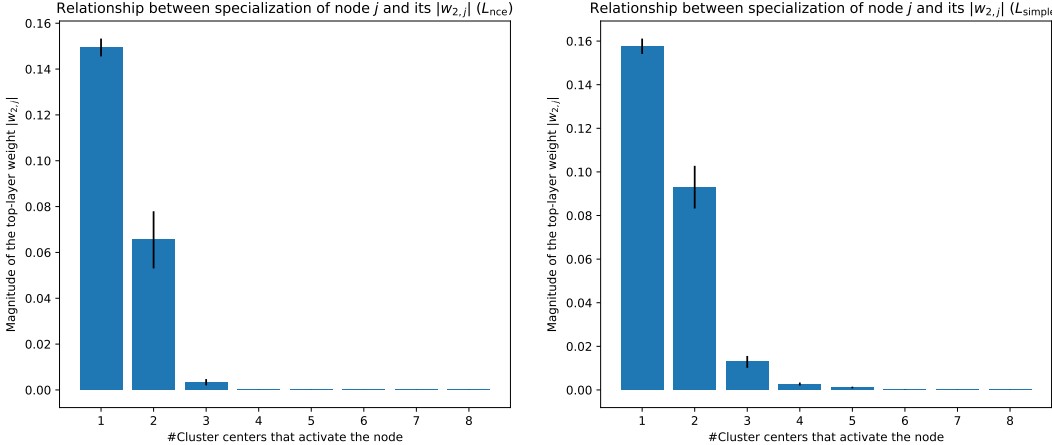

Figure 8: When $|w_{2,j}|$ is high, the corresponding node $j$ is highly selective to one specific cluster of the data generative models. On the other hand, those node $j$ with low selectivity has very small $w_{2,j}$ and does not contribute substantially to the output of the network. Training with $L_{\mathrm{nce}}^{\tau}$ (**Left Plot**) seems to yield stronger selectivity than with $L_{\mathrm{simp}}$ (**Right Plot**).

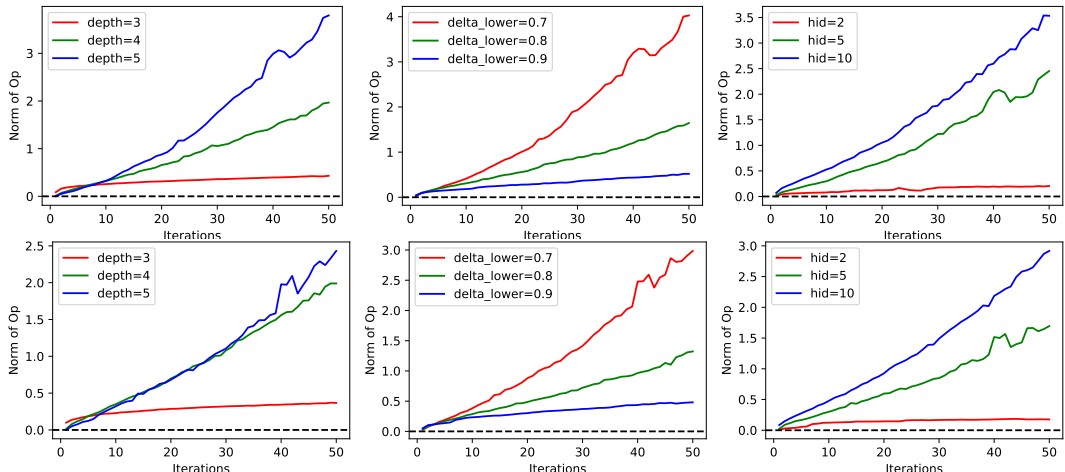

Figure 9: Ablation of how the Frobenius norm of the covariance operator OP changes over training, under different factors: depth $L$, sample range of $\rho_{\mu\nu}$ ($\rho_{\mu\nu} \sim \mathrm{Uniform}[\mathrm{delta\_lower}, 1]$) and over-parameterization $|\mathcal{N}_\mu|$. **Top row**: covariance operator of immediate left latent variable of the root node $z_0$; **Bottom row**: covariance operator of immediate right child of the root node $z_0$.

## G.2 HLTM

We also check how the norm of the covariance operator (OP) changes during training in different situations. For this set of experiments, we construct a complete binary tree of depth $L$. The class/sample-specific latent $z_0$ is at the root, while other nuisance latent variables are labeled with a binary encoding (e.g., $\mu = $ "s010" for a $z_\mu$ that is the left-right-left child of the root $z_0$). Please check Fig. 6 for details.

Again we use SimCLR with the $L_{\mathrm{nce}}^\tau$ loss ($\tau = 0.1$ and $H = 1$) to train the model. $\ell_2$ normalization is used in the output layer. The results are shown in Fig. 9 and Fig. 10. We could see that norm of the covariance operator indeed go up, showing that it gets amplified during training. We perform a grid search of depth = $[3, 4, 5]$, delta_lower = $[0.7, 0.8, 0.9]$ (and the polarity $\rho_{\mu\nu} \sim \mathrm{Uniform}[\mathrm{delta\_lower}, 1]$ at each layer) and over-parameterization parameter hid = $|\mathcal{N}_\mu| = [2, 5, 10]$. For each experiment configuration, we run 30 random seeds.

## G.3 BYOL EXPERIMENTS SETUP

For all STL-10 task, we use ResNet18 as $\mathcal{W}_{\mathrm{base}}$. The extra predictor is two layer. It takes 128 dimensional input, has a hidden layer of size 512, and its output is also 128 dimensional. We use ReLU in-between the two layers. When we add BN to the predictor, we add it before ReLU activation. When we say there is no BN in Tbl. 3, we remove BN in both predictor and projector layer (but not in the encoder). Same as BYOL paper (Grill et al., 2020), symmetric loss function is used with $\ell_2$ normalization in the topmost layer.

We use simple SGD optimizer. The learning rate is $0.03$ (unless otherwise stated), momentum is $0.9$ and weight decay is $0.0004$. The training batchsize is $128$. The ImageNet experiment runs on 32 GPUs with a batchsize $4096$.

## G.4 ADDITIONAL BYOL EXPERIMENTS

Based on our theoretical analysis, we try training the predictor in different ways and check whether it still works.

From the analysis in Sec. 5, we know that the reason why BYOL works is due to the dominance of $\mathrm{Cov}_{\boldsymbol{x}}[\bar{K}_l(\boldsymbol{x}), \bar{K}_l(\boldsymbol{x}; \mathcal{W})]$ and its resemblance of the covariance operator $\mathbb{V}_{\boldsymbol{x}}[\bar{K}_l(\boldsymbol{x})]$, which is a PSD matrix.

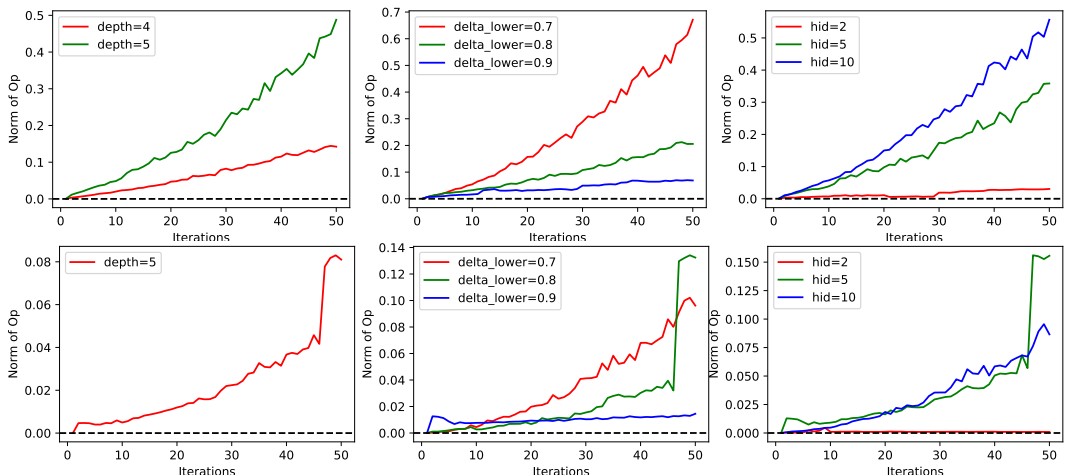

Figure 10: Ablation of how the Frobenius norm of the covariance operator OP changes over training. Same setting as Fig. 9 but focus on lower level. Note that since we have used $\ell_2$ normalization at the topmost layer, the growth of the covariance operator is likely not due to the growth of the weight magnitudes, but due to more discriminative representations of the input features $\boldsymbol{f}_\mu$ with respect to different $z_0$. **Top row**: covariance operator of left-right-left latent variable from the root node $z_0$; **Bottom row**: covariance operator right-left-right-left latent variable from the root node $z_0$.

The dominance should be stronger if the predictor has smaller weights than normally initialized using Xavier/Kaiming initialization. Also, $\mathrm{Cov}_{\boldsymbol{x}}[\bar{K}_l(\boldsymbol{x}), \bar{K}_l(\boldsymbol{x}; \mathcal{W}')]$ should behave more like a PSD matrix, if the predictor's weights are all small positive numbers and no BN is used.

The following table justifies our theoretical findings. In particular, Tbl. 10 shows better performance in STL-10 with smaller learning rate and smaller sample range of the predictor weights.

Table 7: Training one-layer predictor with positive initial weights and no EMA ($\gamma_{\mathrm{ema}} = 0$). All experiments run for 3 seeds.

| Sample range of predictor weight | $[0, 0.01]$ | $[0, 0.02]$ | $[0, 0.05]$ |
|---|---|---|---|
| With BN in predictor | $62.78 \pm 1.40$ | $62.94 \pm 1.03$ | $62.31 \pm 1.80$ |
| Without BN in predictor | $71.95 \pm 0.27$ | $72.06 \pm 0.44$ | $71.91 \pm 0.59$ |

Table 8: Training one-layer predictor with positive initial weights with EMA ($\gamma_{\mathrm{ema}} = 0.996$) and predictor resetting every $T = 10$ epochs. All experiments run for 3 seeds. Note that Xavier range is $\mathrm{Uniform}[-0.15, 0.15]$ and our initialization range is much smaller than that.

| Sample range of predictor weight | $[0, 0.003]$ | $[0, 0.005]$ | $[0, 0.007]$ |
|---|---|---|---|
| With BN in predictor | $65.61 \pm 1.34$ | $70.56 \pm 0.57$ | $70.87 \pm 1.51$ |
| Without BN in predictor | $74.39 \pm 0.67$ | $74.52 \pm 0.63$ | $74.80 \pm 0.57$ |

Table 9: Same as Tbl. 8 but with different weight range. All experiments run for 3 seeds.

| Sample range of predictor weight | $[0, 0.01]$ | $[0, 0.02]$ | $[0, 0.05]$ |
|---|---|---|---|
| With BN in predictor | $68.98 \pm 2.34$ | $66.56 \pm 1.70$ | $68.41 \pm 1.19$ |
| Without BN in predictor | $74.66 \pm 0.81$ | $73.60 \pm 0.32$ | $74.34 \pm 0.77$ |

Table 10: Top-1 Performance on STL-10 with a two-layer predictor with BN and EMA ($\gamma_{\mathrm{ema}} = 0.996$). Learning rate is smaller (0.02) and predictor weight sampled from $\mathrm{Uniform}[-range, range]$. Note that for this, Xavier range is $\mathrm{Uniform}[-0.097, 0.097]$ and our range is smaller.

| Weight range | 0.01 | 0.02 | 0.03 | 0.05 |
|---|---|---|---|---|
| $T = 3$ | $79.48 \pm 0.40$ | $79.70 \pm 0.47$ | $79.66 \pm 0.37$ | $78.63 \pm 0.10$ |
| $T = 5$ | $78.97 \pm 0.62$ | $79.63 \pm 0.23$ | $79.65 \pm 0.37$ | $79.01 \pm 0.27$ |
| $T = 10$ | $79.25 \pm 0.20$ | $79.63 \pm 0.22$ | $79.58 \pm 0.25$ | $79.18 \pm 0.22$ |
| $T = 20$ | $79.15 \pm 0.66$ | $79.91 \pm 0.10$ | $79.78 \pm 0.05$ | $79.54 \pm 0.25$ |

