# OpenReview forum: "Understanding Self-supervised Learning with Dual Deep Networks"
_ICLR.cc/2021/Conference — Reject_

### Official Review · AnonReviewer2 · 2020-10-22
**Interesting theoretical analysis of self-supervised learning**

**Rating:** 6
**Confidence:** 4

**Review:**

This paper provided in-depth theoretical analysis of self-supervised learning, focusing on two network structures (SimCLR, BYOL). They proved that a covariance operator plays an important role in the SGD learning of self-supervised algorithms. Simulation experiments were provided to justify the theoretical findings. The paper is well written, with rigorous mathematical derivations.

Here are a few comments on the technical details:
1) In Theorem 3 there is an assumption that the derivative of L with respect to rk- is a constant. Is this a reasonable assumption in real situations? Can we see how this assumption is justified (or violated) in real data?

2) Below equation (37) in the appendix, the above assumption becomes derivative of L with respect to rk-^2, and H is replaced with n in the constant. It seems that there is some discrepancy with the statement of Theorem 3, and some explanation may be helpful.

3) In the first paragraph of Section 4, it is stated that "all nuisance latents z' are integrated out in K(z_0)". This statement seems to be a little vague to me. More rigorous definitions and derivations may be needed here to support the claim.

4) A related follow-up question is, how would the augmentation distribution impact the analysis? What would be a good augmentation transform?

---

> ### Author Response · Authors · 2020-11-14
> **Author Response to R2**
>
> We thank R2 for reviewing our paper, and appreciate the positive feedback. We address common questions above and and address specific questions raised here:
>
> **Q1**: Below equation (37) in the appendix, the above assumption becomes derivative of L with respect to rk-^2, and H is replaced with n in the constant. It seems that there is some discrepancy with the statement of Theorem 3, and some explanation may be helpful.
>
> **A1**: Thanks for pointing that out. There is a typo in the proof. The letter $n$ should be just $H$.
>
> **Q2**: In the first paragraph of Section 4, it is stated that "all nuisance latents z' are integrated out in K(z_0)". This statement seems to be a little vague to me. More rigorous definitions and derivations may be needed here to support the claim.
>
> **A2**: In Section 4, the input x is a function of two kinds of latent variables $z_0$ and $z’$. So we can write $K = K(x) = K(z_0, z’)$. Therefore, we can write
>
> $$\bar K(z_0) = E_{z'|z_0}[K(z_0,z')] = \int_{z’} K(z_0, z’) p(z’|z_0) \mathrm{d} z’$$
>
> This is what we mean by “integrated out”. We will make it more clear in the next revision.
>
> **Q3**: How would the augmentation distribution impact the analysis? What would be a good augmentation transform?
>
> **A3**: Section 4 gives a few examples on how different generation processes lead to different covariance operators. We leave a comparable study of different data augmentations using this framework for future work.  But the basic intuition that already arises from our analysis is that data augmentation should scramble or randomize irrelevant / nuisance features, while preserving semantically important latent features that may discriminate between different classes in downstream classification tasks.

---

### Official Review · AnonReviewer1 · 2020-10-29
**A very solid work**

**Rating:** 8
**Confidence:** 2

**Review:**

Summary: This paper does theoretical analysis about self supervised learning (SSL), esp. those methods using contrastive learning. It zooms into one-layer and two-layer networks and proves contrastive learning can converge to weights corresponding to largest eigenvector of a covariance matrix. Experiments on synthetic and real datasets are consistent with theoretical conclusions.

Reasons for score:
The covariance operator sheds light to the black-box learning process of contrastive learning. Approximating the data distribution with a hierarchical latent tree model is an interesting technique. This work may inspire practical tools to improve SSL.

Issues:
1. My major concern is the analysis about BYOL. The authors of BYOL disagree with the conclusion in this paper that BatchNorm provides implicit contrastive objective for BYOL, in a newly released paper "BYOL works even without batch statistics". I'd ask the authors, does this invalidate your analysis about BYOL? Or does the GroupNorm + weight standardization also provide implicit contrastive objective, similar as BN?
2. To what degree the validity of the theoretical analysis depends on the type of the loss? Say if I change the InfoNCE loss to some other loss, will that invalidate the whole analysis?
3. A typo in page 4: "we setup the following..." => "we set up the following..."

---

> ### Author Response · Authors · 2020-11-14
> **Author Response to R1**
>
> We thank R1 for reviewing our paper and we appreciate the positive enthusiasm.
>
> We address the three main points raised here:
>
> **A1**: Please check the "Common Question" section above. We do not believe we are in contradiction with the authors of BYOL, and we will edit the paper accordingly to clarify.  In short, our main result is that BYOL behaves like SimCLR when BatchNorm is introduced in that a positive definite covariance operator governs the learning dynamics. Thus BYOL with BN should succeed whenever SimCLR succeeds. We do not wish to claim this is a necessary condition and we will edit this section to reflect that, thereby being consistent with the empirics that BYOL can succeed without BN.
>
> **A2**: Please see the answer to this question in the “Common Questions” section.
>
> **A3**: Thank you - we will fix that typo.

---

### Official Review · AnonReviewer4 · 2020-10-30
**The paper proposes a theoretical framework to understand self-supervised learning methods that employ dual pairs of deep ReLU networks. The finding is interesting and novel. More experiments are needed.**

**Rating:** 5
**Confidence:** 2

**Review:**

The paper proposes a novel theoretical framework to understand self-supervised learning
methods that employ dual pairs of deep ReLU networks. The finding is interesting and novel. The experiments support the conclusion. However, there are several issues.

The paper should explain the motivation to choose simCLR for analysis.
The paper defines the covariance operator from contrast loss and the paper declare that the theory is suitable for deep ReLU networks with dual pairs. How to define the covariance operator for other kinds of loss function of deep ReLU networks with dual pairs, such as triplet loss.
The experiment setting should be listed. Such as the learning rate, the training time and the date augmentation used in the experiments. The paper also needs experiments about the magnitude of the covariance operator and the training time.
The paper analysis the activation gap and the weights of different layer for HLTM. The paper needs experiments about the activation gap grows over time and the weight of the top layer and lower layer change over time.

---

> ### Author Response · Authors · 2020-11-14
> **Author Response to R4**
>
> We thank R4 for the review. We address common questions above and and address specific questions raised here:
>
> **Q1**: The paper should explain the motivation to choose simCLR for analysis.
>
> **A1**: We choose SimCLR because it achieves close to SoTA performance in SSL, has attracted a great deal of attention in the field, and it has a very clean architecture and fits well with the existing teacher-student theoretical framework. We will edit the paper to clarify this motivation.
>
> **Q2**: The experiment setting should be listed. Such as the learning rate, the training time and the date augmentation used in the experiments.
>
> **A2**: We will list all of these experimental details in the next revision and will open source the code used for experiments.  Thank you for pointing this out.
>
> **Q3**: The paper also needs experiments about the magnitude of the covariance operator and the training time.
>
> **A3**: We will put the relevant experiments in the next revision. In HLTM, we actually compute the covariance operator explicitly and see its norm indeed goes up during training. Moreover, as predicted by the theory, the top-level covariance operator goes up faster and drives the low-level training.
>
> **Q4**: The paper analyzes the activation gap and the weights of different layer for HLTM. The paper needs experiments about the activation gap growing over time and the weight of the top layer and lower layer changing over time.
>
> **A4**: The activation gap indeed grows over time, as demonstrated indirectly in Table 2 where the normalized correlation between the activation of a node and the ground truth latent variable goes up. We will add top-layer/lower-layer experiments in the next revision (we assume R4 refers to the two layer case in Sec. 4.2).

---

### Official Review · AnonReviewer3 · 2020-10-30
**This paper presented a novel theoretical framework to analyze some state-of-the-art self-supervised learning methods.  The authors proved that the weights are updated by a covariance operator that amplifies initial random selectivities that vary across data samples during training and explain how the good representations can be learned in this way. And the authors also analyze the reason why BYOL can work with no negative samples.**

**Rating:** 7
**Confidence:** 4

**Review:**

Strengths:
	This paper tried to analyze the “Black-box” of the contrastive learning by inspecting the weight update during contrastive learning. The analysis and conclusion seem promising.
	This paper made a theoretically analysis on BYOL and conducted a variety of ablation studies to study the impacts of BN layer and predictor in BYOL. And some interesting experimental results and conclusions are presented, which sheds light on the further explorations on self-supervised learning.
Weakness:
While I still have some questions:
	In [1], it has been proved that BYOL can work without BN in predictor, which seems contradictory the conclusion given in this paper. I recommend the author to compare with this conclusion.
	In Table 5, it seems that reinitializing the predictor will improve the performance a bit. I wonder if the improvements will gradually decrease or even vanish for longer training epochs.

[1] BYOL works even without batch statistics.

---

> ### Author Response · Authors · 2020-11-14
> **Author Response to R3**
>
> We thank R3 for the review and appreciate the positive response.
>
> R3 raises two main questions and we answer them below:
>
> **Q1**: it has been proved that BYOL can work without BN in predictor, which seems contradictory the conclusion given in this paper. I recommend the author to compare with this conclusion.
>
> **A1**: Our theoretical analysis that BYOL has implicit contrastive terms when one includes BatchNorm does not contradict the result of the cited reference [1] of R3. We will edit the relevant section to make this clear.  Please see also a detailed response to this point in the “Common questions” section of the rebuttal.
>
> **Q2**: In Table 5, it seems that reinitializing the predictor will improve the performance a bit. I wonder if the improvements will gradually decrease or even vanish for longer training epochs.”
>
> **A2**: Yes. These performance improvement will gradually decrease/vanish if the training lasts for longer epochs. That’s why we don’t claim that we have a better algorithm than BYOL. Here the the main point is that restarting the predictor during training, doesn’t seem to hurt the performance. We will clarify this further in the next revised version.

---

### Official Review · AnonReviewer5 · 2020-11-04
**The observations can be interesting, but they are not rigorous theoretical results, while the presentation is disastrous.**

**Rating:** 3
**Confidence:** 4

**Review:**

This paper aimed to understand the self-supervised learning algorithm under a teacher-student network setting. The authors argued that the gradient are specified by a covariance operator, that can amplify extracting the intrinsic features that are invariant to the data augmentation.

I have several concerns on this paper, mainly on the over-simplified setting, unrealistic assumption, and inaccurate claim. Also, the presentation is weird.

The detailed comments:
1. I feel most of the theorem are just simple calculus. It’s inappropriate to claim such calculation results as theorem. At least one theorem should contain some information on the property of the terms of interest.
2. As far as I know, there’s no existing empirical work used the ell_2 loss on the feature as the dissimilarity, though when the features have norm 1 (which cannot simply hold in practice) there are some connections between the ell_2 loss and inner product. I personally would argue that, the authors should first make an empirical justification on this simplification, say such dissimilarity measure can empirically work. I don’t think this is a valid simplification. The authors may argue that methods like simCLR have normalization before computing pairwise similarity, but this will dramatically influence the gradient update. Maybe the remaining conclusion still holds, but at least the authors should include such part.
3. I would like to say that the assumption for Theorem 3, i.e. the gradient of the negative samples are identical, are too strong and unrealistic. Definitely this will not happen, even though the negative samples are from the same distribution, there should be some variance. I cannot accept such assumption. Even take H=1 is better than this assumption.
4. Beta is used without any introduction.
5. As the authors say, the covariance operator are changing over time, it’s hard to claim that this covariance operator let the parameter align with some specific direction that is good to learn a good representation. Take the NTK limit may be one possible way to get some more interested result, however, the authors haven’t done that.
6. And I would argue that, with only analysis on gradient, it’s rather hard to make some strong claim. There are several existing work on homogenous neural network that characterize the optimization and solution with e.g. separable data, which is much more convincing.
7. I feel the description starting from Section 4 on two groups of latent variables are so intuitive and also unrealistic. How can we identify such groups of latent variables and how can we make sure in practice we don’t perturb the class/sample-specific latent? On the other hand, what if we perturb the class/sample-specific latent? Imagine we have a Gaussian mixture of two isotropic components, if we augment the data with the isotropic Gaussian, can we make an informative projection onto low dimension with self-supervised learning? I think the current intuitive and unrealistic setting restricted the potential application in practice.
8. What’s the definition of bar{K}_l(z_0) as the input of this term should be x? What’s the form of the term after integrated out z^\prime? I feel there are so many ambiguities here and I’m afraid I cannot accept such claim.
9. I think the arguments in Section 4.1 need to be justified more formally, if this is a `theoretical’ paper. I wonder what does the author want to say in Section 4.1, the importance of non-linear activation?
10. In Section 4.2, why constrained to 1 output? I would like to ask, as in the derivation in Section 3, there should be no straightforward barrier on considering this more general case?
11. In Theorem 4, is the A matrix fixed during the dynamics? I’m quite skeptical on that, as the w in the definition of u are changed over time, which can influence the indicator. Also, why we have the covariance is equal to zero? Is that the normal case? Also, there are so many ambiguities here, see 9, can the authors give all the formal description of the neural network we considered, the input data distribution, etc. in a clear way? To be honest, I cannot understand the proof of Theorem 4 as well, due to these ambiguities.
12. For the description in the last of Page 7, I totally get confused. What does the authors want to say on that? Even the parameter converge to some point, we cannot say during the optimization it keeps the indicator 1? Can the authors stop using such ambiguous description and give some formal description and claim?
13. What’s the meaning of considering such HLTM? Can it represent the general case of learning? I need to say the authors does not convince me that such setting is general and it’s necessary to consider the multi-layer network with such setting. I would like to say, it’s better first considering the fully-connect network rather than considering the network with local receptive field.
14. I would like to say the organization of Section 4.3 is really weird. Can the authors give a formal description in the neural network and generative model in Section 4.3, even in the appendix with simpler but clear model? I don’t even know the data generating process of x given z, thus does not the meaning of given a sample x how to resampling z^\prime. As the notation are only described in Table 1, I cannot get the meaning of each term in the theorem. Give some description on each of term with a formalized mathematical description on the generative model and network, please.
15. Also, I does not feel the theorem in Section 4.3 give some strong arguments. What we really care in the self-supervised learning is the quality of representation, the sample complexity, the convergence analysis, none of them have been addressed by the author formally. Instead, I feel Theorem 6 presented here is not of particular interest. Also, the discussion is quite intuitive.
16. What’s the sym subscript in Equation 10? Please introduce the notation before used it. Also, what’s the term deltaW_l^{BN} in Equation 12? Please be self-contained.
17. The observation of Theorem 7 can be interesting, however, in practice, what we really do during BN is the latter one, i.e. x - x.mean().detach()? And in fact, the mean we subtract in BN is a moving average statistics. I feel it can explain something, but not why BYOL works. The claim that such observation give an analysis of why BYOL work is not accurate. In the contrast, I think it even predict that BYOL will not work or at least need to be fixed.
18. Overall, what the authors have proposed are better understood as some intuition or observation, not some rigorous theory.

To sum up, I feel the presentation is weird. The settings are over-simplified and the authors even didn’t introduce the setting with formal description. The assumption can be too strong while some explanations are too intuitive. The authors want to argue several points, however, none of them are strong enough and can be claimed as 'theorem' (I feel they are only 'observations'). Meanwhile, I feel the authors include such different points in the main text with the expense of readability.

If this is an empirical paper introduced some observation on the self-supervised learning with detailed ablation study and some kinds of theoretical characterization, I’m happy to accept this paper. However, as I don’t feel the authors provide either strong theoretical results or detailed ablation study, meanwhile, the presentation is totally a disaster, I think this paper is not suitable for publishing right now.

---

> ### Author Response · Authors · 2020-11-11
> **Rebuttal for R5 (Part 1/5)**
>
> We thank R5 for all the comments. We do note that this is the most negative outlier review compared to all other reviews, and some subjective opinions stated in this review are in disagreement with very positive opinions stated by other reviewers (i.e. “The paper is well written, with rigorous mathematical derivations” (R2), “Approximating the data distribution with a hierarchical latent tree model is an interesting technique” (R1), “The finding is interesting and novel” (R4).”  However, we have taken this negative outlier review to heart and we can address the many questions raised to further improve this paper.  Answers to every question are labelled **A1** through **A18**, in one to one correspondence with the questions **Q1** through **Q18** raised by the reviewer.
>
> **A1**: We strongly disagree that a theorem strictly requires techniques beyond calculus to be called a theorem. Indeed, many mathematical theorems are proven by a pure calculation (e.g., Cauchy's integral theorem forms the fundamental basis in complex analysis.) As long as the sequence of calculational steps is rigorously justified, the final outcome can be called a theorem, or fundamentally a mathematical statement or result that is true.  We collected all of our results that we thought of as having special significance as important intermediate steps, and we stated them as “Theorems” simply to highlight those particular intermediate results, for the ease of the reader.  For example, the fact that in SimCLR, the learning dynamics at any instant of time is governed by a positive semidefinite covariance operator is a non-trivial and previously completely unrecognized property and so we highlighted it in the form of Theorem 3. Other reviewers agreed this was an interesting result.
>
> **A2**: For empirical evidence, note that BYOL paper already shows that without L2 normalization it still works (See Table 20 in v3 of https://arxiv.org/abs/2006.07733), despite the fact that the activation might blow up during training. We omit the explanation why we choose l2 loss without normalization due to space limitation, and will add it back in the next revision. Besides, we will also provide additional detailed analysis when l2 normalization is present in the next revision. It almost fits into our theoretical framework but with some caveats, and the covariance operator still exists with a slightly different definition.
>
> **A3**: Note that this assumption holds exactly for simple contrastive loss like $r_+ - r_-$, and is present here largely to simplify our technical analysis. We have already relaxed the assumption by assuming
>
> $$\frac{\partial l}{\partial r_{k-}} = - \sum_j h_j(x)h_j(x_k’)q_j(x_+,x)q_j(x_1,x)q_j(x_{k−},x'_{k})$$
>
> where $h_j > 0$ and $q_j > 0$ are arbitrary positive functions (including constant functions), $x$ is un-augmented datapoint used to generate $x_1$ and $x_+$, and $x_k$ used to generate $x_{k-}$. This relaxed assumption leads to a sum of weighted version of covariance operator and our conclusion still follows. We will put this extension in the next revision.
>
> **A4**: We will fix this omission and define beta when we introduce it.
>
> (to be continued)

---

> > ### Comment · AnonReviewer5 · 2020-11-16
> > **Thanks for your reply. I would like to re-emphasize my idea.**
> >
> > A1: I partially agree with the authors' idea. There are several existing theorems only derived by calculation. But these theorems $\textbf{state the results on the terms of interest}$, other than give a calculation results of the gradients. The terms of interests can be sample complexity, convergence results, the properties of solutions etc. And calculating the gradient is only one step towards these results on the terms of interests, which we typically call a Lemma (and not theoretical results and contributions).
> >
> > A2. I didn't notice the results without normalization in BYOL. Sorry for that. But I would like to see the results with normalization. Typically with normalization we need to additionally track the norm of the output, which can be pretty hard. I will not be surprised by there's a similar covariance term in the gradient, but it should make some of the differences for the optimization results, at least the empirical results show without normalization there will be huge performance drop. I would also like to see the differences between the methods w/ and w/o the normalization.
> >
> > A3. In theoretical analysis we tend to make at least assumption as possible, especially when the assumption can potentially influence the results. For me a more realistic assumption is preferred.

---

> > > ### Author Response · Authors · 2020-11-18
> > > **Thanks for your follow-up questions.**
> > >
> > > We thank R5 for the continued interest in our paper.  We have taken many of your concerns to heart and we are significantly revising our papr. A revised version should be up on the site in a few days. We give detailed responses below, some of which cite revisions that will appear soon to address concerns.
> > >
> > > **A1**:  We are encouraged that R5 now partially agrees with us.   We still disagree though with R5’s overly narrow notion that terms of interest in ML are limited to properties like sample complexity, etc.  Indeed, even basic frameworks for understanding SSL have not yet been established.  Furthermore, existing “terms of interest” in ML theory have repeatedly been shown to not accurately describe practice. For example, papers like:
> > >
> > > *C. Zhang et al. Understanding deep learning requires rethinking generalization (ICLR 2017 best paper)*
> > >
> > > *V. Nagarajan and J. Kolter. Uniform convergence may be unable to explain generalization in deep learning (NeurIPS 2019, Winner of The Outstanding New Directions Paper Award)*
> > >
> > > show that some of very basic concepts like model capacity based bounds on generalization error don’t actually characterize generalization error well, motivating the search for new concepts in supervised learning (e.g. data-dependent sample complexity).  But even more importantly, it is unclear what R5 means by “sample complexity” purely within the context of self-supervised learning whose loss function does not explicitly reference any supervised task.  The first step in understanding SSL lies in understanding what features of a data distribution are picked up by an SSL loss function and architecture. This is poorly understood even in the limit of infinite data, and this is the question set out to address.  Showing that the gradient of SIMCLR reduces to a covariance operator that remains positive semidefinite over the entire training process is an important result along this path is an important conceptual result and we highlighted it as a Theorem.  Other reviewers agree with us and disagree with R5 (R4: “the finding is interesting and novel”, R1: “The covariance operator sheds light to the black-box learning process of contrastive learning.”  Furthermore we do discuss “properties of solutions” in illustrative examples in Section 4 for SIMCLR.  Additionally, we will add a completely new section in the appendix discussing properties of solutions of BYOL with linear architectures in the next revision.
> > >
> > > We hope R5 agrees with us that in emerging fields like SSL, where theory is far behind explaining empirical phenomena, one should be very open minded in recognizing interesting and newly discovered “terms of interest”, where traditional ML theory has little to say.
> > >
> > > **A2**: The gradient update analysis with $\ell_2$ normalization on the topmost layer will be updated in the next revision.
> > >
> > > **A3**: First, for simple contrastive loss $r_+ - r_-$, the Theorem 2 and the constant negative gradient assumption in Theorem 3 hold exactly.
> > >
> > > The triplet loss $\max(r_+ - r_- + \alpha, 0)$ and InfoNCE loss satisfy Theorem 2 at all differentiable points using simple gradient check. Thanks to the questions raised by the reviewers, we recently found that those losses lead to a **weighted covariance operator** under a condition that is much weaker than the constant negative gradient assumed in the current version of the paper. We will update this part in the next revision. As a brief introduction, the weighted covariance operator $\mathrm{Op}$ is computed as the following:
> > >
> > > $$
> > > \mathrm{Op}=\frac{1}{2}\mathbb{E}_{\mathbf{x},\mathbf{x}'\sim p(\cdot)}\left[\xi(r) (\bar K_l(\mathbf{x}) - \bar K_l(\mathbf{x}')  (\bar K_l(\mathbf{x}) - \bar K_l(\mathbf{x}')^\intercal \right]
> > > $$
> > >
> > > Here $r := \frac12\|\mathbf{f}_L(\mathbf{x})-\mathbf{f}_L(\mathbf{x}')\|^2_2$ is the squared distance between the outputs of the two sample $\mathbf{x}$ and $\mathbf{x}'$, and $\xi(r) \ge 0$ is a weight function. For simple contrastive loss $\xi(r) \equiv 1$, for triplet loss $\xi(r) \equiv \mathbb{I}(r \le \alpha)$ and for InfoNCE loss $\xi(r) \propto \frac{1}{\tau}e^{-r/\tau}$. Note that for triplet loss and for InfoNCE, still some conditions are needed, which are weaker than the constant gradient condition in the current version of the paper. Intuitively, the triplet and InfoNCE loss are basically add different weights to focus on the pair of samples that are close under the current representation.  For simple contrastive loss, $\xi(r) \equiv 1$ reduces to $\mathbb{V}[\bar K_l(\mathbf{x})]$.
> > >
> > > Both triple loss and InfoNCE are realistic losses used extensively in many papers (and mentioned by other reviewers). For InfoNCE, we have updated our assumption to be weaker than “constant negative gradient” which R5 states a preference for. This is the weakest assumption we have come up with so far.  Given this assumption is weaker than what R5 prefers, we hope R5 is more favorably disposed to it than our initial assumption. Thank you for raising this point.

---

> ### Author Response · Authors · 2020-11-11
> **Rebuttal for R5 (Part 2/5)**
>
> **A5**:  We view the dynamical changing of the covariance operator as a key insight into the power of SSL with nonlinear networks.  After all, if the covariance operator were fixed, then SSL would simply reduce to principal components analysis, computing features that align with maximal eigenmodes of the covariance operator. However, the fact that at any instant of time, which weight patterns get amplified are completely controlled by a time-dependent positive semi-definite covariance operator (that remains PSD at any given time), is a fundamental observation about SimCLR that was unknown (and that other reviewers find interesting).  Indeed this result opens up entirely new avenues for understanding what SSL does by analyzing the dominant time-dependent eigenmodes of this covariance operator.  Thus we believe the time-dependence of this PSD operator is of fundamental interest and importance, and elucidating its existence is a fundamental contribution (and other reviewers agree).
>
> We do not believe that taking the neural tangent kernel (NTK)  limit (infinitely wide limit where learning is slow and the neural tangent kernel remains the same as it was at initialization) would be a way to get an interesting result.  Even in the case of supervised learning, it is unclear that the NTK limit describes well what happens in practice.  For example, recent work (https://arxiv.org/abs/2010.15110) shows that the NTK is not constant, and instead changes rapidly within a few epochs in practice.  Also, the goal of unsupervised learning is feature learning, and the NTK limit corresponds to a lazy training regime where features are not learned (https://arxiv.org/abs/1812.07956). We suspect that performing SSL in the NTK limit with nonlinear networks would simply reduce to kernel PCA with a fixed specific kernel.  In contrast, we show that SSL with finite width networks is much richer.  For example,  as shown in Sec. 4.2, under the covariance operator, dynamics of high-layer weights leads to faster low-layer training.
>
> Finally, note that our covariance operator and the NTK are fundamentally different objects.  The NTK is defined in the sample space and is full-rank if samples are distinct. For super wide networks (compared to the sample size), NTK seldom changes during training and leads to a convex optimization landscape. On the other hand, our covariance operator is defined per layer on any data distribution and network with finite width, and does not grow in size with larger sample size. Finally, while NTK has nothing to do with data augmentation, our covariance operator arises as an emergent property of both the data augmentation and the SSL architecture. Since the covariance operator alone determines learning, its further analysis can aid in the design of both the augmentation procedures and architectures that generate it.
>
> **A6**:  We are not sure what does the reviewer mean by “separable data'' in the context of self-supervised learning settings in which data have no labels. Note that “separable data” is itself a quite strong assumption.
>
> (to be continued)

---

> > ### Comment · AnonReviewer5 · 2020-11-16
> > **Thanks for your reply. I would like to re-emphasize my idea.**
> >
> > A5. I'm not convinced by these arguments. If the covariance operators change over time, then how do we know what happens during the gradient flow? As the authors does not characterize the solutions we get, how can the authors rule out the case the covariance operator will amplify the signal from different directions at different steps, and thus lead to a chaos? I agree with the arguments that this covariance operator can be useful, but this is not enough. How to use this operator? In other words, what's the main argument of this paper? Introducing such covariance operator does not give any interesting results on any terms of interests.
> >
> > A6. I would just want to say that what we really want is the conclusion on some terms of interests, e.g. there are existing work on deep ReLU network for separable data that proves to converge to the max-margin solution, which may help us know what deep ReLU work do for separable data. This paper does not provide any kinds of similar results, e.g. what is the SSL loss doing with deep ReLU network? Will SSL loss with deep ReLU network converge to some specific solution that have desirable property? How fast will it converge? How many samples are we needed for find good solutions? I think these are the questions we want to ask about SSL. We may not be restricted to separable data, say if we have a Gaussian mixture model with different SNR, what will SSL do and how will the performance vary with different SNR? That's also interesting. What I would like to say is, there's no clear conclusion on SSL in this paper.

---

> > > ### Author Response · Authors · 2020-11-19
> > > **Thanks for your follow-up questions.**
> > >
> > > **A5 and A6**:  As recognized by other reviewers, the main point of this paper is to discover the covariance operator during the training of SSL (Sec. 3) for multiple loss functions, show that it is PSD for all training time (Sec 3),  study its properties under different generative models (Sec. 4), show how to use this covariance operator to describe the solutions found under training on those generative models (Sec. 4), and use it to analyze the mysterious property of BYOL, namely that it works negative pairs (Sec. 5 and new appendix in revised version), and perform experiments to test predictions of the theory (Sec. 6).
> > >
> > > In particular, we show in an illustrative example, the hierarchical latent tree model (HLTM), how this covariance operator leads to the amplification of features in intermediate layers of a multilayer ReLU network, that correspond to directly to latent variables at a corresponding layer of the HLTM. While the case we analyze is relatively simple (i.e. assuming the top-down Jacobian $J$ satisfies $J^\intercal J = I$, which we will make it clear in the revised version), we provide a direct example of what R5 claims we do not do: namely we use the covariance operator in an example to show that important latent variables are amplified in the internal representations of a ReLU network, without chaos. Given the above, we strongly disagree with R5 that “there is no conclusion on SSL in our paper.” Moreover, other reviewers agree with us and disagree with R5; 4 other reviewers all gave higher scores than R5. (R1: “Approximating the data distribution with a hierarchical latent tree model is an interesting technique.”).
> > >
> > > We believe the time-dependence of the covariance operator is interesting, and we hope that our work will inspire others to study this time dependence in many settings. Indeed addressing what R5 asks for  (e.g,. convergence proofs for SSL with general deep nonlinear networks, as well as a new as of yet completely undefined generalization of the notion of sample complexity from supervised learning to the SSL setting) would require years of work, and the joint efforts of the community. We feel demanding such results in an early paper on understanding SSL is not reasonable.  We hope our paper  can serve as a starting point towards such efforts. And we have already added some new results on properties of solutions found by BYOL in linear architectures, which will appear in the revised version.
> > >
> > > However, returning to R5’s original Q5 that started this line of thought was a suggestion to take an NTK limit to get rid of the time dependence of the covariance operator.  We responded by saying this would lead to the relatively uninteresting limit of kernel PCA with an NTK kernel. Indeed we already showed in Section 4 that in a linear architecture, where the covariance operator is constant, we show that SIMCLR reduces to PCA within a subspace conserved by data-augmentation.  However, given R5’s belief that the NTK limit would be an interesting direction, we are happy to work this out by the camera ready version; we believe it should be straightforward to do.  But for the revised version which will appear in a few days, we chose to focus on what we feel are more impactful directions, including clarifying the presentation, relaxing assumptions, and understanding BYOL without BN.  However, we will take R5’s concern to heart and include a discussion of SSL in the NTK limit for the camera ready.  We again thank the reviewer for pointing out this direction.

---

> ### Author Response · Authors · 2020-11-11
> **Rebuttal for R5 (Part 3/5)**
>
> **A7**: We first note that Section 4 is meant to simply illustrate our general theory of SSL in specific settings, to explain when it works and when it fails.  To create such illustrative examples, we made two specific assumptions about the data generation and augmentation process:
>
> **First**: we assumed the data is generated through a combination of class specific latents and a nuisance latents.
> **Second**: we assumed the augmentation procedure perturbed the nuisance latents while preserving the class latents.
>
> These assumptions are not required for our general theory, and are only properties of our specific examples.  However, we strongly disagree that these assumptions are unrealistic and difficult to realize in practice.  Indeed we argue that these assumptions are likely required for SSL to succeed in practice, and therefore exemplify properties of both data and augmentation procedures used in practice when SSL succeeds.
>
> For example, many empirical SSL works have already shown that data augmentation is critical for the performance of SSL methods. E.g., SimCLR shows using a correct set of data augmentation is critical for its performance (Fig. 5 in https://arxiv.org/pdf/2002.05709.pdf). Conversely, the recent technical report (https://arxiv.org/pdf/2011.02803.pdf) shows that simply adding semantically or class-specific invariant features in data augmentation immediately kills SSL performance, regardless of the loss function. Both suggest that to understand why SSL works, imposing the right assumption of data augmentation is important, while adding a simple isotropic Gaussian won’t get SSL working.
>
> R5 questions whether it is possible to only perturb the nuisance latent variable. Here is one example: random color distortions are an excellent way to scramble many low level, likely nuisance features, while preserving high level semantically important features, like shape.  For example, many humans could recognize the class identity of color distorted objects, like animals (e.g. a color distorted horse still is recognizable as a horse).  Thus color distortion provides a clear example of what the reviewer claims is impossible to realize in practice.
>
> Therefore, we believe our assumptions, for illustrative purposes only, that augmentation specifically changes nuisance latents, but not class latents, is highly realistic, and reflects what implicitly occurs in practice whenever SSL succeeds.
>
> **A8**: We apologize for the confusion here. Integrating out z^\prime is physics language.  It simply refers to the augmentation averaged connection, averaging over z^\prime, but conditioning on z_0. See the definition of augmentation averaged connection in Theorem 3.  In Sec. 4, the input x = x(z_0,z’) (first paragraph), so we could write bar{K}_l(z_0) after averaging over  z’, which is the only variable changed under data augmentation. As shown in Sec. 4, for different latent variable distributions, we will have different closed forms (or no closed form) for bar{K}_l(z_0).
>
> **A9**:  The entire section 4 is a set of worked examples that demonstrate what the covariance operator looks like under different generative models of data, augmentation procedures and different network architectures, and how the patterns in the dataset are learned in SSL. In the single neuron case and generative models with 1D translation nuisance latent variables, we indeed show that the ReLU activation is important for learning the patterns.
>
> **A10**: can be easily generalized to multiple outputs by replacing a scalar w_{2,j} with a vector. Here we discuss the scalar case for illustration and notational simplicity.
>
> **A11**: As pointed in the paper, the network to be studied is in Fig.2(c). The matrix A changes during training due to the change of w, but remains a PSD matrix since it is a covariance matrix (Eqn. 8). The condition Cov[u_j, u_k] = 0 is a technical condition to allow simple and concise dynamics. Otherwise things can be much complicated and harder to interpret. We will add more general cases without the condition of Cov[u_j, u_k] = 0 in our next revision. Also what does R5 mean by “see 9”?
>
> **A12**: We are confused. What does R5 mean by the “indicator function” at the end of Page 7 (we assume R5 means the paragraph after Theorem 7)? We didn’t appear to use any indicator function there.
>
> We guess that R5 means Page 5 instead. If our guess is correct, we precisely say that the indicator function I(w_j . x > 0) can change during optimization, except when the activation is linear and the gating is indeed a constant 1. So we are not sure about the concerns (maybe R5 misses the phrase “while for linear node”?). We will expand this analysis to make it more clear in the appendix in our next revision.
>
> (to be continued)

---

> > ### Comment · AnonReviewer5 · 2020-11-16
> > **Thanks for your reply. I would like to re-emphasize my idea.**
> >
> > A7. I'm not convinced by the authors' arguments. Here's what I want to ask:
> > (1). Can we identify the two components in practice? In other words, how can we demonstrate that the augmentation, e.g. random color distortion, does not change the class/sample-specific latent? Considering we want to ask the texture of some objects, which may be determined by the color, will this change the class/sample-specific latent?
> > (2). Will such assumption rule out some scenarios of interests, for example, I mentioned in the review? It's hard to imagine such augmentation that perturb both the class/sample-specific latent and nuisance latent will not work in SSL.
> > Overall, my point is the authors introduce this assumption without further validation and demonstration. Meanwhile, although it could be a possible mechanisms on how SSL works, it really rules out some scenarios of interests. I don't think introducing such assumption is necessary. And if the authors want to introduce this assumption, please give more formal description and give strong positive and negative results on this assumption.
> >
> > A8. I would like the authors make a revision on this part. It's pretty hard to understand.
> >
> > A9. I'm happy to see illustrative examples in the paper. The main question is, what does the illustrative examples want to show? If this is a paper show the necessarily of the ReLU activation, I think such example is proper. But turns out this is not a paper about how ReLU activation works in SSL, right?
> >
> > A10. It's interesting to me if multiple output will make the gradient confusion. Can the authors make more discussion on that?
> >
> > A11. I would like to say that, for a theoretical paper, we need to make every part concise. I prefer every part need to be rigorous is presenting in a formal way, as I said in the 9th question. I don't think make such assumption on the condition convincing. But hope the authors can give more convincing results in the next revision.
> >
> > A12. I make a mistake on that. I mean in the last of Page 5 rather than Page 7. The issue is why the gate is always $1$, as during the optimization the parameters are changing over time, and at some point it can change from $1$ to $0$, can we always rule out this possibility? What's the meaning of linear nodes, we are talking about ReLU network right? I'm totally lost on that.

---

> > > ### Author Response · Authors · 2020-11-18
> > > **Thanks for your follow-up questions.**
> > >
> > > **A7**:  It is not at all implausible to believe that the data augmentation procedures used in practice (e.g. crop, rotate, color distortion, skew, scaling, etc) do not alter class specific latents, especially variables that correspond directly to class labels. Indeed this is widely taken as common sense in the field of SSL practice, so we are surprised R5 continues to insist that this assumption is unreasonable, even if only limited to simple illustrative examples in Sec. 4 and not assumed in our general theoretical framework.  We encourage R5 to examine Figure 4 of the SIMCLR paper (https://arxiv.org/pdf/2002.05709.pdf) which shows 10 different augmented views of a dog.  Every single augmented view, including the color distortion, is clearly recognizable as a dog.  Thus augmentation procedures used in practice do not destroy or change the class label. Of course it is extremely difficult to formally prove this, because there is as of yet no formal mathematical expression for the presence or absence of a dog in a pixel image.  But we believe Fig. 4 (and many other figures like it in the literature) conclusively demonstrate that our model of data-augmentation as modifying nuisance features but preserving at least class specific latents is not at all unreasonable and aligns well with practice.
> > >
> > > Our theory suggests that augmentation procedures that modify class specific latents should only impair performance.  Again, the covariance operator, which, interestingly, R5 continually criticizes as an insignificant contribution, plays a key role in this argument.  The covariance operator is obtained by first averaging all the augmented views of each data point.  Then the covariance of these augmentation averaged data points is computed, and this operator drives learning at any given time.  If the augmentation transforms latents specific to one class, to that of another class, then these two latents will be averaged together, before the variability in the data is computed.  This averaging process will render it more difficult to distinguish the two classes. We hope this example use of the covariance operator provides another illustration of its usefulness in reasoning about SSL.
> > >
> > > Finally, we note that R5 makes a strong assertion that our assumptions about data augmentation (again used for illustrative purposes only in Sec. 4) “really rules out some scenarios of interests. I don't think introducing such assumption is necessary.”  However R5 provides no logical argument for why some form of this assumption is unnecessary, nor provides a concrete example of when all class specific latents are perturbed by data augmentation yet SSL can still work.  We in contrast provide a covariance operator based argument that SSL is unlikely to work in such a setting. However if R5 has a concrete example in mind in which all class specific latents are changed by data augmentation but SSL still works, we are happy to discuss it further.  We do not view intermediate cases in which some class specific latents are perturbed at at least some others are preserved as qualitatively distinct from our assumption, and for illustrative purposes we believe it is more instructive to focus on clear examples where class specific latents are preserved.
> > >
> > > **A8**: We again apologize for confusions we may have generated. We will make this clearer in the revised version to appear in a few days.
> > >
> > > **A9**: As we have said, Sec. 4 are already illustrative examples used to show how the representation is learned during training, by checking the explicit form of covariance operator under different generative models. They cover different cases and some might discuss the property of ReLU networks, while others don’t, as a contrast.  This paper is about the covariance operator and all cases in Sec. 4 are basically examples demonstrating its explicit form and its roles in different scenarios, and it is natural to have ReLU activation in some of the examples.  We also provide an example in which a ReLU unit can learn but a linear unit cannot.
> > >
> > > **A10**: We don't understand “make the gradient confusion” mentioned by R5. In the revised appendix we provide an example of BYOL with multiple outputs.
> > >
> > > **A11**: What part is not rigorous? Please show us. The assumption is used to get a concise weight dynamics and we will show more general cases in the next revision as well as clarify the proof.
> > >
> > > **A12**: We never say $A_j$ is constant in the ReLU case. As we have mentioned in the original version of the paper (Page 5, last paragraph. we repeat the exact sentences here for clarity): “Note that if we consider ReLU neurons, $A_j$ changes with $\mathbf{w}_{1,j}$; while for linear nodes, $A_j$ is a constant, since the gating $\mathbb{I}(\mathbf{w}^\intercal_{1,j} \mathbf{x} > 0)$ is always $1$”. The hidden layer nodes can be either linear (where the gating is always 1) or has ReLU activation (where the gating changes over time). Here we discuss two situations at the same time.

---

> ### Author Response · Authors · 2020-11-11
> **Rebuttal for R5 (Part 4/5)**
>
> **A13**: The motivation behind the HLTM model is to have a hierarchical generative process for generating data at multiple levels of hierarchy.  Intuitively, this captures the idea that different objects (top level of hierarchy) are made of parts (lower level) that are themselves made of smaller subparts (even lower level).  The abstract feature vectors we use at different levels could for example reflect 1 hot vectors for different objects (top level), and k-hot vectors for the presence or absence of k parts in a list of parts, conditional on object identity (lower level).    These features could combine with nuisance features (e.g. poses or relative orientations of subparts) to generate pictures of objects, for example.  This was the underlying motivation for the hierarchical tree model.  We believe this picture is very general.
>
> Of course the actual distributions we chose for the tree were quite simple.  In this simple setting, we wanted to address a fundamental question: when do deep neural networks with multiple layers learn neurons at intermediate layers whose outputs across stimuli correlate with intermediate layers of the generative model  (i.e. can SSL learn part detectors in intermediate layers, while only having access to low level representations of objects in the input layers).  We believe this is a fundamental question, and the HLTM provides the simplest nontrivial setting in which this question can be settled. To the best of our knowledge, no prior work has done such an analysis in SSL settings.
>
> We note that R1 commented that “Approximating the data distribution with a hierarchical latent tree model is an interesting technique.”  We apologize that our motivation was not clear. We will give an expanded description of the HLTM in the appendix.
>
> But for now, just for clarity, to more specifically connect to our notation, we let  z_0 represent label of different objects (e.g., cat, human, computer, etc), and z’ to be variables that determine the location and the size of each object part (e.g., the location of a head, the limbs if the input image contains a human, etc). As illustrated in Fig. 2(d) and mentioned in the first paragraph of Sec 4.3, when “generating” one object, we first generate its object category, then generate the location z’ of its parts given the object label, and generate the subparts (also z’), until the image is generated as x, which serves as the input of the neural network. In this case, data augmentation on the input image x naturally changes z’ but not z_0 (e.g., a rotation/scaling on the image changes the location/size of each object part, but not the object identity). We consider locally connected rather than FC settings because the analysis is easier.
>
> **A14**: While the description of the HLTM and generative model is logically complete and self-contained (and other reviewers appreciated it), we do sympathize with this reviewer that the description of the HLTM and the neural architecture for performing SSL is quite terse.  We will provide a greatly expanded description of both in the appendix. But at a high level, the HLTM is a branching diffusion process (a well known process) and the neural network is simply a multilayer neural network (again well known).
>
> **A15**: In contrast to R5’s criticism, Theorems in Sec 4.3 indeed give hints on the quality and the meaning of the intermediate representation. Theorem 6 shows that even without training, from an over-parameterized deep network, one can find intermediate nodes that are correlated with the latent variables at the same corresponding layer (and the correlation can be lower bounded). We also explain why this correlation could increase over training. As we mentioned, all statements not in the theorems are not intended to be rigorous but instead intended to give valuable intuition as to how things work. We leave more detailed analysis (e.g., sample complexity and convergence analysis) to the future work.
>
> Finally, we note that the definition of “quality of the representation” is somewhat up for grabs.  We have defined it to be whether or not the neural network can learn features that correlate with the intermediate latent variables in the hierarchical latent tree model.  There is precedent for this idea in general - where data is generated by important latent variables, and it is asked whether or not neural network training can recover these latent variables. See e.g. https://arxiv.org/pdf/1310.6343.pdf  and https://arxiv.org/abs/1902.09229 . In the paper it is assumed that downstream classification tasks of interest have class labels that depend only on important latent variables, so an important goal of SSL then involves learning explicit representations that are linearly related to the latent variables.
>
> (to be continued)

---

> > ### Comment · AnonReviewer5 · 2020-11-16
> > **Thanks for your reply. I would like to re-emphasize my idea.**
> >
> > A13. I see the motivation now, but will the learning of neural networks follow this kinds of human-level understanding, especially when we don't know the knowledge of the generative model? From the description I feel the network can only be built with the known generative model.
> >
> > A14. I would like to say, even the process and network structure are well-known, it's not proper to include only a figure as the description of these terms of interests in a theory paper. It's not concise, and will lead to confusion. The most important thing for a theory paper is accurate, right?
> >
> > A15. I need to say, due to the lack of necessary explanations of the notations, I may not understand the whole picture in Section 4.3 correctly. I would like to ask what is $v_j(1)$ and $v_j(0)$, and what will their difference measure. I also have questions on Theorem 6:
> > (1) Is this a static result, i.e. does not include the optimization? That is to say, even we have lucky nodes, how can we provably find these lucky nodes?
> > (2) If we only know the lucky nodes exist, and don't know how to find the lucky nodes, then will the downstream task be influenced by the unlucky nodes? Can we give provable results on the risks of downstream tasks?
> >
> > Also, in [AKK+19] there is no generative model assumption and in [ABG+13] the generative model is not as complicate as the authors'. I would like to say, the conclusion in their papers are directly related to the terms of interests. For example [AKK+19] gives the performance of learned representation on downstream tasks, while this paper does not have such results.

---

> > > ### Author Response · Authors · 2020-11-18
> > > **Thanks for your follow-up questions (1/2)**
> > >
> > > **A13**: We are glad R5 now appreciates the motivation behind our HLTM. To mathematically elucidate what features in a data distribution SSL learns, we fundamentally need a model of the data distribution. We made several assumptions of increasing complexity, culminating in the HLTM, which again other reviewers liked (R1: “Approximating the data distribution with a hierarchical latent tree model is an interesting technique”).  Without any assumptions about the data generation process, we might end up with much weaker conclusions which could be more general but might not be useful in practice. This could be regarded as “weakness” but we argue this should be treated as strength since the assumption abstractly reflects aspects of the real-world data distribution (hierarchical structure of objects in different poses) and we care about performance in the real world.  Understanding SSL in even more realistic models of the data constitutes a years long research program, and we hope our work can inspire future progress along these directions.
> > >
> > > **A14**: We acknowledge Section 4.3 can be dense and difficult to understand. We are dealing with binary hierarchical generative models, and $v_j(z_\mu) = \mathbb{E}_z[f_j | z_\mu]$ is the expected activation (Table 1), conditioned on the hidden value of $z_\mu$. Since $z_\mu$ is a binary variable, we only need to check $v_j(0)$ and $v_j(1)$. If they are far apart, then the node $j$ is strongly correlated with $z_\mu$. That’s the meaning behind the metric $|v_j^2(0) - v_j^2(1)|$. We take the square because the math expression is easy to compute. We are revising this section to clarify; the revision should appear in a few days.
> > >
> > > **A15**: Theorem 6 is a static result at initialization, without any SGD training or optimization. We don’t need to find the lucky node explicitly. As long as the metric $|v_j^2(0) - v_j^2(1)|$ can grow during the SGD training, a good representation emerges during training that is correlated with the latent variable **without** its direct supervision, which is the point we want to show. We are not dealing with downstream tasks in this paper and leave it for future work.

---

> > > ### Author Response · Authors · 2020-11-18
> > > **Thanks for your follow-up questions (2/2)**
> > >
> > > Regarding the two papers we gave as examples, and which R5 brings up in response, our only reason for bringing them up was to give examples of where making assumptions about the data were useful.  We never intended to initiate a direct and detailed comparison between those papers and ours.  Their goals are fundamentally different from ours, and we believe both those papers and ours are all interesting and complementary. However, since R5 brings those papers up in a more detailed manner, we will respond in a more detailed fashion to emphasize contrasts.
> > >
> > > **[AKK+19]** (http://proceedings.mlr.press/v97/saunshi19a/saunshi19a.pdf):
> > > *Sanjeev Arora, Hrishikesh Khandeparkar, Mikhail Khodak, Orestis Plevrakis, Nikunj Saunshi, A Theoretical Analysis of Contrastive Unsupervised Representation Learning, ICML 2019.*
> > >
> > > In our paper, one of the many aspects is to attempt to open the blackbox of the neural network and try to check whether there is a correlation between the intermediate layer of the deep models and the intermediate latent variables. To achieve this level of understanding, we assumed a generative model with intermediate latent variables is natural (and is the only way we could think about). [AKK+19] focused on contrastive loss functions and not BYOL,  and didn’t open the blackbox. In fact, they assumed an abstract function class which might not have anything to do with deep models. Therefore such a “complicated” assumption of data generation is not needed. While there are indeed fewer assumptions, it is important to be aware that we also pay the price that insight of what deep models is not revealed as well. Finally this paper focused specifically on downstream tasks, whereas we focus instead on what SSL learns in parametric models of data, without having to refer to downstream tasks.  Hence the papers are quite complementary.
> > >
> > > Here we feel that it is necessary to bring the discussion to a higher level. It is not always good to encourage theoretical papers to have very few (and general) assumptions. With very few assumptions, we might get a nice bound, but that bound can also be vacuous and is not relevant to the real problems. The best assumptions are those who connect closely with the real world and those that make or break the algorithm. We would argue that the assumptions we made about data augmentations are precisely like that.
> > >
> > > **[ABG+13]** (https://arxiv.org/pdf/1310.6343.pdf)
> > > *Sanjeev Arora, Aditya Bhaskara, Rong Ge, Tengyu Ma, Provable bounds for learning some deep representations, ICML 2014*
> > >
> > > Also, we never intended to directly compare to [ABG+13], we only intended to cite it as nice example of assuming a generative model of data. There are many differences between this paper and ours:
> > >
> > > 1. It uses layer-wise training which is different from SGD/GD we are considering.
> > > 2. It also assumes a random generative model.
> > > 3. More importantly, it assumes that the samples of latent variables at each intermediate layer are known (see Alg. 1 in Sec. 4, and paragraphs before Sec. 6 in https://arxiv.org/pdf/1310.6343.pdf), so that the edges between consecutive layers can be recovered via Graph Recovery algorithms.
> > >
> > > We emphasize that all these are strong assumptions that are not that related to empirical practice in deep learning. The last assumption (3) is particularly strong. One of the main puzzles of the deep models is how back-propagation (i.e., training end-to-end) works and how the intermediate representation emerges **without** the direct supervision of the latent variables. In contrast, our paper made an initial attempt to answer these questions in SSL. Since the problems addressed by [ABG+13] and by our paper are completely different, the detailed comparison of the assumptions doesn’t make too much sense.  But the idea that assumptions about data are required to get interesting theoretical results about the behavior of deep learning is common amongst all 3 papers.
> > >
> > > From these two papers, we hope R5 can also understand how difficult it is to show mathematically rigorous results whose assumptions are aligned with empirical settings. It takes years of efforts and a joint effort between researchers working on theory and practice. We hope R5 appreciates our efforts and every small step matters.

---

> ### Author Response · Authors · 2020-11-11
> **Rebuttal for R5 (Part 5/5)**
>
> **A16**: “sym” means that W=W’ (the weight of the online network is the same as that of the target network), defined right before Eqn. 10.  deltaW_l^{BN} is defined right before Eqn. 13, which means the difference between gradient before and after adding BN.
>
> **A17**: We believe R5 has a misunderstanding about the details of BatchNorm.
>
> 1. In BatchNorm, the sample mean is back-propagated. R5 can check the original BatchNorm paper https://arxiv.org/pdf/1502.03167.pdf, page 4, in equations right before Sec. 3.1, the expression of $\partial l / \partial x_i$ contains both $\partial l / \partial \mu_B$ and $\partial l / \partial \sigma_B$, and thus the gradient with respect to x will backpropagate through both sample mean and standard deviation. Therefore, in terms of zero-mean operations, BN uses “x - x.mean()” rather than “x-x.mean().detach()” as suggested by R5.
>
> 2. During training, the mean subtracted by BN is NOT moving average statistics, but precisely the sample mean in the minibatch. R5 could check Alg. 1 block in the original BN paper on page 3. Also In PyTorch implementation, we can clearly see that during training it uses batch statistics rather than running mean/std: https://github.com/pytorch/pytorch/blob/master/aten/src/ATen/native/Normalization.cpp#L314-L320
>
> Our analysis on Sec. 5 gives a “sufficient condition” under which BYOL will behave like SimCLR, and thus can work. While Theorem 7 is rigorous, in its current version, the remaining statement (at the end of Page 7) is a rough reasoning. We will turn it into a rigorous theorem in the next revision.
>
> Being a “sufficient reasoning”, our argument on BYOL does not rule out other possibilities that enable BYOL to work without BN, e.g, using Weight Standardization and Group Norm without cross sample statistics.  We will rewrite the main message of this section to be that with BN, BYOL behaves like SimCLR in that a PSD covariance operator governs the dynamics.
>
> **A18 and summary**:  We are not sure how to respond to the claim that our presentation is “weird”  especially when other reviewers enjoyed the paper (e.g. R2 says “The paper is well written, with rigorous mathematical derivations” and “Simulation experiments were provided to justify the theoretical findings.”).   Regarding presentation, R5 focuses on distinctions between observations, intuitions, rigorous theory and empirics.  We believe all four approaches constitute useful avenues for advancing our conceptual understanding and we freely engage in all four approaches, including the use of instructive examples, and making clear when we are making rigorous claims by codifying them in Theorems, and putting intuition outside of these theorems (again we disagree that true mathematical statements derivable from calculus cannot be labelled theorems).
>
> We hope that our explanations in this comment furthermore justify why we made some of the assumptions we made in our illustrative examples, why they are natural and relevant to practice, and why some other assumptions can be relaxed.  We have taken R5 complaints about lack of clarity to heart also and we will add substantially expanded descriptions of various aspects of this paper in the appendix, especially the HLTM and the motivation behind our illustrative examples, in order to ensure that the confusions that occurred for R5 do not occur for future readers.
>
> Overall, we thank very much R5 for the many comments, as they will significantly help us improve our paper. Given R5 is an outlier negative reviewer, amongst 4 other more positive reviewers, we sincerely hope Rev.5 will consider our comments seriously and consider improving his/her score if he/she finds our comments satisfactory. If not we are very happy to engage further with any remaining concerns.

---

> > ### Comment · AnonReviewer5 · 2020-11-16
> > **Thanks for your reply. I would like to re-emphasize my idea.**
> >
> > A17. Thanks for pointing out and I obfuscate the mechanism of batch normalization during the training and testing phase. Sorry for that.
> >
> > Overall: I would still like to say, if we view this paper as a paper discussing rigorous theoretical properties on SSL, then I cannot find a main point the authors want to argue theoretically and argue rigorously. Probably I have some misunderstanding on part of the arguments, but I cannot see how this paper benefits our understanding on the terms of interests on SSL, e.g. convergence, properties of solutions, sample complexity, etc. This paper indeed contains several arguments, all of them are interesting, but none of this arguments give a clear conclusion on any terms of interests under any kinds of assumptions. In contrast, most theoretical paper would focus on only one points. For example, [AKK+19] argued how SSL will give a provable guarantee on the downstream tasks throughout the paper, with different kinds of assumptions and algorithms. That's the main reason I think this paper is not ready for publication. Also, a theoretical paper should be concise and clear. I'm happy to discuss. And if we can reach a consensus on what a theoretical paper should be like, I would raise my score.

---

> > > ### Author Response · Authors · 2020-11-18
> > > **Thanks for your follow-up questions**
> > >
> > > **Overall**: We do thank the reviewer for acknowledging that our paper contains “several arguments” and that “all of them are interesting.”  However,  we find summary dismissals of our paper by R5 as “containing no conclusions,”  and “has no main point” to be overly harsh, and far outside the reasonable standards of review (as quantitatively evidenced by the fact that the reviewer’s score is a significantly low outlier amongst 5 scores).
> > >
> > > We hope we have argued convincingly above the many contributions of our paper, which we repeat here. Our fundamental goal is to understand what features are picked up by SSL in different data distributions without having to reference downstream tasks to assess performance. This is a fundamental unanswered question, so we focused on the simplest nontrivial cases.  We demonstrate a covariance operator governs SSL learning dynamics (Sec. 3) for multiple loss functions, show that it is PSD for all training time (Sec 3),  study its properties under different generative models, augmentation procedures, and architectures of increasing complexity (Sec. 4), show how to use this covariance operator to describe the solutions found under training on those generative models (Sec. 4), and use it to analyze the mysterious property of BYOL, namely that it works without negative pairs (Sec. 5 and new appendix in revised version dealing with the case without BatchNorm), and perform experiments to test predictions of the theory (Sec. 6).
> > >
> > > As stated above, demanding convergence proofs for deep ReLU networks trained on complex generative models is an extremely high, and unreasonable bar for publication as this would be a years long effort. Moreover, focusing on concepts like sample complexity in an unsupervised setting also seems like an unreasonable demand given such issues are completely open in many supervised settings.  Finally, as noted above, [AKK+19] is a nice paper but has fundamentally different goals from ours, and is very complementary. Finally we disagree that there is any strict rule that a “theoretical paper should only focus on one point.”
> > >
> > > Finally, we are puzzled by the final statement of the reviewer: “if we can reach a consensus on what a theoretical paper should be like, I would raise my score.” We are not sure what R5 means by this, but we suspect R5 may have an overly rigid view as to what a useful contribution to our conceptual understanding of deep learning in different settings can look like. R5 has often referred to some platonic ideal of a “theory” paper that is limited to discussing a very specific set of concepts (convergence, sample complexity, etc.) using highly mathematical proofs that make very few assumptions (and in R5’s initial comments these proofs must strictly use mathematical techniques that go beyond calculus). This view is emblematic of the perspective of a pure mathematician.
> > >
> > > We (and evidently the other 4 reviewers) seem to have a more open minded view as to many more ways to obtain conceptual insights into an interdisciplinary field as complex as deep learning that rely on multiple methods, including direct calculations, instructive toy examples, reasonable and perhaps even restrictive assumptions that nevertheless mirror in abstract the structure of real world data, and intuition corroborated by a combination of calculations, proofs and experiments.
> > >
> > > We encourage R5 for example to look at the references to a review article (https://www.annualreviews.org/doi/abs/10.1146/annurev-conmatphys-031119-050745)  that highlights how conceptual insights into the nature of deep learning can be derived by combining many different methods of inquiry, spanning simulations, toy models, and useful calculations, often generated by physicists and applied mathematicians, rather than pure mathematicians.  Indeed pure mathematicians may not agree that such papers are “theory” papers, whatever that means. But the field as a whole cannot deny that the conceptual insights gained from this body of work are not useful and important (as quantitatively evidenced by citation counts).
> > >
> > > We harp on this point only because R5 has stated “if we can reach a consensus on what a theoretical paper should be like, I would raise my score.” We encourage R5 to be more inclusive and open minded about what constitutes conceptual insights, especially in an emerging field like SSL in which “terms of interest” R5 is used to from other fields are either not useful or are currently far from within reach. And we encourage R5 to be not overly rigid about platonic notions of what constitutes a theory paper. With this open mindedness, the fact that R5 believes our paper contains “several arguments” and that “all of them are interesting.” should be enough to raise his/her score outside the negative outlier regime amongst 5 reviews.  We hope R5 will like our revised version, which will be enhanced for clarity specific to the concerns of R5 and will contain new results about BYOL without BatchNorm.

---

### Author Response · Authors · 2020-11-14
**Author Response on Common issues (Part 1/2)**

We thank all our  reviewers for their insightful comments. Here we answer common questions that arose. We also respond specifically to each reviewer about questions that arose specific to each reviewer.

**Q**: Does the recent paper [1] “BYOL works even without batch statistics” invalidate this submission / change its conclusion substantially [R1, R3]?

**A**: We note that [1] was  released on Oct. 20, after the ICLR deadline (Oct. 2), which is the reason why we didn’t cite it or analyze it. Second, the main contribution of our analysis is in the setting of SimCLR, revealing that a time dependent positive definite covariance operator (that depends critically on the data distribution, augmentation distribution, and network architecture) governs SIMCLR learning dynamics.  Thus [1] has no impact on these contributions.

The only relevant section in relation to [1] is, Sec. 5, which gives approximate conditions under which BYOL will behave like SimCLR, in that its learning dynamics will also be approximately governed by a positive definite covariance operator as well. Basically we show that when BN is included and when the extra predictor has relative small weights, then a positive definite covariance operator arises for BYOL.   Thus under these conditions, our analysis shows BYOL will work whenever SimCLR works.  However, our conditions are only sufficient and not necessary, in the sense that they do not rule out other possibilities under which BYOL could work,  e.g. by  using Weight Standardization and Group Norm without cross sample statistics as in [1].

We acknowledge that our wording (e.g., “BN plays a critical role in BYOL”) in Sec. 5 can be misleading and we will tone down this wording and clarify our discussion to emphasize the consistency with [1].  Namely, this section will be renamed to something like BYOL behaves like SimCLR with BN.  And in the section we will explicitly state that we do not claim that BN is required for BYOL to work, citing the new Ref. [1].  Thus both of these works will be entirely consistent with each other.

Furthermore, within our theoretical framework, we are currently working on theoretical explanations for [1]. The intuition here is that if $E_x[\bar K(x)] = 0$, then BYOL again reduces to approximate SimCLR no matter whether BN is present or not. And WS+GN might achieve this sufficient condition if some assumption holds. We think R1’s guess about “GroupNorm + weight standardization also provide implicit contrastive objective, similar as BN” is on the right track.

**Q**: Can the analysis extend to other kinds of loss, in addition to InfoNCE (e.g., triple loss)?

**A**: Note that this condition holds exactly for simple contrastive loss like $r_+ - r_-$. Triple loss (https://arxiv.org/abs/1503.03832, https://en.wikipedia.org/wiki/Triplet_loss ) is $\max(r_+ - r_- + \alpha, 0)$, which is equivalent to $(r_+ - r_-) * Indicator(r_+ - r_- + \alpha > 0)$ when taking gradients. So it naturally satisfies all our assumptions (including Theorem 2 and our assumption $\partial L / \partial r_- < 0$ and is a constant in Theorem 3). Therefore, Theorem 3 can be applied. The role of the indicator is to change the data distribution dynamically so that the training naturally focuses on the data pairs that are not well trained (i.e., $r_+$ is large compared to $r_-$).

**Q**: Is the condition $\partial l / \partial r_{k-} = -\beta / H$ too strong or unrealistic [R1, R2, R4, R5]:

**A**: As mentioned above, the condition holds true for triple loss and simple contrastive loss. It is an approximate for contrastive loss, used to simplify our technical analysis. Furthermore, we can relax the assumption by assuming

$$\frac{\partial l}{\partial r_{k-}} = - \sum_j h_j(x)h_j(x_k’)q_j(x_+,x)q_j(x_1,x)q_j(x_{k−},x'_{k})$$

where $h_j > 0$ and $q_j > 0$ are arbitrary positive functions (including constant functions), $x$ is un-augmented datapoint used to generate $x_1$ and $x_+$, and $x_k$ used to generate $x_{k-}$. This relaxed assumption leads to a sum of weighted versions of covariance operators and our conclusion still follows. We will put this extension in the next revision.

Note that since we know the sufficient condition that the covariance operator appears, we might find other new loss functions that follow Theorem 2, which we leave for future work.

(to be continued)

---

> ### Author Response · Authors · 2020-11-19
> **Updated answer for the common question of strong constant gradient condition**
>
> **Update Answer** to strong constant gradient condition $\partial L/\partial r_{k-} = -\beta / H$.
>
> Thanks to the questions raised by the reviewers, after some study, we recently found that both triplets loss and InfoNCE loss can lead to a **weighted covariance operator** under a condition that is much weaker than the constant negative gradient assumed in the current version of the paper. We will update this part in the next revision. As a brief introduction, the weighted covariance operator $\mathrm{Op}$ is computed as the following:
>
> $$
> \mathrm{Op}=\frac{1}{2}\mathbb{E}_{\mathbf{x},\mathbf{x}'\sim p(\cdot)}\left[\xi(r) (\bar K_l(\mathbf{x}) - \bar K_l(\mathbf{x}')  (\bar K_l(\mathbf{x}) - \bar K_l(\mathbf{x}')^\intercal \right]
> $$
>
> Here $r := \frac12\|\mathbf{f}_L(\mathbf{x})-\mathbf{f}_L(\mathbf{x}')\|^2_2$ is the squared distance between the outputs of the two sample $\mathbf{x}$ and $\mathbf{x}'$, and $\xi(r) \ge 0$ is a weight function. For simple contrastive loss $\xi(r) \equiv 1$, for triplet loss $\xi(r) \equiv \mathbb{I}(r \le \alpha)$ and for InfoNCE loss $\xi(r) \propto \frac{1}{\tau}e^{-r/\tau}$. Note that for triplet loss and for InfoNCE, still some conditions are needed, which are weaker than the constant gradient condition in the current version of the paper. We will list these conditions in the revision. Intuitively, the triplet and InfoNCE loss are basically add different weights to focus on the pair of samples that are close under the current representation.  For simple contrastive loss, $\xi(r) \equiv 1$ reduces to $\mathbb{V}[\bar K_l(\mathbf{x})]$.

---

### Author Response · Authors · 2020-11-14
**Author Response on Common issues (Part 2/2)**

**Q**: Are some assumptions made in the paper unrealistic [R5]?:

**A**: Compared to existing literature on theoretical understanding of DL, our framework doesn’t make strong assumptions throughout the analysis. There is no simple parametric assumption for the data distribution (e.g,. Gaussian, mixture of Gaussian, linear separable, generated by a fixed teacher network) and no strong assumption on the size of the network (e.g., infinite input dimension/width/depth). We study ReLU activation that is extensively used in the empirical work, analyze SimCLR/BYOL that are very recent SoTA empirical methods in SSL, use standard training methods like GD/SGD, and focus on loss functions that are extensively used. The covariance operator exists throughout the training rather than just at the initialization (which is the main focus on many theory papers).

For generative models and data augmentation, we made two basic assumptions:

**First**: we assumed the data is generated through a combination of class specific latents and a nuisance latents.
**Second**: we assumed the augmentation procedure perturbed the nuisance latents while preserving the class latents.

First note that these two assumptions are not necessary for the general theory at all.  We only use them in our illustrative examples to illustrate the general theory in specific cases.  However, we do believe these two assumptions are closely related to what occurs in empirical practice.

First, many empirical SSL works have already shown that data augmentation is critical for the performance of SSL methods. E.g., SimCLR shows with different sets of data augmentation, the performance is very different (Fig. 5 in https://arxiv.org/pdf/2002.05709.pdf). Conversely, the recent technical report (https://arxiv.org/pdf/2011.02803.pdf) shows that simply adding semantically or class-specific invariant features in data augmentation immediately kills SSL performance, regardless of the loss function. Therefore, in order to understand why SSL works, we must model the process of data augmentation, and we believe the success of existing data augmentation procedures likes in their ability to scramble irrelevant nuisance features, while preserving semantically important class distinctive features.

R5 questions whether the second assumption is realistic. Here is one example: random color distortions are an excellent way to scramble many low level, likely nuisance features, while preserving high level semantically important features, like shape.  For example, many humans could recognize the class identity of color distorted objects, like animals (e.g. a color distorted horse still is recognizable as a horse).  Thus color distortion provides a clear example of what the reviewer claims is impossible to realize in practice.

---

### Author Response · Authors · 2020-11-24
**Revision**

**Summary of updates to paper in response to reviewers**

We again thank the reviewers for their careful reviews of our work.  We have taken these reviews seriously and made many major updates to our paper, which we have uploaded in our newly submitted revision. Here we summarize the major updates (in addition to these major updates, we have made numerous modifications for clarity which we do not list here).

**First [R1, R3]:** To address the questions about the relation between our paper and the paper “BYOL works even without batch statistics” [1] (which appeared after our paper was submitted), we have derived an entirely new analysis of BYOL for shallow linear architectures without any batch normalization whatsoever.  We find very interesting analytic results that yield conceptual insights into why BYOL can learn useful, non-collapsed representations without having any contrastive terms induced either by BatchNorm or by explicit negative pairs, in what is likely to be the simplest non-trivial setting of shallow linear networks. The basic idea is that collapsed solutions do exist, but they are unstable, while the non-collapsed solutions are stable.  Therefore BYOL, in our simple setting, avoids the former and converges to the latter.  Moreover, we show how the existence of a predictor network is critical for establishing instability of the collapsed-solutions by dramatically modifying the learning dynamics. While for more complicated encoder like ResNet, existing blogpost indeeds shows that BYOL without any normalization still collapses, our analysis provides a non-trivial example showing an important difference with/without the predictor.

All of this analysis can be found in an entirely new Section F in the appendix, and we now refer to these important results in the abstract, introduction, and results sections.  We hope the reviewers will appreciate our responsiveness to a paper that appeared after our paper was submitted.

We furthermore are explicit about the limitations of our new result, stating in the results that “we leave an analysis of other normalization techniques in nonlinear settings [for BYOL] for future work.”

**Second [R1, R3]:** We note that none of the above results invalidate any of our analytic work showing that BatchNorm does indeed introduce contrastive terms in BYOL.  This statement remains true.  However, of course, we never claimed that other normalization schemes that don’t introduce cross-batch statistics could not also make BYOL work, since we never experimented with such techniques, as [1] did. We have now prominently cited reference [1] in our introduction and clarified that while BN does indeed introduce contrastive terms in BYOL, other normalization methods that don’t can still work.  We have been much more careful with our wording in the revised abstract, introduction and results sections regarding all of these issues, and emphasizing consistency between our results, those of [1], and our new results in Appendix F.

**Third [R1, R2, R4, R5]:** We expected the qualitative conclusions of our results would be relatively robust to the choice of loss function. While our analysis holds rigorously for simple contrastive loss $r_+ - r_-$, here we have bolstered that expectation by providing new theorems (Theorem 4 in the revised version) for the soft-triplet loss and InfoNCE ($H=1$, single negative pair as suggested by R5) **without** the constant gradient assumption. It formally shows the existence of a (weighted) covariance operator for the two losses, and comes with an upper bound estimation of the error term.

**Fourth [R4, R5]:** We have greatly expanded both our motivation for and our description of the Hierarchical Latent Tree Model in Section D.2 where we have added a new Figure (Fig. 6), expanded the table of notation, and added more than a page of text to explain the setting. We also add experiments on the growth of the magnitude of the covariance operator during training in HLTM (Appendix G.2)

**Fifth [R4, R5]:** We extended Sec. 4.2 to multi-dimensional outputs and derive the dynamics without the condition $\mathrm{Cov}[\mathbf{u}_j, \mathbf{u}_k] = 0$ (Eqn. 65) under a new setting ($W_2$ is diagonal), which is much easier to achieve. We also perform experiments (e.g., weight growth, and learning of specific Gaussian components) in Appendix G.1.

**Sixth [R5]:** We have extended Lemma 1 to handle more general case of reversible layers, and added an analysis in the presence of $\ell_2$ normalization at the topmost layer (Appendix A.2).

**Seventh [R5]:** We have turned the BYOL sections into rigorous statements (Corollary 2) and added EMA analysis  (Theorem 9).

**Eighth [R4]:** We have added detailed experimental setup information in Appendix G.2 (and Fig. 6) and Appendix G.3.

We hope all of these extensive changes will make reviewers even more positively predisposed to this paper.  Thank you again for a constructive and helpful review process.

---

### Decision · Program_Chairs · 2021-01-07
**Final Decision**

**Decision:**

Reject

**Comment:**

This paper presents new analysis for self-supervised learning. All reviewers are positive about some new perspectives of the analysis. However, some serious concerns have been raised about the rigorousness and the presentation clarity. The paper would be significantly improved, if the authors could address the concerns.